

# Three-Compartment, Two Parameter, Concentration-Driven Model for Uptake of Excess Atmospheric CO$_2$ by the Global Ocean

Stephen E. Schwartz

School of Marine and Atmospheric Sciences
Stony Brook University, Stony Brook NY 11794 USA

*Correspondence to*: Stephen.Schwartz@stonybrook.edu

**Abstract**, This paper develops, applies, and examines a transparent three-compartment model for the amounts of CO$_2$ (dissolved inorganic carbon, DIC) in the mixed-layer and deep ocean, over the Anthropocene driven by the observed amount of atmospheric CO$_2$. The model has two independent parameters, the piston velocity $v_p$ characterizing the rate of water

exchange between the mixed-layer ocean (ML) and the deep ocean (DO), and the atmosphere-ocean deposition velocity for low- to intermediate-solubility gases $k_{am}$. The net uptake of CO$_2$ into the ocean is only weakly dependent on $k_{am}$, so the net uptake rate depends almost solely on $v_p$. This piston velocity is determined from the measured the rate of uptake of heat by the global ocean from the 1960's to the present as $7.5 \pm 2.2$ m yr$^{-1}$, 1-$\sigma$. The resultant modeled net uptake flux of anthropogenic atmospheric CO$_2$ by the global ocean at year 2022 is $2.84 \pm 0.6$ Pg yr$^{-1}$; the corresponding net transfer

coefficient, the net anthropogenic uptake flux divided by the stock of excess atmospheric CO$_2$ is $0.010 \pm 0.002$ yr$^{-1}$. This net transfer coefficient appears to decrease slightly (~17 %) over the Anthropocene, attributed to the decrease of the equilibrium solubility of CO$_2$ (as dissolved inorganic carbon) in seawater due to the uptake of additional CO$_2$ over this period and to increasing slight return flux from the DO to the ML. Modeled DIC in the global ocean and net atmosphere-ocean fluxes compare well with observations and with current carbon cycle models (both concentration-driven and emissions-driven).

Uptake of anthropogenic carbon by the terrestrial biosphere is calculated as the difference between emissions and the sum of increases in atmospheric and ocean stocks. The model is examined for radiocarbon over the industrial era, over the period during which radiocarbon was influenced by emissions of $^{14}$C-free CO$_2$ mainly from fossil fuel combustion, and the period dominated by $^{14}$C emissions from atmospheric weapons testing. A variant of the model with only two compartments and one parameter, $v_p$, treating the atmosphere and the mixed-layer ocean as a single compartment in equilibrium, performs

essentially as well as the three-compartment, two-parameter model. Although the concentration-driven model developed here cannot be used prognostically (to assess model skill in replicating atmospheric CO$_2$ over the industrial period or to examine response to changes in emissions), it is useful diagnostically to examine the disposition of excess carbon into the pertinent global compartments as a function of time over the Anthropocene and for confidently representing ocean uptake of excess CO$_2$ in emissions-driven models.





**30   Short Summary**

The net uptake coefficient of anthropogenic $CO_2$ by the global ocean (net uptake flux divided by excess atmospheric $CO_2$ stock above preindustrial) calculated with a simple, transparent, three-compartment concentration-driven model with two independent parameters is determined to be $0.010 \pm 0.002$ yr$^{-1}$ (net uptake flux at year 2022 $2.84 \pm 0.6$ Pg yr$^{-1}$). This result compares well with observations and with much more complex carbon cycle models.

**Graphical Abstract**

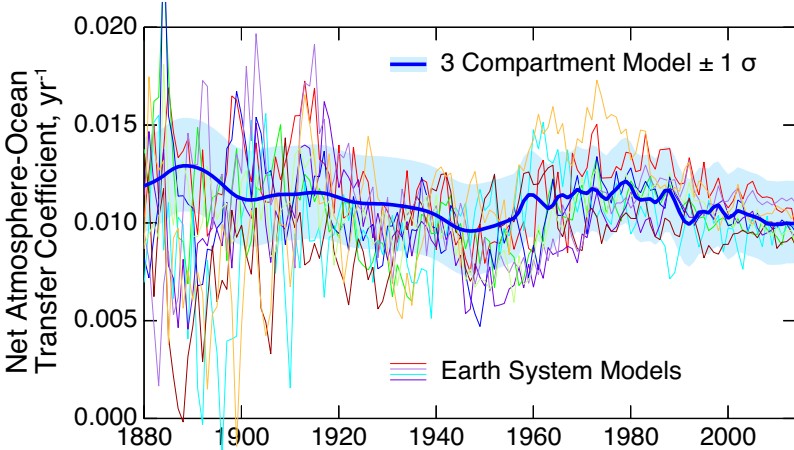



## 1 Introduction

About 250 years ago humankind initiated what has been characterized (Revelle and Suess, 1957; Ramanathan, 1988) as an inadvertent global geophysical experiment by emitting carbon dioxide, $CO_2$, into the atmosphere in conjunction with combustion of fossil fuels, other industrial activities, and changes in land use, thereby changing Earth's climate. Anthropogenic $CO_2$, the amount in excess of preindustrial (PI), affects Earth's radiation budget, with present radiative forcing relative to preindustrial about 2.2 W m$^{-2}$ (Forster et al., 2021), and acidifies the ocean, with the present decrease in

pH of the surface ocean relative to preindustrial about 0.11 (Jiang et al., 2019).

  Much attention has been paid to uptake of excess $CO_2$ by the global ocean because of the importance of this uptake as a sink for anthropogenic $CO_2$. The processes governing this uptake are relatively straightforward and rather well understood: dissolution of $CO_2$ at the air-sea interface and its dependence on the abundance of atmospheric $CO_2$, and transport and mixing of dissolved inorganic carbon as a conservative tracer in the global ocean. Exchange processes between the

atmosphere and the Mixed Layer ocean (ML; upper ocean, roughly 100 m depth) and within the ML are rapid but, because of the small volume of the ML do not contribute greatly to uptake of $CO_2$ by the global ocean. The majority of this uptake is into the Deep Ocean (DO), with the overall rate of uptake controlled mainly by the rates of transport and mixing between the ML and the DO, which takes place on the decadal time scale, (Broecker and Peng, 1974; Oeschger et al., 1975; Sarmiento et al., 1992, Graven et al., 2012). A recent review (Gruber et al., 2023) finds that the current net rate of uptake of $CO_2$ by the

global ocean is $2.7 \pm 0.3$ Pg(C) yr$^{-1}$ (1 Pg = $10^{15}$ g). This net uptake constitutes about 24 % of total anthropogenic emissions from fossil fuel combustion and land-use change, about 11 Pg yr$^{-1}$.

  Historically, studies examining the budget of excess $CO_2$ in the atmosphere and carbon in other reservoirs were based on so-called compartment models, in which the global and annual amounts of excess atmospheric $CO_2$ and carbon derived from atmospheric $CO_2$ were represented in a small number of compartments, up to about a half dozen. Strengths of such models

were that the number of parameters characterizing the transport of carbon between the reservoirs was small (also up to about a half dozen), that such transfer coefficients could be constrained by observations, and that the propagated uncertainties associated with the transfer coefficients could be readily examined. In other words, the models were transparent. In recent decades, especially with the increase in understanding of the processes controlling transfer of carbon among the reservoirs (mainly air-sea exchange, ocean transport, and terrestrial photosynthesis and respiration) together with the increase in

numerical modeling capability afforded by modern computers, the tendency has been toward detailed modeling of the carbon cycle, by so-called carbon cycle models with increasingly high spatial (both horizontal and vertical) and temporal resolutions, representing the rates of processes in tens of thousands of compartments by dozens to hundreds of parameters that in turn depend on numerous situational variables such as temperature and wind speed and, for the terrestrial biosphere (TB), water availability, insolation, and other controlling variables for numerous vegetation types. While this approach has





led to much more detailed representation of the exchange of carbon between the atmosphere and the ocean and the TB, it has resulted in attendant loss of the transparency of the models.

The present study returns to the earlier approach of representing global stocks of anthropogenic carbon in a few global compartments. The present paper is the first in a series that use models with a small number of global compartments to represent the evolution of anthropogenic carbon in the atmosphere and in closely coupled compartments of the biogeosphere.

In this paper a simple concentration-driven model for net uptake of atmospheric $CO_2$ by the global ocean is developed and results presented and compared to observations and other models; the model is driven by measured stock $S_a$ of $CO_2$ in the atmosphere (preindustrial, PI, and as a function of time over the Anthropocene, taken here as commencing at year 1750). Uptake of anthropogenic $CO_2$ by the ML and the DO is actively modeled using transfer coefficients developed here. Uptake into the TB is taken as equal to the difference between total emissions and the uptake and growth in the atmosphere and the

ocean. The anthropogenic stock in the TB and the net flux between the atmosphere and the TB are presented only for reference to the total budget. A central objective of the present study is that the parameters of the model be traceable to observations. Although there are other recent studies that develop and use models with a small number of compartments (e.g. Glotter et al. 2014; Martinez Montero et al., 2022), key parameters in those studies are simply specified as "reasonable" or as "adjusted to match the dynamics of more complex carbon cycle models," and are thus not traceable to observations.

Because the model presented here is concentration-driven, it is diagnostic, not prognostic. That is, the results show the disposition of anthropogenically emitted carbon in the atmosphere, the ML, and the DO, and, by difference, the TB, as function of time over the Anthropocene. Importantly the concentration-driven model cannot be used to calculate the stocks in the several compartments for historical emissions or for prospective future emissions. Nonetheless, to the extent that this concentration-driven model accurately yields this disposition, the representation of the processes in the model can then be

incorporated with confidence into emissions-driven models, as will be treated in subsequent papers.

*Notation.* $S_i$ denotes stock of $CO_2$, of dissolved inorganic carbon (DIC), or of organic carbon, as mass of carbon, C (not $CO_2$) in Pg ($10^{15}$ g), in a given compartment $i$ denoted by subscripts a, m, d, t, for the atmosphere, mixed-layer ocean (ML), deep ocean (DO), and terrestrial biosphere (TB), respectively. $S_i^{ant}$ denotes anthropogenic stock (excess above preindustrial) in compartment $i$. $F_{ij}$, in Pg yr$^{-1}$, denotes gross flux from compartment $i$ to compartment $j$; $F_{ij}^{net}$ denotes net flux from

compartment $i$ to $j$ and is equal to $F_{ij}$ - $F_{ji}$. Transfer coefficients $k_{ij}$, yr$^{-1}$, denote flux $F_{ij}$ per stock of leaving compartment $S_i$; similarly $k_{ij}^{net}$ denotes net flux per anthropogenic stock in the leaving compartment. $Q_{ant}$, in Pg yr$^{-1}$, denotes anthropogenic emission rate, the sum of emissions from fossil fuel combustion and cement manufacture $Q_{ant}^{ff}$ and land use change $Q_{ant}^{lu}$. Subscripts or superscripts pi, pd, and ant denote the preindustrial, present-day, and anthropogenic component of a stock or flux, respectively.





This paper is organized as follows. **Section 2** presents an overview of the compartments comprising the carbon system. **Section 3** develops the transfer coefficients describing the rates of transfer between the several compartments used as input to the model. **Section 4** presents the preindustrial stocks in the several compartments and the time-dependent stock of $CO_2$ in the atmosphere used to force the model calculations. It also presents historical emissions, which, although not directly used in the model calculation, are needed to calculate, by difference, the stock in the terrestrial biosphere and its rate of growth

over the Anthropocene. **Section 5** develops the model and presents results obtained in the calculations for normal $CO_2$ and for radiocarbon and also develops and examines a variant of the model in which the stocks in the atmosphere and the ML are treated in equilibrium. **Section 6** compares the results obtained in the model calculations with results from other model studies and with observations. **Section 7** presents discussion and conclusions. There are two Appendices. **Appendix A** discusses the equilibrium solubility of $CO_2$ in seawater, specifically the dissolution reactions to form bicarbonate and

carbonate ions. **Appendix B** presents the treatment of the rate of mass transfer of $CO_2$ between the atmosphere and the ocean, taking into account the solubility equilibria.

The paper makes extensive use of figures to present the results of the calculations, principally as time series, and to compare to other calculations and to observations. The intent of this paper is to make the model, the reasoning that went into development of the model, and the results, as fully transparent as possible. All the data from the present model calculations

are provided as Supplementary Data in a single Excel Workbook.

**2 Overview**

The framework of the global carbon budget developed here is given in **Fig. 1**. The template for this budget is Figure 7.3 of the Fourth IPCC Assessment Report (AR4, Denman et al., 2007), which, in turn, is based on a figure given initially by Sarmiento and Gruber (2002); values of anthropogenic stocks and fluxes updated to the early 2020's time frame. Similar

figures are presented in the several recent IPCC Assessment reports: the Fifth Report, AR5, (Ciais et al., 2013, Fig. 6.1, Table 6.1); and the Sixth Report, AR6 (Canadell et al., 2021, Fig. 5.12), with antecedents going back at least to AR1, (Watson et al., 1990, Fig. 1.1). This budget is based on observations: PI and present day (PD) values and PD annual increment of atmospheric $CO_2$ mixing ratio, inventories of annual and cumulative emissions, and calculations (ML and DO stocks of dissolved $CO_2$ (DIC)). Calculation of the stocks and fluxes shown in the figure, as a function of time over the

Anthropocene, is a key objective of the present study.



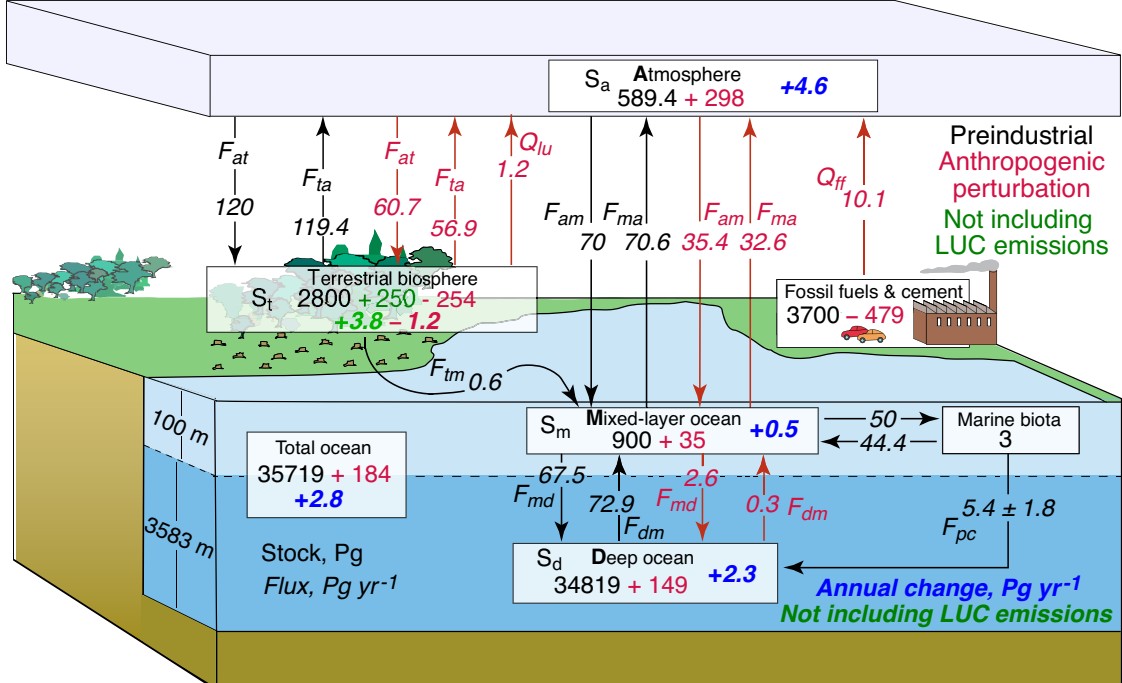

**Figure 1**. Stocks $S_i$ of carbon in principal compartments pertinent to atmospheric $CO_2$ and fluxes between compartments $F_{ij}$ in response to concentration-driven forcing by atmospheric $CO_2$ according to measurements over the Anthropocene. According to the convention introduced by Sarmiento and Gruber (2002) stocks (Pg C) are indicated by upright (roman) type; fluxes and annual changes in stocks (Pg C yr$^{-1}$) are indicated by slant (italic) type. Preindustrial quantities are given in black; perturbations resulting from anthropogenic emissions in red; present-day annual changes in stocks $dS_i/dt$ (Pg C yr$^{-1}$) in blue; or, excluding land-use change (LUC) emissions, in green. $Q_{ff}$ and $Q_{lu}$ denote anthropogenic emissions from fossil-fuel combustion (including cement production) and from net land-use change, respectively. Depths of the ocean mixed layer and of the DO are shown at left. The figure is adapted and modified substantially from AR4 (Denman et al., 2007), with quantities developed here and updated to the early 2020's time frame. Quantities are shown with more precision than is justified by the accuracy with which they are known to permit differencing.

The quantities shown in **Fig. 1** are stocks in each of the several compartments $S_i$ and gross (not net) fluxes $F_{ij}$, positive in the direction of the arrow), the annual changes in several of the compartments, and anthropogenic emissions $Q_{ff}$ and $Q_{lu}$; $Q_{lu}$, the net land-use–change (LUC) emission, represents the net annual carbon flux from the TB into the atmosphere from net



deforestation, *i.e.*, the flux from deforestation minus that from afforestation. There are four compartments pertinent to the distribution of anthropogenic $CO_2$: the atmosphere, the ML, the DO and the TB. By conservation of matter, the sum of the changes in stocks in those compartments is equal to anthropogenic emissions, all as a function of time over the Anthropocene. Several of the quantities in the figure are little more than guesses – the preindustrial stocks in the TB and the
fossil-fuel reserve; current estimates of these quantities vary widely.

For the PI, the atmospheric stock is accurately known from measurements. The ML stock is calculated for assumed near equilibrium between the atmosphere and the ML, for ML depth taken as 100 m; this depth was apparently assumed in previous versions of **Fig. 1**, but seems to be explicitly stated only in the predecessor figure shown in AR3 (Prentice et al., 2001). A mean thermocline depth of 75 m used by several prior investigators, importantly, Bolin and Eriksson (1959),
appears to derive from the early study by Craig (1957) who gave it as 75 ± 25 meters, or, equivalently, 2 % of the volume of the global ocean. The depth of the DO, $z_d$, is obtained as the difference between the mean depth of the global ocean, 3683 m, and the ML depth. The PI ML and the DO are in steady state. This is manifested by the upward PI flux $F_{dm}$ that is due to exchange of water between the ML and the DO slightly exceeding the downward flux $F_{md}$; this is necessary to account for a gravitational flux of biogenic particulate carbon from the ML to the DO, which contributes to the total downward flux. This
"do-nothing" cycle is shown in the figure for completeness, but it plays no role in the analysis here and is neglected in this analysis. The PI flux from the atmosphere to the TB is taken here as equal to the estimate of gross primary production of Beer et al., (2010), but this quantity is highly uncertain; the reverse flux maintains the PI steady state, with a small riverine flux from the TB into the ML ocean, which is then matched by the slight difference in the PI fluxes between the ML and the atmosphere.

The anthropogenic increments in the several stocks and fluxes are the differences between PD and PI values. The principal objective of the present study is to determine the annual changes in the two ocean compartments, as forced by the annual increment in the atmospheric stock $dS_a/dt$, which is obtained from measurement. Attention is called to the PD rate of increase in the total ocean stock, 2.9 Pg yr$^{-1}$, which will be compared with results from measurements and other modeling studies. Of this total annual increment, the great majority, 2.4 Pg yr$^{-1}$ is increase in the stock of the DO, so the processes
controlling this increment are of the greatest interest. Also shown in **Fig. 1**, for reference, are the emissions from fossil fuel combustion (including cement production) and from land-use change, which sum to, 11.3 Pg yr$^{-1}$. The difference between this total annual emission and the sum of the annual increments in the atmosphere, the ML, and the DO is the annual increment in the TB, 3.8 Pg yr$^{-1}$. In constructing the budget shown in **Fig. 1** the anthropogenic flux from the atmosphere to the TB is assumed to scale linearly with the atmospheric $CO_2$ stock; the reverse flux is less than the forward flux as a
consequence of uptake of anthropogenic carbon by the TB. It should be stressed that the fluxes between the atmosphere and the TB shown in the figure are little more than guesstimates, that are presented to give a rough sense of their magnitudes.



These fluxes themselves are not important in this study; rather it is the net fluxes that are important, and, as noted, those fluxes are not actively modeled here but are determined by difference between emissions and actively modeled quantities.

The objective of the present study is to develop and apply a three-compartment concentration-driven model (denoted

3C-CDM) to represent the processes governing the evolution of the system shown in **Fig. 1**. As noted by Melnikova et al. (2023), by ensuring consistent $CO_2$ amounts across models, concentration-driven simulations readily permit consistent comparison of carbon-cycle processes across with different models. The variables included in the 3C-CDM are time dependent global stocks of $CO_2$ (or DIC, or C) in the atmosphere, the ML, the DO, and the TB. The measured stock of atmospheric $CO_2$ is used as a forcing in the set of ordinary differential equations describing the evolution of the system, and the stock in

the TB is not actively modeled but is evaluated, mainly for reference, as the residual between time-dependent integrated anthropogenic emissions and the anthropogenic stocks in the other three compartments. Hence the stocks in the ML and the DO are the only actively modeled quantities. The model requires four parameters: the four transfer coefficients describing rates of transfer between the atmosphere and the ML and between the ML and the DO. However, because it is the same processes that drive exchange in opposite directions, the number of required independent parameters is reduced to two. Use

of a concentration-driven model readily permits examination of the sensitivity to model structure, to parameter values, and to the time-dependent distribution of anthropogenic carbon in the receiving compartments. Importantly, in the concentration-driven model it is possible to readily compare the full model, in which stocks in the ML and the DO are both actively modeled, with an equilibrium variant of the model in which the stock in the ML is treated as in equilibrium with that in the atmosphere. As the atmosphere and the ML are in near equilibrium, treating these two compartments as a single

concentration-driven compartment closely reproduces the results obtained with the stock in the ML being actively modeled, reducing the number of receiving compartments to one, the DO, and thereby reducing the number of independent parameters to one.

In the model the stocks, gross fluxes, and transfer coefficients are related as

$$F_{ij} = k_{ij}S_i$$

(2.1)

The transfer coefficients are taken as constant over the industrial period except for the transfer coefficient from the ML to the atmosphere, $k_{ma}$. As developed in **Section 3**, the transfer coefficients are constrained by rather well characterized and independently measured rates of exchange of material or heat between compartments: an air-sea gas exchange coefficient that is more or less universal for low-to medium solubility gases, and the measured rate of heat uptake by the global ocean over the past several decades, yielding transfer coefficients coupling the ML and the DO. From inspection of **Fig. 1** it

becomes apparent that the most important transfer coefficient is that governing transfer from the ML to the DO, $k_{md}$; the reverse transfer coefficient $k_{dm}$ is of secondary importance, simply because the reverse flux is an order of magnitude smaller





than the downward flux. Hence much attention is given here to determination of $k_{\mathrm{md}}$. The transfer coefficients between the atmosphere and the ML are of relatively minor importance to the evolution of the system because of the near equal and opposite anthropogenic fluxes, indicative of near equilibrium between the compartments, even under the continuing

anthropogenic perturbation. In fact, as noted above and shown in **Section 5**, there is little change in the budget even if the ML is assumed to be in equilibrium with the atmosphere, showing insensitivity to the transfer coefficients.

### 3. Transfer coefficients.

This section develops the transfer coefficients between the ML and the DO and between the atmosphere and the ML.

### 3.1 Transfer between the ML and the DO

Transfer of tracers between the ML and the DO is a physical process, the rate of which is governed by the rate of water volume (or mass) exchange between the two compartments; the tracer just goes along for the ride. The amount of the tracer that is in the compartment having higher concentration is diminished by this exchange and the amount of tracer in the compartment with lower concentration is augmented by this exchange. The fractional rate of exchange relative to the stock in the leaving compartment, $k_{ij}$, dimension $T^{-1}$, is equal to the rate of volume exchange $F_{\mathrm{V}}$, dimension ($L^3\ T^{-1}$), divided by

the volume of the leaving compartment $V_i$, dimension ($L^3$), of the leaving compartment, and is thus of dimension $T^{-1}$.

$$k_{ij} = \frac{\left(\dfrac{dS_i}{dt}\right)}{S_i} = \frac{F_{\mathrm{v}}}{V_i}$$

(3.1)

The rate of volume exchange $F_{\mathrm{V}}$ may be viewed as an area (here the area of the global ocean, $A_{\mathrm{O}}$, dimension $L^2$) times a velocity (dimension L $T^{-1}$), commonly denoted piston velocity $v_{\mathrm{p}}$. Similarly the volume itself is the product of the area times the depth, $z_i$. Hence

$$k_{ij} = \frac{\left(\dfrac{dS_i}{dt}\right)}{S_i} = \frac{F_{\mathrm{v}}}{V_i} = \frac{v_{\mathrm{p}} A_{\mathrm{O}}}{A_{\mathrm{O}} z_i} = \frac{v_{\mathrm{p}}}{z_i}$$


(3.2)

In other words, $k_{\mathrm{md}} = v_{\mathrm{p}} / z_{\mathrm{m}}$ and $k_{\mathrm{dm}} = v_{\mathrm{p}} / z_{\mathrm{d}}$. The question is thus how to obtain an independent measure of $v_{\mathrm{p}}$ based on some tracer other than $CO_2$, to avoid circular reasoning.

The piston velocity quantifying the rate of transfer of water, and by extension of any tracer, between the ML and the DO is of great importance here and in many geophysical applications. An early determination of this quantity, 3 to 3.5 m yr$^{-1}$, was

obtained from the difference in the ratio of $^{14}CO_2$ to $^{12}CO_2$ in the upper ocean versus that in the DO, using the half-life of $^{14}CO_2$ as a clock (Broecker and Peng, 1982, pp. 236-243; also Sarmiento and Gruber, 2006, hereinafter SG06, p. 12). Using a global inverse model, DeVries *et al.*, 2017 quantified temporal variations in basin-wide and global scale volume exchange





rate between the ML and the DO, finding global-mean volume exchange rate, expressed as piston velocity, 4.98, 6.29, and 4.28 m yr$^{-1}$ for the 1980's, 1990's and 2000's, as respectively.

The approach taken here comes from recognition that the piston velocity governing exchange of $CO_2$ between the ML and the DO is the same as that governing the flux of heat energy from the ML to the DO that has been induced by the increase in global mean surface temperature over the Anthropocene era, especially given the similar time history of these perturbations. The basis for determination of the flux between the ML and the DO in the present study is the measured rate of heat uptake by the global ocean in response to the increase in global temperature over the past 50 years. The so-called "nexus" between

uptake of heat and $CO_2$ by the global ocean was highlighted in the IPCC Sixth Assessment Report (Monteiro et al, 2021), which calls attention to the commonality of the transport processes that govern ocean uptake of excess $CO_2$ and heat. Bronselaer and Zanna (2020) found a linear relation between the increase in the heat content in the global ocean, as determined by direct measurement, and the increase of global DIC, obtained by an inverse carbon cycle model based on assimilation of potential temperature, salinity, radiocarbon, and CFC-11 observations (DeVries, 2014); the data present by

Bronselaer and Zanna (2020) for the time period 1965-2017 exhibit a coefficient of determination ($r^2$) between excess heat and excess ocean carbon stock of 0.991. Although that relation cannot be used to obtain a measure of the transfer flux between the ML and the DO, because it deals with the total ocean heat content, not the ML and DO separately, and because the developed relation characterizes the response in ocean heat content to all forcings, not just $CO_2$ forcing, the tight relation shown in that study lends strong support to the approach taken here of using the rate of heat transfer from the ML to the DO

together with the increase in global mean surface temperature to determine the piston velocity pertinent to transfer of any tracer, importantly including DIC, between the two compartments.

**Figure 2** shows the several quantities needed to evaluate the rate of increase of the heat content of the global ocean per increase in global temperature anomaly,

$$\kappa_{\mathrm{H}} = \frac{d\frac{dH_{\mathrm{O,tot}}}{dt}}{dT},$$

(3.3)

where $H_{\mathrm{O,tot}}$ denotes the heat content anomaly of the total global ocean, that is the sum of that for the ML and the DO. **Figure 2a** shows a roughly linear increase in global mean surface temperature anomaly (GMST, $\Delta T$; GISTEMP Team, 2024) with time over the period 1960 to the present. **Figure 2b** shows the increase in global ocean heat content $\Delta H_{\mathrm{O,tot}}$ over the same time period, as presented in a recent review of observational data by Cheng et al (2024). **Figure 2c** shows the time derivative of $\Delta H_{\mathrm{O,tot}}$, $dH_{\mathrm{O,tot}}/dt$; like GMST $dH_{\mathrm{O,tot}}/dt$ exhibits a linear increase with time. The slope of the fit in **Fig. 2d**

of $dH_{\mathrm{O,tot}}/dt$ vs $T$ gives the desired quantity $\kappa_{\mathrm{H}}$, having value 13.8 ± 3.2 ZJ yr$^{-1}$ K$^{-1}$, equivalent to 1.20 ± 0.28 W m$^{-2}$ K$^{-1}$; throughout this paper uncertainties are given as 1-$\sigma$. This slope exhibits considerable uncertainty, 24 %, a manifestation of the rather large fluctuations of $dH_{\mathrm{O,tot}}/dt$ shown in **Fig. 2c and 2a**, the latter due mainly to fluctuations in reported values of $\Delta H_{\mathrm{O,tot}}$ (**Fig. 2b**). Those fluctuations may be actual fluctuations in this quantity occurring on a time scale of a few years (as with $\Delta T$), or may be artifacts arising from errors in the measurements themselves, or from inadequacies in spatial coverage, or both.



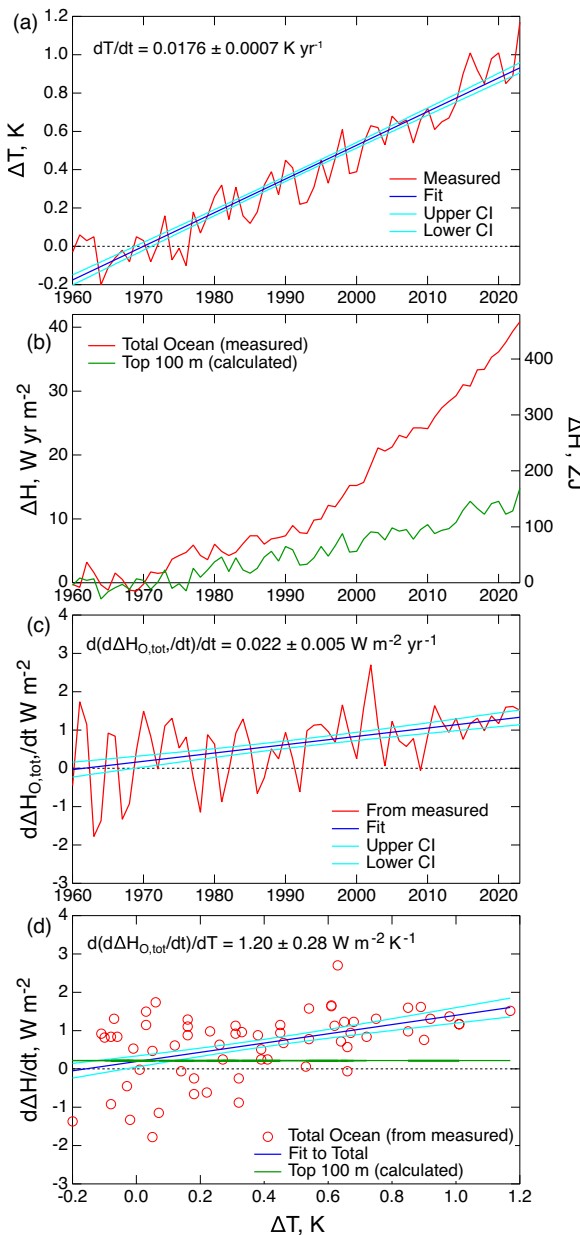

**Figure 2**. Time dependence of quantities required to quantify uptake of heat by the global ocean over the years 1960-2023. (a) Global mean surface temperature anomaly $\Delta T$, (GISTEMP Team, 2024). (b) Heat content anomaly of the



global ocean $\Delta H_{\text{O,tot}}$, referenced to 1960, from the assessment of measurements by Cheng et al. (2024) and as calculated for the top 100 m of global ocean, $\Delta H_{\text{ML}}$, for assumed thermal equilibrium with global temperature

anomaly $\Delta T$; right-hand scale gives $\Delta H$ in systematic (SI) units, as employed by Cheng et al., where the prefix Z denotes $10^{21}$; left-hand scale gives $\Delta H$ in unit more readily related to the energy budget of the global ocean. per area of the global ocean $A_{\text{O}}$. (c) Time derivatives $d\Delta H_{\text{O,tot}}/dt$, based on measured $\Delta H_{\text{O,tot}}$, and calculated $d\Delta H_{\text{ML}}/dt$, based on slope $d\Delta T/dt$ in $a$. (d) Time derivatives $d\Delta H_{\text{O,tot}}/dt$ and $d\Delta H_{\text{ML}}/dt$, plotted against $\Delta T$. Confidence intervals (CI) and fitting coefficients, 68 % CI, are shown with the fits.

The total ocean heating rate is the sum of two components, the heating rate of the ML and the heating rate of the DO, the latter corresponding to the transfer flux of heat from the mixed layer to the deep ocean,

$$\frac{d\dfrac{dH_{\text{O,tot}}}{dt}}{dT} = \frac{d\dfrac{dH_{\text{m}}}{dt}}{dT} + \frac{d\dfrac{dH_{\text{d}}}{dt}}{dT} \quad , \tag{3.4}$$

where the second term on the right is the quantity of principal interest here to obtain the piston velocity pertinent to transfer of tracers from the ML to the DO. **Table 1** outlines the calculations leading to evaluation of this piston velocity. In

customary units $d\left(dH_{\text{O,tot}}/dt\right)/dT = 1.207$ W m$^{-2}$ K$^{-1}$ referred to the area of the global ocean (row 4 of table). The rate of increase of the heat content of the ML is evaluated as the rate of increase of global temperature times the heat capacity of the ML ($z_{\text{m}}$ times the volumetric heat capacity of seawater, $C_{\text{vol}}^{\text{sw}}$),

$$\frac{dH_{\text{m}}}{dt} = \frac{dT_{\text{m}}}{dt} C_{\text{vol}}^{\text{sw}} z_{\text{m}} \quad . \tag{3.5}$$

(Here the assumption of thermal equilibrium between the ML and the atmosphere is not required, but only the weaker

assumption that the rate of increase of temperature is the same for both compartments.). For $dT_{\text{m}}/dt = 0.0176 \pm 0.0007$ K yr$^{-1}$ (**Fig 2(a)**), $C_{\text{vol}}^{\text{sw}} = 4.11 \times 10^{6}$ J m$^{-3}$ K$^{-1}$, and $z_{\text{m}}$ again taken as 100 m, $d\left(dH_{\text{ML}}/dt\right)/dT = 0.229$ W m$^{-2}$ (row 8); this quantity must be subtracted from the total $d\left(dH_{\text{O,tot}}/dt\right)/dT$ to yield the increase in deep ocean heating per increase in GMST, $\kappa_{H} = d\left(dH_{\text{DO}}/dt\right)/dT = 0.976$ W m$^{-2}$ K$^{-1}$, a quantity that is of broad geophysical interest beyond its application here. In turn the piston velocity, which is related (row 10) to the heat transfer coefficient as

$$v_{\text{p},H} = \kappa_{H}/C_{\text{vol}}^{\text{sw}} \quad , \tag{3.6}$$

is 7.50 m yr$^{-1}$, $\pm$ 30 % 1-$\sigma$. The piston velocity determined in this way from measurements of the heating rate of the global ocean provides an independent, observationally based measure of this quantity (which is similarly of broad geophysical interest) that can be used with confidence in determining components of global $CO_2$ budget. The corresponding transfer coefficients are



$$k_{md} = v_p / z_m = 0.075 \ \text{yr}^{-1} \quad \text{and} \quad k_{dm} = v_p / z_d = 0.0021 \ \text{yr}^{-1},$$


(3.7)

both likewise uncertain to ±30 %.

**Table 1**. Evaluation of piston velocity and transfer coefficients between Mixed Layer Ocean and Deep Ocean.

| | Quantity | Unit | Value | Uncertainty (1-σ) |
|---|---|---|---|---|
| 1 | $d\dfrac{\frac{dH_{tot}}{dt}}{dT}$ | J yr$^{-1}$ K$^{-1}$ | 13.76 E21 | 3.25 E21 |
| 2 | | s yr$^{-1}$ | 3.156 E7 | |
| 3 | $A_O$, area of global ocean | m$^2$ | 3.619 E14 | |
| 4 | $d\dfrac{\frac{dH_{tot}}{dt}}{dT}$ | W m$^{-2}$ K$^{-1}$ | 1.205 | 0.283 (23.5 %) |
| 5 | $\dfrac{dT}{dt}$ | K yr$^{-1}$ | 0.0176 | 0.0007 |
| 6 | $C_{vol}^{sw}$ | J m$^{-3}$ K$^{-1}$ | 4.11 E6 | |
| 7 | $z_m$ | m | 100 | |
| 8 | $d\dfrac{\frac{dH_m}{dt}}{dT}$ | W m$^{-2}$ K$^{-1}$ | 0.229 | |
| 9 | $\kappa_H = d\dfrac{\frac{dH_d}{dt}}{dT} = d\dfrac{\frac{dH_{O,tot}}{dt}}{dT} - d\dfrac{\frac{dH_m}{dt}}{dT}$ | W m$^{-2}$ K$^{-1}$ | 0.976 | 0.233 |
| 10 | $v_{p,H} = \dfrac{\kappa_H}{C_{vol}^{sw}}$ | m yr$^{-1}$ | 7.50 | 2.17 (30 %) |
| 11 | $z_d$ | m | 3583 | |
| 13 | $k_{md} = v_p / z_m$ | yr$^{-1}$ | 0.075 | 30 % |
| 14 | $k_{dm} = v_p / z_d$ | yr$^{-1}$ | 0.00209 | 30 % |

The values of the transfer coefficients $k_{md}$ and $k_{dm}$ obtained in this analysis can be compared to values inferred from the budgets depicted in versions of **Fig. 1** presented in recent IPCC assessment reports, as shown in **Table 2**. Here the values of

$k_{md}$ and $k_{dm}$ are obtained from the preindustrial stocks and gross fluxes shown in the indicated figures in the several assessment reports by inversion of **Eq 3.1**, specifically for the preindustrial fluxes between the ML and the DO. as



$$k_{md} = F_{md}^{pi} / S_m^{pi} \quad \text{and} \quad k_{dm} = F_{dm}^{pi} / S_d^{pi}.$$

(3.8)

Also shown in **Table 2** are the piston velocities obtained from the transfer coefficients obtained by inversion of **Eq 3.7**. The PI stocks and fluxes are essentially unchanged between the 2007 and 2013 reports. Values of $v_p$ inferred from the fluxes presented in these reports, 10 m yr$^{-1}$, are somewhat greater than that inferred here, $7.5 \pm 2.2$ m yr$^{-1}$, slightly greater than the 1-$\sigma$ range. In contrast, the value for $v_p$ given in the 2021 report is a factor of 2.9 higher than values in the earlier IPCC reports. This value, which cannot be correct, would seem to result from a misreading of the results of a single paper by the authors of the pertinent chapter of AR6. The values and uncertainty range for $v_p$ obtained here are used in the model calculations presented here. As seen in **Section 5**, in the examination of the sensitivity of modeled stock in the DO to uncertainty in piston velocity, a greater piston velocity such as is inferred from the Fourth and Fifth Assessment reports, relative to that determined here, would result in a greater modeled stock in the DO over the Anthropocene, relative to that determined here, by about 30 %.

**Table 2**. Transfer coefficients and piston velocities between the Mixed Layer Ocean and the Deep Ocean inferred from recent IPCC Assessment Reports and determined in this study.

| Quantity | Unit | AR4 (2007) Fig 7.3* | AR5 (2013) Fig 6.1 | AR6 (2021) Fig 5.12 | This study| |
|---|---|---|---|---|---|
| $S_m^{pi}$ | Pg | 900 | 900 | 900 | 900 |
| $F_{md}^{pi}$ | Pg yr$^{-1}$ | 90.2 | 90 | 264 | 67.5 |
| $k_{md}$ | yr$^{-1}$ | 0.100 | 0.100 | 0.293 | 0.075 |
| $z_m$ | m | 100 | 100 | 100 | 100 |
| $v_p$ | m yr$^{-1}$ | 10.0 | 10 | 29.3 | 7.5 |

*Identical values for Sarmiento and Gruber (2002).

Here it should be underscored that the fundamental measure of the rate of transfer of DIC from the ML to the DO (or from the DO to the ML) is the flux. The transfer coefficient $k_{md}$ is inversely proportional to $z_m$, whereas the stock in the ML is proportional to $z_m$; specifically $S_m = C_m z_m A_O$; both $C_m$, the concentration of DIC in the ML, and $A_O$ are independent of $z_m$. Hence

$$F_{md} = k_{md} S_m = \frac{v_p}{z_m} C_m z_m A_O = v_p C_m A_O,$$

(3.9)

from which it is seen that the piston velocity, not the transfer coefficient, is the fundamental parameter governing the fluxes between the ML and the DO.



**3.2 Transfer coefficients coupling the atmosphere and ML**

**3.2.1 Transfer from the atmosphere to the ML**

Gas exchange from the atmosphere to the ML ocean for all low- to medium-solubility gases, being controlled by mass transport on the water side of the atmosphere-ocean interface, is governed by the same physics, namely turbulent transfer. (For higher-solubility gases the mass transport rate is increasingly influenced by the rate of turbulent transfer in the atmosphere; Schwartz, 1992). The gross (one-way, not net) mass transport flux of a soluble gas, $\phi$, between the gaseous and aqueous phases is related to the concentration on the leaving side as

$$\phi_{ij} = v_{ij}c_i$$,
                                                                                                                  (3.10)

where $\phi_{ij}$ is the gross flux from phase $i$ to phase $j$ having dimension of amount (or mass) per area and time, and $c_i$ denotes the concentration in the leaving side, respectively amount or mass concentration, and $v_{ij}$ is a mass-transfer coefficient, (dimension $L\ T^{-1}$); because of this dimension the mass-transfer coefficient is commonly denoted a velocity.

The water-side mass-transfer coefficient $v_{\text{ma}}$ depends strongly on turbulent mixing, which in turn, in the atmosphere-ocean
environment depends on the rate of turbulent mixing on the water side. The diffusivity of the dissolved gas in seawater plays a lesser role in the rate of mass transfer, the greater the diffusion coefficient, the greater the transfer coefficient. These dependences have been thoroughly studied in the laboratory and the ocean environment, and are rather well characterized and understood, (e.g., SG06). Because the controlling physics is the same for all gases, the water-side mass transfer coefficient characterizing transport of low- to medium-solubility gases into seawater can be considered a more-or-less
universal quantity, albeit one that is not all that precisely known. SG06 examine laboratory studies and field measurements, e.g. with the long-lived (5730-yr half life) isotopic variant of $CO_2$, $^{14}CO_2$, and short-lived (3.8 day) $^{222}$radon. Multiple formulations give comparable values and wind speed dependence. Specializing to $CO_2$, that assessment concludes for preindustrial mixing ratio of $CO_2$ and suitable global-mean temperature and ocean salinity SG06 the global- annual-mean gas exchange velocity for water-side mass transport of $CO_2$, expressed in terms of the concentration of $CO_2$ (not DIC) on
the water side of the interface is about 17 cm hr$^{-1}$, and the transport coefficient represents gross flux, not net flux, which would depend on the difference between concentrations in the two phases. The several prior studies compared by SG06 give values of $v_{\text{ma}}$ that range from 11.2 to 18.7 cm hr$^{-1}$. These estimates result from averages of local and instantaneous transfer velocity, a quantity that depends strongly on wind speed (roughly as the second power) and to lesser extent on other situational variables (*e.g.*, Liss and Merlivat, 1986; Takahashi *et al.*, 2009; Wanninkhof *et al.*, 2013).

To evaluate gross flux from the atmosphere to the ocean it is desired to express the mass-transport coefficient in terms of gas-phase $CO_2$ concentration, and ultimately to atmospheric $CO_2$ stock. The transfer coefficient between the atmosphere





and the ML referred to gas-phase $CO_2$ $v_{am}$ is related to the transport coefficient between the two compartments referred to aqueous-phase $CO_2$, $v_{ma}$, by the dimensionless volumetric Henry's law solubility coefficient $H_s^{cc}$; here the notation of Sander *et al.* (2021) is used, where the subscript s denotes solubility (aqueous per gas, to distinguish from volatility, gas per

aqueous) and the superscripts c refer to concentration (amount per volume, to distinguish from mixing ratio, mole fraction and other units) in the two phases (aqueous concentration per gas concentration), as

$$v_{am} = v_{ma} H_s^{cc} \tag{3.11}$$

(*e.g.*, Liss and Slater, 1974; Schwartz, 1992). For $CO_2$, $H_s^{cc}$ is near unity (0.83 at 291 K; Weiss, 1974; Lewis and Wallace, 1998). For $v_{ma}$ taken as 17 cm $hr^{-1}$ the resulting global- and annual-mean value of $v_{am}$ is 14.2 cm $hr^{-1}$, again with

uncertainty ±30 %. Converting to conventional units taking into account the area of the global ocean $A_O$ ($3.619 \times 10^{14}$ $m^2$; Eakins and Sharman, 2012) and the volume of the atmosphere $V_A$,

$$k_{am} = \frac{A_O}{V_A} v_{am}, \tag{3.12}$$

where $V_A$ is evaluated from the number of moles in the global atmosphere $1.765 \times 10^{20}$ mol (Prather et al., 2012), yields the transfer coefficient $k_{am} = 0.107$ $yr^{-1}$. This value is essentially the same that can be inferred from the ratio of preindustrial

$F_{am}$ / $S_a$ given in the predecessor figure to **Fig. 1**, (Sarmiento and Gruber, 2002), 0.119 $yr^{-1}$, which evidently results from similar reasoning. Estimates of $k_{am}$ as inferred from recent IPCC Assessment Reports have varied slightly, **Table 3**. Here the value $k_{am} = 0.119$ $yr^{-1}$ from Sarmiento and Gruber (2002) is retained here to avoid proliferation of values; as shown in **Section 5** the evolution of DIC over the industrial period is quite insensitive to the value of $k_{am}$, so the range of values of $k_{am}$ given in **Table 3** is of little consequence. The transfer coefficient $k_{am}$, being an intensive quantity, denoting the global-

and-annual mean gross transfer rate of $CO_2$ from the atmosphere to the surface ocean divided by the amount of $CO_2$ in the global atmosphere, is independent of the amount of $CO_2$ in the atmosphere and of the amount or concentration of dissolved $CO_2$ in the ocean. The resulting gross flux from the atmosphere to the ML

$$F_{am} = k_{am} S_a, \tag{3.13}$$

scales linearly with the atmospheric stock.



**Table 3.** Atmosphere-ocean transfer coefficient (deposition velocity) for $CO_2$ inferred from recent assessments.

| | Quantity | Unit | Sarmiento and Gruber 2002 | AR4 (2007) Fig 7.3 | AR5 (2013) Fig 6.1 | AR6 (2021) Fig 5.12 |
|---|---|---|---|---|---|---|
| PI atmospheric stock | $S_a^{pi}$ | Pg | 590 | 597 | 589 | 591 |
| Gross PI transfer flux | $F_{am}^{pi}$ | Pg yr$^{-1}$ | 70 | 70 | 60 | 54.0 |
| Gross transfer coefficient | $k_{am}$ | yr$^{-1}$ | 0.1186 | 0.1173 | 0.1019 | 0.0914 |

### 3.2.2 Transfer from the ML to the atmosphere

For transfer of $CO_2$ from the ML to the atmosphere the situation is considerably more complicated because the substance that crosses the interface is $CO_2$, whereas the dissolved substance, the stock of which is the modeled quantity, consists of a mixture of hydrated $CO_2$ and the bicarbonate (mainly) and carbonate ions, collectively dissolved inorganic carbon, DIC. The

dissociation equilibria relating the concentrations of these species are well characterized and understood (e.g., SG06), with the effect that the solubility of $CO_2$ in seawater and the transfer coefficient characterizing the rate of transfer of DIC from seawater to the atmosphere are dependent on pH or alternatively, on the stock of DIC in the ML. It is desired to express the fluxes in both directions in terms of the stocks. This situation is readily dealt with using well established equilibrium relations between concentrations of DIC and $H_2CO_3$ but has the consequence that the transfer coefficient $k_{ma}$ is dependent

on the concentration of DIC and is thus not a constant in the same sense as $k_{am}$. The approach taken here is developed in **Appendix B**. Summarizing the results of the derivation given there, the return flux from the ML to the atmosphere is given in terms of the transfer coefficient for the forward flux $k_{am}$ times the stock in the ML $S_m$ and a differential equilibrium constant $K'_{ma}(S_m)$ , where the dependence on $S_m$ is explicitly noted, as

$$F_{ma}(S_m) = k_{am} \cdot K'_{ma}(S_m)S_m \tag{3.14}$$

where

$$K'_{ma}(S_m) = \left( \frac{dS_a}{dS_m} \right)_{eq} . \tag{3.15}$$

Defining an effective transfer coefficient $k'_{ma}$

$$F_{ma}(S_m) = k'_{ma}S_m \tag{3.16}$$

then

$$k'_{ma} = F_{ma}(S_m)/S_m = k_{am} \cdot K'_{ma}(S_m) . \tag{3.17}$$

Noting that

$$\left( \frac{dS_a}{dS_m} \right)_{eq} = \frac{V_A}{A_O z_m H_s^{cc}} \frac{d[H_2CO_3]}{d[DIC]} , \tag{3.18}$$



then

$$K'_{ma}(S_m) = \frac{V_A}{A_O z_m H_s^{cc}} \frac{d[H_2CO_3]}{d[DIC]}$$

(3.19)

and in turn

$$k'_{ma} = k_{am} \left( \frac{dS_a}{dS_m} \right)_{eq} = k_{am} \frac{V_A}{A_O z_m H_s^{cc}} \frac{d[H_2CO_3]}{d[DIC]} .$$

(3.20)

This expression is used in the differential equations for describing the evolution of the system in response, here, to the concentration driven forcing.

## 4. Historical atmospheric CO$_2$, Preindustrial stocks, Emissions

### 4.1 Atmospheric CO$_2$

The 3C-CDM is driven by observed atmospheric CO$_2$ stock, $S_a$, obtained from the global mean mixing ratio of atmospheric CO$_2$ by the conversion factor 2.120 Pg ppm$^{-1}$. Here the atmospheric mixing ratio of CO$_2$, $x_{CO2}$, is taken as the integral of the annual growth rate of CO$_2$ mixing ratio presented in the historical tab of the data table accompanying the 2023 Global Carbon Budget (Friedlingstein et al., 2023; GCB23). Values of $x_{CO2}$ in the earlier part of the modeled period (1750 – 2022) were obtained from measurements in air trapped in ice in time-resolved cores taken in Antarctica, and in the later part of the period (subsequent to 1959) from direct measurements of CO$_2$ in air, initially at Mauna Loa and the South Pole, and subsequently at multiple locations. The several measurement data sets, which are in close agreement during the overlap periods, can be used with high confidence. Preindustrial $x_{CO2}$ was taken as 278 ppm at year 1750. The time-dependent anthropogenic atmospheric stock $S_a^{ant}(t)$ is obtained as the difference $S_a(t) - S_a^{pi}$.

### 4.2 Preindustrial stocks

Preindustrial stocks are required to initiate the model. In principle if the system were linear the modeled quantities could be the anthropogenic perturbations to the stocks and all initiated as zero. However, as noted, the equilibrium solubility of CO$_2$ in seawater is nonlinear, depending on the amount of DIC in the seawater, and this nonlinear dependence of CO$_2$ solubility must be accounted for in the model. The nonlinearity is dramatically illustrated in **Fig. 1**. In the preindustrial state the ratio of DIC in the ML to atmospheric CO$_2$ is 900/589.4 or roughly 1.5. In contrast for the anthropogenic increment the ratio is 35/298, or roughly 0.12, more than an order of magnitude less. Hence it is necessary to use actual measured CO$_2$ to drive the model and to calculate the anthropogenic stock in the ML as the difference between the calculated ML stock and the PI ML stock, taken here as the stock that would be in equilibrium with the PI atmospheric stock.





The PI stock in the ML is in near equilibrium with the stock in the atmosphere. Under PI conditions these two compartments would be in equilibrium except for a slight departure that is due to uptake of carbon by the TB that is delivered to the ML ocean by riverine fluxes, resulting in a slight excess of DIC in the ML above its value that would be in equilibrium with the atmosphere. This non-equilibrium situation under PI conditions is neglected here as it does not affect anthropogenic stocks and fluxes examined in this model, but the associated net PI flux (0.6 Pg yr$^{-1}$, **Fig. 1**) must be taken into account when comparing calculations to measurements. A second departure from equilibrium at PI conditions, also neglected here, is due to particulate carbon resulting from biological activity sinking from the ML to the DO, where it is oxidized and ultimately returned to the ML via exchange of water between the two compartments, resulting in the concentration of DIC in the DO being greater than that in the ML, again neglected here. However this model explicitly evaluates the net transfer of anthropogenic DIC from the ML to the DO that occurs by exchange of water between the two compartments, using the coefficients characterizing the exchange between the two compartments (mainly from the ML to the DO) developed in Section 3.

Because of nonlinearity in the relation between DIC in the ML and atmospheric $CO_2$ stock, the model explicitly calculates total ML DIC stock rather than anthropogenic stock; time-dependent anthropogenic $CO_2$ stock is evaluated as the difference between modeled total DIC and PI DIC. Anthropogenic stock in the DO is set to zero at initiation of the model run, and because the exchange between the ML and DO is linear in stocks the model simply calculates anthropogenic stock directly from the net exchange between the two compartments.

### 4.3 Emissions

Although anthropogenic emissions are not required to drive the 3C-CDM, they provide context for the fraction of emissions that is present in the atmosphere (determined by measurement), the fraction taken up by the global ocean as modeled here (and by others), as well as by a few measurements, and the fraction taken up by the terrestrial biosphere (obtained by difference). Historical anthropogenic emissions over the Anthropocene consist of the sum of emissions from fossil-fuel combustion (including cement production) and land-use–change emissions. The GCB23 data file presents LUC emissions only subsequent to 1850, by which time these emissions are already substantial, and, for that matter, substantially greater than fossil-fuel emissions. In the absence of reported emissions prior to 1850, in the present study emissions between 1750 and 1850 were taken as a linear interpolation between 0 at 1750 and the GCB23 value at year 1850 (0.72 Pg yr$^{-1}$). This results in a slightly greater value of integrated emissions in the present study than given by GCB23. These data are also shown in **Section 5**.





### 5. Model and Results

The model developed and presented here consists of a set of two coupled ordinary differential equations in the stocks in the ML ocean and the DO. The model is forced by the increasing values of atmospheric stock $S_a$. The gross rate of transfer from the atmosphere to the ML is given simply as the product of the transfer coefficient $k_{am}$ times the atmospheric stock $S_a$

$$F_{am} = k_{am}S_a$$
(5.1)

However, as developed in **Appendix B**, because the transferred entity, $CO_2$, is not related in constant proportion to the stock in the ML, $S_m$, the reverse transfer rate, expressed in terms of $S_m$, must account for the shift in the equilibrium relation between $CO_2(aq)$ and DIC. The equilibrium relation between $S_m$ and $S_a$ is

$$K_{ma} = \left( \frac{S_a}{S_m} \right)_{eq} = \frac{V_A}{A_O z_m H_s^{cc}} \left( \frac{[CO_2(aq)]}{[DIC]} \right)_{eq},$$
(5.2)

permitting calculation of $S_m$ for a specified value of $S_a$. Here $K_{ma}$ is not a true equilibrium constant (in the physical chemistry sense) because it incorporates the geophysical quantities $V_A$, the volume of the atmosphere at standard conditions, $A_O$, the area of the global ocean and $z_m$, the depth taken for the ocean mixed layer, in addition to the Henry's law solubility of $CO_2$, $H_s^{cc}$ (see **Section 3.2**), and the equilibrium ratio of aqueous $CO_2$ to total DIC. The net transfer rate is evaluated in terms of the departures from phase equilibrium in the two stocks as

$$F_{am}^{net} = k_{am}(S_a - S_a^{eq}) - k_{am}K'_{ma}(S_m - S_m^{eq})$$
(5.3)

Here the equilibrium amounts of the stocks in the two phases are based on the sum of the stocks in the two phases, and $K'_{ma}$ is a "differential" equilibrium constant relating the two phases analogous to $K_{ma}$ but in terms of the derivative relations between $S_a$ and $S_m$.

$$K'_{ma} = \left( \frac{dS_a}{dS_m} \right)_{eq} = \frac{V_A}{A_O z_m H_s^{cc}} \left( \frac{d[CO_2]}{d[DIC]} \right)_{eq}.$$
(5.4)

$K_{ma}$ and $K'_{ma}$ and the equilibrium stocks $S_a^{eq}$ and $S_m^{eq}$ are readily evaluated for specified $S_a$, here by the program CO2SYS (Lewis and Wallace, 1998). In addition to the concentration of DIC and the geophysical quantities $V_A$ and $A_O$, $K_{ma}$ and $K'_{ma}$ depend on multiple situational variables, importantly temperature and alkalinity, both taken as constant over the industrial period (here taken as global-annual mean surface temperature, 18 ˚C, and alkalinity 2349 μmol $kg_{sw}^{-1}$).

As transfer between the ML and the DO is based on the physical transfer rate, the net transfer rate between the ML and the DO is simply,

$$F_{md}^{net} = k_{md}S_m - k_{dm}S_d$$
(5.5)



The set of differential equations to be solved is for the stock in the ML and the anthropogenic stock in the DO,

$$\frac{dS_{\mathrm{m}}}{dt} = k_{\mathrm{am}}(S_{\mathrm{a}} - S_{\mathrm{a}}^{\mathrm{eq}}) - k_{\mathrm{am}}K_{\mathrm{ma}}'(S_{\mathrm{m}} - S_{\mathrm{m}}^{\mathrm{eq}}) - k_{\mathrm{md}}S_{\mathrm{m}}^{\mathrm{ant}} + k_{\mathrm{dm}}S_{\mathrm{d}}^{\mathrm{ant}}$$

$$\frac{dS_{\mathrm{d}}^{\mathrm{ant}}}{dt} = k_{\mathrm{md}}S_{\mathrm{m}}^{\mathrm{ant}} - k_{\mathrm{dm}}S_{\mathrm{d}}^{\mathrm{ant}}$$

(5.6)

,

driven by the external forcer $S_{\mathrm{a}}(t)$. The initial conditions are $S_{\mathrm{m}}(0) = S_{\mathrm{m}}^{\mathrm{pi}}$ and $S_{\mathrm{d}}^{\mathrm{ant}}(0) = 0$, with $S_{\mathrm{m}}^{\mathrm{ant}} = S_{\mathrm{m}} - S_{\mathrm{m}}^{\mathrm{pi}}$, where $S_{\mathrm{m}}^{\mathrm{pi}}$
denotes the preindustrial stock in the ML, evaluated as in equilibrium with the preindustrial atmospheric stock $S_{\mathrm{a}}(0)$.

This set of two differential equations is readily solved by an ordinary differential equation (ODE) solver, here the Igor
package (www.wavemetrics.com). The results are shown in **Fig. 3** in three ways, all as the anthropogenic increments in the
several quantities. **Figure 3a** shows the stocks (and also, for comparison, integrated emissions) as a function of time
commencing at year 1750. The atmospheric stock $S_{\mathrm{a}}^{\mathrm{ant}}(t)$ that is used to drive the model, which is based on measurements,
is from the GCB23 project (Global Carbon Budget, Friedlingstein et al., 2023), with time 0 taken as 1750 and initial
atmospheric stock taken as $S_{\mathrm{a}}(0) = 2.120$ Pg ppm$^{-1} \times 278$ ppm $= 589.36$ Pg. The integrated emissions, shown for reference,
is obtained from inventories as given by GCB23; here the net emissions for land-use change for years prior to 1850, which
were not given by GCB23, are obtained by linear interpolation between 0 in 1750 and 0.72 Pg yr$^{-1}$ in 1850, at which time
they are substantial and substantially exceed fossil fuel emissions. The total anthropogenic increase in ocean stock, $S_{\mathrm{o}}^{\mathrm{ant}}$,
which is the quantity that is commonly reported in other model studies (and in measurements), is the sum of the
anthropogenic stocks in the ML and the DO, which are the quantities modeled here (and which are shown separately in the
graph). Also shown by the light blue band is the $\pm 1\sigma$ range of uncertainty associated with the modeled stock due to
corresponding uncertainty range in the piston velocity $v_{\mathrm{p}}$, readily evaluated by using the corresponding $1\sigma$ uncertainty values
of $v_{\mathrm{p}}$. The anthropogenic stock in the TB due to net transfer from the atmosphere $S_{\mathrm{t}}^{\mathrm{ant}}$ is obtained by difference between
integrated total emissions (fossil plus biogenic) and the sum of the atmospheric and ocean stocks and thus does not include
the decrease in terrestrial stock due to land-use–change emissions. The numbers at the right give the fraction of integrated
emissions in the several compartments at the end of the model run. Attention is called to the total stock in the ocean being
dominated by the stock in the DO, the stock in the ML being a relatively minor contribution to the total, analogous to heat,
**Fig. 2b**. Also shown, for comparison discussed below, is the modeled ocean stock as obtained by integration of the net fluxes
given in the historical tab of GCB23.



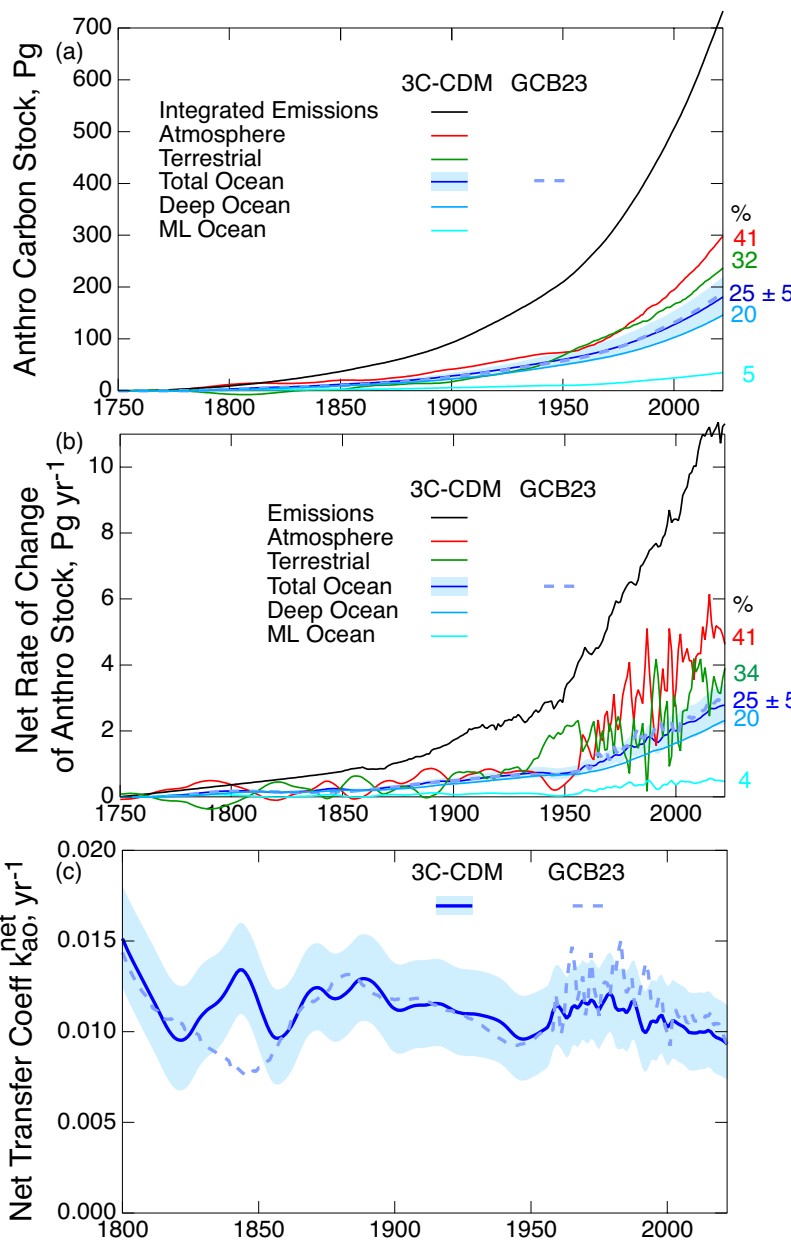

**Figure 3.** (a) Anthropogenic stocks of carbon in the several compartments of biogeosphere from the 3C-CDM. (b) Rates of change of the several stocks. (c) Net transfer coefficient between the atmosphere and the total ocean



compartment. Here and in subsequent figures the dark blue curve denotes the central value as calculated by the 3C-CDM and the shading denotes 1-$\sigma$ uncertainty in the modeled quantity based on corresponding uncertainty in piston velocity $v_\text{p}$. Numbers at right give percentage of integrated emissions (a) or annual emissions (b) in the indicated compartments for final year of the calculation, 2022. Dashed curves in all three panels show for comparison values of the corresponding quantities from the GCB23 compilation (Friedlingstein et al., 2023).

The second representation of the results is as the rates of change of the several anthropogenic stocks, **Fig. 3b**, evaluated as the time derivatives of the corresponding stocks. The rate of change of the atmospheric stock, based on measurements and models, is taken from the historical tab of the data file accompanying the GCB23 report. As noted earlier, there is a slight natural flux from the ocean to the atmosphere ~0.6 Pg yr$^{-1}$ due to riverine flux into the ocean that must be accounted for in comparisons with observations. Neglecting that, the net flux from the atmosphere into the ocean is treated as anthropogenic.

The net flux from the atmosphere to the ocean is equal to the rate of increase of the total ocean stock

$$F_{\text{ao}}^{\text{net}} = \frac{dS_\text{o}}{dt}$$

.            (5.7)

The 1-$\sigma$ uncertainty range in the rate of increase in the ocean stock is calculated as for the stock itself. For the total ocean, and also the TB (again, not modeled, but calculated by difference), the rates of change represent the net flux from the atmosphere into the respective compartment; for the DO this rate of change is equal to the net flux from the ML to the DO.

The numbers at the right represent the fraction of annual emissions taken up annually by the several compartments for the final year of the model run. Attention is called to the much greater fluctuations in the atmospheric stock subsequent to 1959 versus prior; this is a consequence of the use of measurements of $S_\text{a}$ from $CO_2$ concentrations in air, versus from ice cores prior to 1959 in which fluctuations are damped out in the data. The fluctuations are likewise enhanced in the stock in the total ocean and in the ML ocean because of rather tight coupling of the ML to the atmosphere. In contrast the stock in the

DO remains rather smooth after the transition to use of annual atmospheric data to drive the model; this is a consequence of the DO stock being (roughly) proportional to the integral of the ML stock, which damps fluctuations. The curves for the rates of change in the several stocks are roughly proportional to the stocks, the constant of proportionality being the rate of change of atmospheric stock to the stock itself.

The third means of representing the results of this model, **Fig. 3c**, takes cognizance that whereas the stocks (**Fig. 3a**) in the

several compartments and fluxes between the compartments are extensive properties of the system, the net transfer coefficients describing these fluxes (ratios of net fluxes to stocks of leaving compartments) are intensive properties of the system. Thus it is of utility to define, evaluate, and compare net transfer coefficients, as a function of time within a given model or across models. As intensive properties the net transfer coefficients are much more constant over the Anthropocene



than the extensive properties, the stocks or the fluxes, because of removal of the dependence on the secular growth of these

quantities. Here, the net flux $F_{ao}^{net}$, **Fig. 3b** is in the direction atmosphere to ocean. For the leaving compartment being the atmosphere, the net transfer coefficient from the atmosphere to the ocean is defined as

$$k_{ao}^{net} \equiv F_{ao}^{net} / S_a^{ant} ,$$ (5.8)

shown as a function of time over the Anthropocene in **Fig. 3c**. (The initial years of the model run are omitted from the figure as they are quite noisy because of the low values of the denominator quantity $S_a^{ant}(t)$ at small $t$.). As anticipated the net

transfer coefficient exhibits much less secular change than either the stocks or the fluxes, permitting more detailed examination of the time dependence. As seen below this near constancy permits examination of possible time dependence of $F_{ao}^{net}$.

The simple model described and presented here yields considerable insight. As an illustration, comparison of the net transfer coefficient for uptake of carbon by the DO $k_{md}^{net}$ obtained from the model results to the actual transport coefficient obtained

from the piston velocity $k_{md} = v_p/z_m$, **Fig. 4**, illustrates the gradual decrease of $k_{md}^{net}$ from its initial value, equal to $k_{md}$, by about 12 % over the Anthropocene, that is a consequence of return flux of anthropogenic carbon from the DO to the ML.

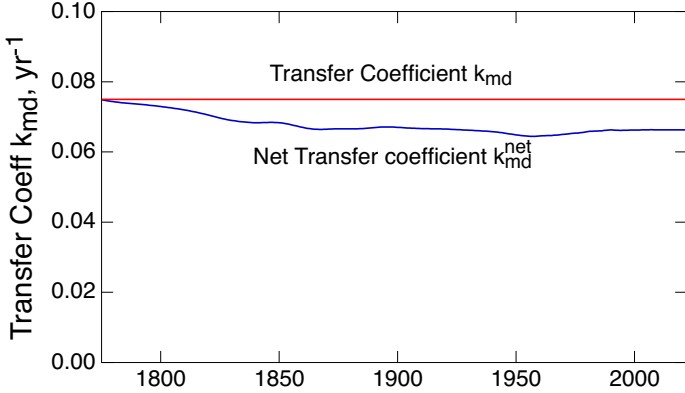

**Figure 4**. Transfer coefficient $k_{md}$ and net transfer coefficient $k_{md}^{net}$ describing transport between the ML and the DO as a function of time over the Anthropocene.

Another important example is comparison of the gross anthropogenic fluxes between the atmosphere and the ML, **Fig. 5a**, and the gross and net anthropogenic fluxes between the ML and the DO, **Fig. 5b**. As shown in **Fig. 5a**, the return flux from the ML to the atmosphere is nearly equal to the forward flux from the atmosphere to the ML (7 % difference at the end of the run). Such near equality in gross fluxes is indicative of near equilibrium between the two compartments. In contrast, the difference in the gross fluxes between the ML and the DO is about an order of magnitude, with the gross flux between those



compartments and the net flux, expressed as the fractional difference, about 11 %. Thus the rate-limiting step governing the
     overall net transfer from the atmosphere to the ocean is transfer from the ML to the DO.

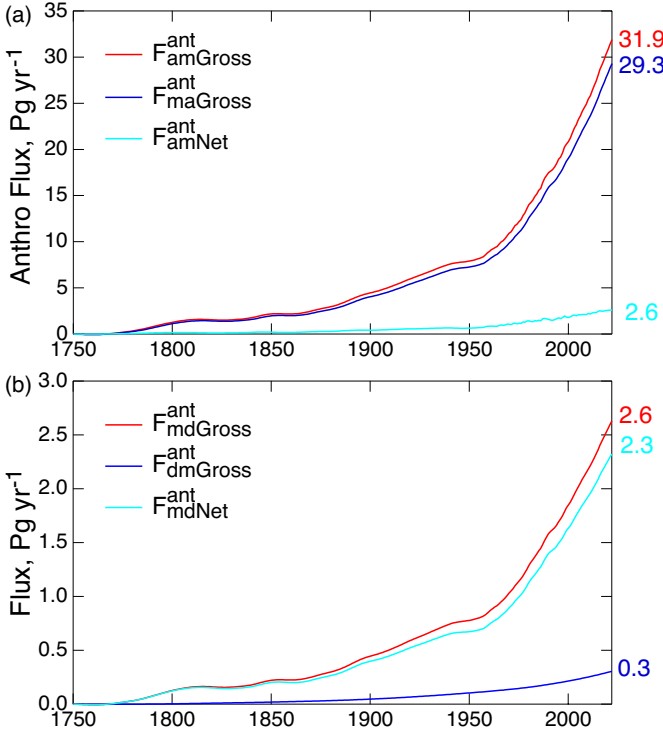

**Figure 5.** Gross and net anthropogenic fluxes, **(a)** between the atmosphere and the ML, and **(b)** between the ML and
the DO over the Anthropocene as calculated by the 3C-CDM. Numbers at right give values for year 2022.

The near equality of the forward and reverse fluxes between the atmosphere and the ML suggests that treating those two
     compartments at equilibrium would yield only slight differences in the evolution of the system. The pertinent calculation is
     readily carried out by integration of a single ODE, with the forcing driving transfer from the ML to the DO being the stock in
     the ML, $S_m$, where $S_m$ is taken as at equilibrium with the atmospheric stock, $S_a$. That is,

$$\frac{dS_d^{ant}}{dt} = k_{md}S_m^{ant} - k_{dm}S_d^{ant}.$$
(5.7)

for initial condition $S_d^{ant} = 0$, where

$$S_m^{ant}(t) = S_m(t) - S_m^{pi} \quad \text{and} \quad S_m(t) = S_m^{eq}(t) = K_{am}(S_a)S_a(t).$$
(5.8)





Treating the ML as at equilibrium with the atmosphere and forcing the model by the ML stock reduces the model to a two-compartment model, 2C-CDM. Treating these compartments at equilibrium would afford the advantage of eliminating one model parameter, the deposition velocity $k_{am}$, thereby reducing the number of parameters in the model to one, the piston velocity $v_p$. (This equilibrium assumption removes any dependence of the model results on the kinetics of transfer of $CO_2$ between the atmosphere and the ML, eliminating the need to represent the nonlinear relation between $k_{ma}$ and $k_{am}$.) Calculation of the rate and extent of net transfer into the ML and the DO under this equilibrium assumption lends strong support to the near-equilibrium hypothesis, with the differences in the stocks in the two compartments between the full kinetic treatment and the treatment under the assumption between the atmosphere and ML at the end of the model run being slight, only 7 or 8 %, **Fig. 6**.

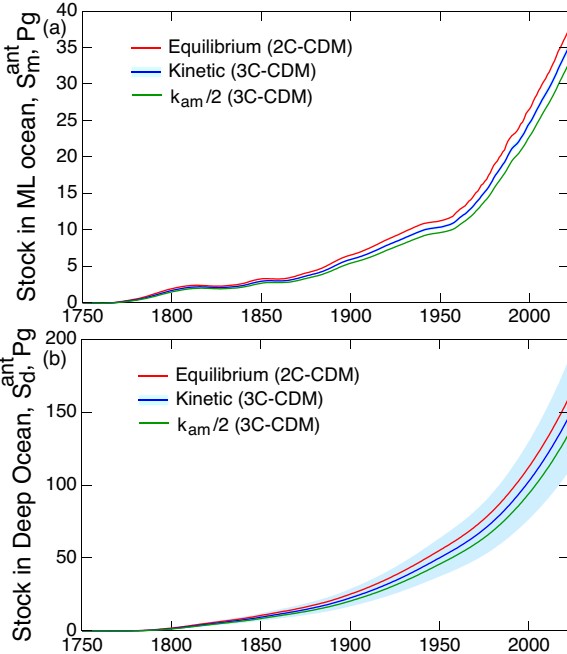

**Figure 6**. Time dependence of anthropogenic stocks, (a) in the ML and, (b) in the DO as calculated with the 3C-CDM (denoted "kinetic", transfer coefficient $k_{am} = 0.119$ yr$^{-1}$), with $k_{am}$ decreased by a factor of 2, and with a model variant 2C-CDM that treats the stocks in in the atmosphere and the ML as at equilibrium. Shading denotes 1-$\sigma$ uncertainty based on corresponding uncertainty in piston velocity $v_p$; shaded band in *a* is barely visible.

The sensitivity of uptake into the ML and the DO to the value of $k_{am}$ was further examined by reducing the value of this transfer coefficient by a factor of 2, also shown in **Fig. 6**. **Figure 6a** shows that the anthropogenic stock in the ML depends



only weakly on $k_{am}$ (-7 % at the end of the model run for $k_{am}$ divided by 2, or +7 % for the equilibrium state, equivalent to $k_{am} = \infty$); this insensitivity is a further demonstration of near equilibrium between the two compartments. The anthropogenic

ML stock is wholly insensitive to the value of the piston velocity $v_p$, with the range of that uncertainty propagated into the uncertainty in the ML stock (~25 %), yet a further consequence of the stock in the ML being in near equilibrium with the atmosphere. In contrast the extent of uptake into the DO, **Fig 6b**, is controlled essentially only by that piston velocity, which governs the rate of transfer from the ML to the DO.

Comparing the results obtained with the equilibrium model versus the full kinetic model with the propagated uncertainty due

to uncertainty in $v_p$, **Fig. 7a**, shows that the difference in total anthropogenic ocean stock, taken as the sum of the anthropogenic stocks in the ML and DO, between the equilibrium model and the kinetic model (9 % at the end of the model run) is well less than the propagated uncertainty in this quantity due to uncertainty in $v_p$, shown by the shaded regions surrounding the respective central values, about ±20 %. This finding holds also, **Fig. 7b** for the net atmosphere-ocean flux, and **Fig. 7c**, for the intensive quantity, the net transfer coefficient from the atmosphere to the total ocean $k_{ao}^{net}$ (**Eq. 5.8**). As

the difference between the quantities for the kinetic model and the equilibrium model are well within the uncertainty due to uncertainty in $v_p$, for most practical purposes the equilibrium model might be used with confidence, although for completeness here it is the results from the kinetic model that are compared to the results of other models in **Section 6**. (Shown also in the several panels of **Fig. 7** are the results from the kinetic model with the transfer coefficient $k_{am}$ diminished by a factor of 2, all well within the propagated uncertainty from uncertainty in $v_p$, further illustrating the rather

small uncertainty in the several measures of the extent and rate of uptake of $CO_2$ into the global ocean due to uncertainty in $k_{am}$.)

Also shown in **Fig. 7c** are the ratio of the anthropogenic stock in the ML to that in the atmosphere, for both the equilibrium and kinetic models; as expected this ratio is less for the kinetic models than for the equilibrium model. Importantly, the solubility curves exhibit a gradual but clear decrease over the years 1800-2022, 17 % for the equilibrium model. Because of

fluctuations in the net transfer coefficient, the decrease in solubility is more readily discernable than the decrease in the transfer coefficients; plotting the quantities proportionally on the same graph highlights the decrease in the net transfer coefficient, the pertinent intensive measure of the uptake of anthropogenic $CO_2$ rate over the industrial period. Comparison of the curves for $k_{ao}^{net}$ and $S_m^{ant}/S_a^{ant}$ suggests that $k_{ao}^{net}$ may likewise have decreased by a similar amount over the period 1800-2022, the decrease perhaps greatest and most evident subsequent to about 1960. Return flux from the DO to the ML,

**Fig. 4** would also contribute to decrease in $k_{ao}^{net}$. Plotting the net uptake rate as the intensive quantity $k_{ao}^{net}$, **Fig. 7(c)** rather than as the extensive quantity, the net flux $F_{ao}^{net}$, as has been customary, makes it possible to discern any slight decrease in the net uptake rate due to change in equilibrium solubility, the effect of which is swamped in $F_{ao}^{net}$ by the increase in net flux due to increase in the atmospheric stock.



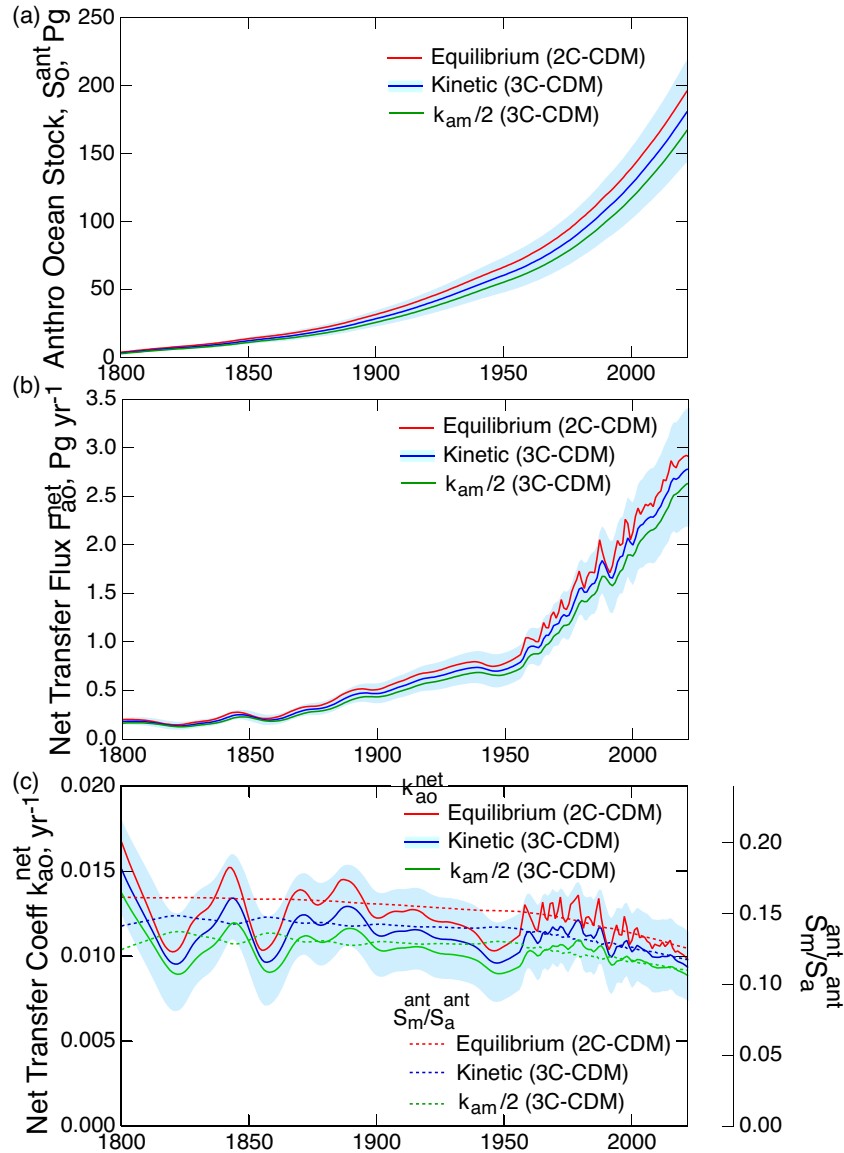

Figure 7. Time dependence of (a) anthropogenic stock in the global ocean, (b) net atmosphere-ocean flux, and (c) (left-hand axis), the net atmosphere-ocean transfer coefficient $k_{ao}^{net}$, as calculated with the 3C-CDM, which explicitly represents the kinetics of mass transport between the atmosphere and the ML, with a model variant, the 2C-CDM, which treats the stocks in those two phases as at equilibrium, and with $k_{am}$ artificially diminished by a factor of 2 in



the 3C-CDM. Dashed curves in (c), with values shown by the auxiliary right-hand axis, proportional to left-hand axis,
denote ratio of anthropogenic stock in the ML to that in the atmosphere in the three model variants.

Although in the 3C-CDM the stock growth rate of anthropogenic carbon in the TB and transfer coefficient from the atmosphere to the TB are determined by difference, the stock and growth rate in the sum of the ocean and TB compartments and corresponding transfer coefficient from the atmosphere to these two compartments are determined entirely by observation,

$$S_{ot}^{ant} \equiv S_o^{ant} + S_t^{ant} = \int Q_{ant}dt - S_a^{ant}.$$   (5.9)

and similarly for the net transfer flux

$$\frac{dS_{ot}^{ant}}{dt} = Q_{ant} - \frac{dS_a^{ant}}{dt}.$$   (5.10)

and net transfer coefficient

$$k_{a\to o+t}^{net} = \frac{Q_{ant} - \frac{dS_a^{ant}}{dt}}{S_a^{ant}}.$$   (5.11)

Here the anthropogenic emissions $Q_{ant} = Q_{ff} + Q_{lu}$ is used for the first time in this analysis. Because of uncertainty in this quantity, the value of $k_{a\to o+t}^{net}$ must be viewed with caution. Nonetheless examination of the three net transfer coefficients, $k_{a\to o+t}^{net}$, $k_{at}^{net}$, and $k_{ao}^{net}$, **Fig. 8** is informative; here $k_{at}^{net}$ and $k_{ao}^{net}$ are shown for $v_p$ = 7.5 m yr$^{-1}$ and $k_{am}$ = 0.119 yr$^{-1}$. The net transfer coefficient $k_{ao}^{net}$ as calculated with the 3C-CDM is as shown in Fig. 7(c); variation in $k_{ao}^{net}$ is dwarfed by that in $k_{a\to o+t}^{net}$ and $k_{at}^{net}$. As noted, $k_{a\to o+t}^{net}$ is independent of the model calculations; $k_{at}^{net}$ is only weakly dependent on the
model calculations through subtraction of the fairly constant (on the scale of **Fig. 8**) value of $k_{ao}^{net}$. On the time scale of the figure, 1850-2022, none of the quantities exhibits appreciable time dependence; as noted in conjunction with **Fig. 7c** there may be slight decrease over the time period that might be discerned in comparison with the ratio of the equilibrium stocks in the atmosphere and the ocean. The large fluctuations of $k_{a\to o+t}^{net}$ and $k_{at}^{net}$ in the earlier years of the record may be due in large part to fluctuations in $Q_{ant}$, amplified by small values of $S_a^{ant}$ in the denominator of **Eq. 5.11**. Raupach et al. (2014)
called attention to the appreciable decrease in $k_{a\to o+t}^{net}$ at time subsequent to 1959. The slopes of the least-squares fits to $k_{a\to o+t}^{net}$ and $k_{at}^{net}$ over this period are essentially identical to each other and several fold greater than the slope of $k_{ao}^{net}$, suggesting that any decrease in the net removal rate of excess $CO_2$ from the atmosphere over this period would be dominated by decrease in the net uptake rate into the TB. However the large fluctuations in these quantities in the earlier years of the record might raise questions regarding the confidence that can be placed in that inferred decrease.





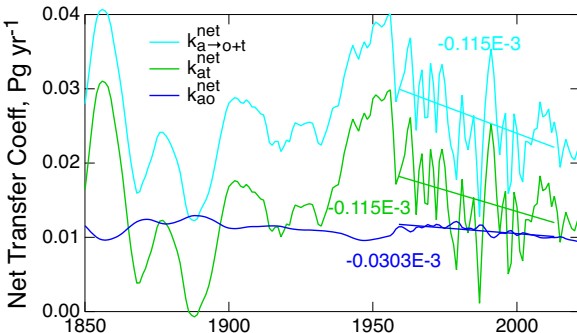


**Figure 8.** Time dependence of the net transfer coefficients from the atmosphere to the global ocean and terrestrial biosphere taken together, $k_{a \rightarrow o+t}^{net}$, as determined from observations; to the global ocean, $k_{ao}^{net}$, as determined by the 3C-CDM; and their difference, $k_{at}^{net}$. Also shown are least-squares fits to the three quantities over the time period 1959-2022 and their slopes in units (Pg yr$^{-1}$) yr$^{-1}$.

**5.1 Radiocarbon**

Examination of the stocks of radiocarbon in the two ocean compartments (and their sum) and comparison with models and observations provides additional assessment of the skill of the present (or any) model, especially as the stock of atmospheric $^{14}CO_2$ exhibited a very different time history than ordinary $CO_2$, as a consequence of the pulse in concentration due to atmospheric weapons testing (generally denoted "bomb carbon") and its rapid decline subsequent to cessation of these tests

in 1963. The concentration-driven model radiocarbon is similar to that for ordinary carbon. However, as the amount of radiocarbon that is transferred between the atmosphere and the ML ocean is so slight that it does not perturb the DIC equilibrium, the simple (DIC-dependent) equilibrium constant between the two phases is used. The rate of transfer from the atmosphere to the ML is the same as for ordinary carbon, **Eq. 5.1**; kinetic and equilibrium isotope effects are small and hence neglected. For the stocks of radiocarbon denoted $R$ and the fluxes denoted $G$, the gross flux of $^{14}CO_2$ from the

atmosphere to the ML is

$$G_{am} = k_{am} R_a$$ 

(5.12)

and the gross flux from the ML to the atmosphere is

$$G_{ma} = k_{ma} R_m$$ 

(5.13)

where


$$k_{ma} = \frac{k_{am}}{K_{am}}$$

(5.14)



and

$$K_{am} = \left( \frac{S_m}{S_a} \right)_{eq}$$

(5.15)

The corresponding differential equations are

$$\frac{dR_m^{ant}}{dt} = k_{am} R_a^{ant} - k_{am} K_{ma} R_m^{ant} - k_{md} R_m^{ant} + k_{dm} R_d^{ant}$$

$$\frac{dR_d^{ant}}{dt} = k_{md} R_m^{ant} - k_{dm} R_d^{ant}$$

, (5.16)

analogous to **Eq 5.6**, but with the differential equilibrium constant $K'_{ma}$ being replaced by the ordinary equilibrium constant $K_{ma}$, which is an order of magnitude greater (**Appendix A, Fig. A1**), and the first equation being in the stock itself rather than in the departure from equilibrium. The initial condition is that all the anthropogenic stocks are zero at year 1750. The equations are linear with constant coefficients except for the dependence of $K_{am}$ on time due to the decrease in $CO_2$ solubility over the industrial period (~45 %, **Fig. A1**). As with the calculations for $CO_2$, the anthropogenic stock of $^{14}C$ in

the terrestrial biosphere is evaluated as the difference between integrated emissions and the increases in the stocks in the atmosphere and the ocean.

The results of the model calculation are shown in **Fig. 9**; here the several quantities are presented in units of $1 \times 10^{26}$ atoms in the global atmosphere; also shown for reference are integrated emissions, which are not used in the calculations. The atmospheric stock that drives the calculation is from measurements: Stuiver et al. (1998) and the tabulations of Graven et al.

(2017), and Hua et al. (2022). Most notable is the strong increase in anthropogenic atmospheric emissions due to bomb carbon) over the years 1950-1963, and resultant increases in the stocks in the other several compartments. This was then followed by an abrupt decrease in emissions resulting from the ban on atmospheric testing of nuclear weapons that took effect in late 1963 and resultant more gradual decrease in atmospheric $^{14}CO_2$ stock.

Prior to 1950 the atmospheric stock increased gradually. This increase in stock is opposite in sign to the well known decrease

in isotopic ratio, $^{14}\Delta CO_2$, which is a different way of expressing the same measurements and which is the hallmark of the Suess effect (Suess 1955) that served as an early demonstration of the increase of atmospheric $CO_2$ due to emissions of fossil fuel $CO_2$. The increase in atmospheric stock of $^{14}CO_2$ over this period is a consequence of the increase in the stock of cold $CO_2$ from fossil fuel emissions outweighing the decrease in $^{14}\Delta CO_2$ (Schwartz et al. 2024).



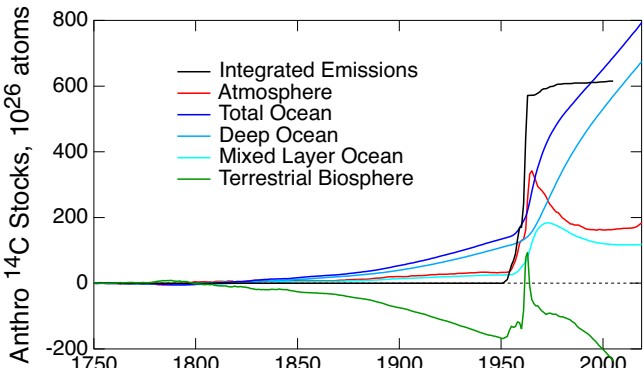

**Figure 9**. Anthropogenic stocks of radiocarbon in the several compartments of biogeosphere as calculated with the 3C-CDM. Atmospheric stock is calculated from measurements (Stuiver et al., 1998; Graven et al., 2017; Hua et al., 2022. Emissions data were provided by Tobias Naegler (personal communication, 2020), based on the Yang et al. (2000) emissions inventory.

Over the pre-bomb period the stocks the stocks of $^{14}$C in the two ocean compartments increase in response to the increase in atmospheric stock. As the anthropogenic stock in the DO is the integral of net transfer from the ML, essentially all from the ML to the DO, this stock increases continuously over the entire time period and is the major contribution to the total anthropogenic ocean stock. The sum of the increases in the three compartments substantially exceeds the anthropogenic emissions over this period, which are essentially zero. Where is this $^{14}$C coming from? It must be coming from the terrestrial biosphere, for which the change in stock, evaluated as the difference between emissions (essentially zero) and the increases in the stocks in the atmosphere and ocean, is negative throughout this period. Why is the stock in the TB decreasing? At preindustrial time, prior to significant emissions of cold $CO_2$ from fossil fuel combustion, the system was in isotopic equilibrium, that is equal and opposite fluxes of both cold and hot $CO_2$ between the atmosphere and the TB. As the atmosphere became depleted in hot $CO_2$, the net flux of hot $CO_2$ from the TB to the atmosphere exceeded the net flux from the atmosphere to the TB, and hence the decrease in hot C in the TB and the increase in hot $CO_2$ in the atmosphere. All of this is captured by the 3C-CDM

### 6. Comparison with other models and with observations

Results obtained with the 3C-CDM, specifically the anthropogenic increase in the ocean stock, the time derivative of this increase, and the associated net transfer coefficient, are compared here with the corresponding quantities from other models. These models include concentration-driven and emissions-driven models. The 3C-CDM calculates the stocks in the two ocean compartments, with a key finding being that the major fraction (~80 %) of the ocean uptake of anthropogenic carbon





over the Anthropocene is in the DO compartment with only a minor fraction (~20 %) in the ML, **Fig. 3a**, with the uptake into the total ocean being governed mainly by $k_{md}$. Most models against which the 3C-CDM is compared provide only net uptake of anthropogenic carbon from the atmosphere into the total ocean. Hence it is the sum of the anthropogenic stocks in the ML and the DO, and the net anthropogenic flux from the atmosphere into the ML (which is the sum of the rate of change

of the stocks in the two compartments) that can be compared across models here.

As an initial comparison, **Figure 10** which shows a compilation (Wang et al., 2016) of net atmosphere-ocean flux over the period 1881-2005 from 13 carbon-cycle models that participated in the CMIP5 (Coupled Model Intercomparison Project, Phase 5) of the World Climate Research Program, together with the net atmosphere-ocean flux from the 3C-CDM, as shown in **Fig 3b**. The CMIP5 models were emissions-driven, rather than concentration-driven as in the 3C-CDM. The net uptake

flux calculated with the 3C-CDM closely matches the center of gravity of the net flux of the CMIP5 models (after upward adjustment of those fluxes by 0.3 Pg yr$^{-1}$ to account for preindustrial riverine flow), all increasing from roughly 0.2 Pg yr$^{-1}$ at year 1881 to the 2005 value of about 2.3 Pg yr$^{-1}$. However the spread of the fluxes of the CMIP5 models (among the set of models and as characterized by the large fluctuations of the results of individual models) substantially exceeds the 1-$\sigma$ uncertainty associated with the 3C-CDM results throughout most of the modeled period, up until about 2005, by which time

the uncertainty of the 3C-CDM results, which scales in proportion to the net flux, becomes comparable to the intermodel spread of the CMIP5 models. This situation suggests that the atmosphere-ocean flux calculated with the 3C-CDM is performing comparably to the individual CMIP5 models, albeit without the large sub-decadal fluctuations exhibited by most of those models.

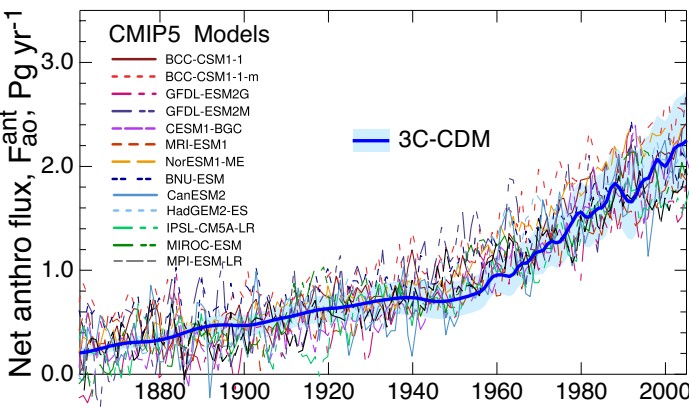

**Figure 10**. Comparison of net uptake flux, as in **Fig. 7(b)**, of anthropogenic atmospheric $CO_2$ into the global ocean as determined with the 3C-CDM with that calculated by several models participating in the CMIP5 intercomparison.



CMIP5 data are as compiled by Wang et al. (2016, their Fig. 1), augmented by 0.3 Pg yr$^{-1}$ to account for preindustrial natural flux.

The results obtained with the 3C-CDM were compared above in **Fig. 3** with several measures of the net ocean uptake of
excess atmospheric $CO_2$, as reported regularly by the GCB project by Friedlingstein and colleagues. The quantity denoted here as net atmosphere-ocean flux is denoted "ocean sink" by Friedlingstein and colleagues, although this cannot be considered a true, irreversible sink, because of return flux from the DO to the ML that would diminish net ocean uptake. This quantity, which is reported in the historical tab of the data tables that accompanied the 2023 Global Carbon Budget (GCB23, Friedlingstein et al., 2023), is a synthesis of multiple model and observational results. The central values of the several
measures of the net uptake of excess $CO_2$ by the global ocean obtained with the 3C-CDM – the anthropogenic ocean stock (**Fig. 3a**), the net flux (**Fig. 3b**), and the net transfer coefficient (**Fig. 3c**) – are essentially identical, within the uncertainty associated with the 3C-CDM results, to the results presented by the GCB23 budget.

As noted above, the intensive quantity, the net transfer coefficient, **Fig. 7c**, is capable of yielding a much more sensitive comparison than might obtained with either the anthropogenic stock, **Fig. 7a**, or the rate of increase of that stock (equivalent
to net atmosphere-ocean flux), **Fig. 7b,** both of which are extensive quantities. For that reason much of the following comparison is on the net transfer coefficient from the atmosphere to the ocean, $k_{ao}^{net}$, defined and evaluated by **Equation 5.8**.

The results of the 3C-CDM are now compared in more detail with the data presented by GCB23. For time prior to 1959, at which time there was a change in the methodology of calculation of the sink rate given by GCB23, the ocean sink rate reported by GCB23 was the average of two data sets, that of DeVries et al. (2014) and that of Khatiwala et al. (2013).
Briefly, DeVries and colleagues determined the net atmosphere-ocean transfer flux by means of a steady-state global Ocean Circulation Inverse Model (OCIM) constrained by momentum balance and the continuity equation taking into account the frictional, Coriolis, and barotropic pressure forces, and the imposed surface wind stress and baroclinic pressure forces. The model is discretized on a grid with 2˚ horizontal resolution and 24 unevenly spaced vertical levels in the ocean with thickness ranging from 30 m at the surface to 500 m at depth. The parameters of the model are constrained by global
observations of potential temperature, salinity, radiocarbon, and CFC-11; these tracers were assimilated into the model to obtain optimal estimates of the climatological mean (steady state) ocean circulation, ventilation, and air-sea gas exchange rates. The optimized circulation and air-sea gas exchange rates from this model were then used to simulate the oceanic uptake of anthropogenic $CO_2$ over the industrial era, with the amount of estimated as the difference between a time-varying run in which surface DIC concentration in equilibrium with global and annual mean atmospheric $CO_2$ was increased
according to observation versus being held constant. The DeVries time series data are available from the cited paper.




Khatiwala et al. (2013) presented a synthesis of observational and model-based estimates of the storage and transport of anthropogenic DIC in the global ocean, obtained by three approaches, 1) examining the perturbation to the carbonate equilibrium, 2) a Green's function method representing tracer transport, and 3) a transit time distribution method. Those three approaches yielded the anthropogenic inventory for year 1994 $106 \pm 17$ Pg, $108 \pm 14$ Pg, and $114 \pm 22$ Pg, respectively, with

the best estimate for year 2010 $155 \pm 31$ Pg. No time series was presented for the Khatiwala data; however it was possible to infer the time series from the GCB average of the two time series and the DeVries time series.

The three time series are shown in **Fig. 11** as net atmosphere ocean transfer coefficient $k_{ao}^{net}$. Also shown in the figure is the net transfer coefficient obtained with the 3C-CDM, together with its associated uncertainty, as in **Fig. 3c**. The comparison of the net transfer coefficient with the with the two time series, and their average, shows that the 3C-CDM results rather closely

match all three time series. From about 1860 to 1959, the values of $k_{ao}^{net}$ obtained with the 3C-CDM agree with the two data sets, and their average, well within the uncertainty range of the 3C-CDM due to 1-$\sigma$ uncertainty in $v_p$.

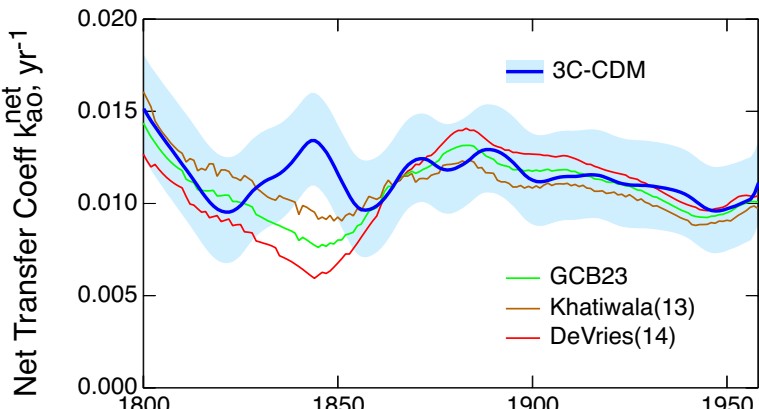

**Figure 11**. Time dependence of the net transfer coefficient from the atmosphere to the global ocean $k_{ao}^{net}$ over years 1800-1958, as given by the GCB23 project (data tables accompanying Friedlingstein et al., 2023, historical tab),

DeVries (2014) and Khatiwala et al. (2013, as inferred from the GCB average and the DeVries (2014) data set).

For time subsequent to year 1959 GCB23 presented global and annual mean net atmosphere to ocean flux as an average from two sets of data, both of which can be considered concentration-driven. The first set consisted of what GCB23 denoted as "observation-based $f CO_2$ products" for eight regression models, where $f$ denotes the fugacity of $CO_2$, for all practical purposes equivalent to the stock as used here to drive the 3C-CDM. The second set consisted of 10 global ocean

biogeochemistry models (GOBMs). The approaches taken in those studies are briefly summarized here and results of the individual model runs are presented here as global and annual mean net atmosphere-ocean transfer coefficient evaluated by



**Eq 5.8** from the net fluxes presented by GCB23 using the global and annual mean anthropogenic stock in the atmosphere as presented by GCB23. These results are then compared to the results obtained with the 3C-CDM.

For the $f$CO$_2$ study the spatial resolution of most regression models was 1˚ × 1˚ (with one 2˚ × 2.5˚); the temporal resolution
for most models was monthly. Carbon dioxide concentration for the different models was obtained from a variety of time- and space-dependent data sets regridded to the model resolution by inversion methods, such as multi-dimensional regressions, neural networks, and machine learning. Further information on the methods employed by the several models is presented in Table S3 of the Supplementary Material of GCB23. Here in calculating the global-mean net transfer coefficient the global-mean anthropogenic by **Eq 5.8** the atmosphere-ocean flux data were used as provided in the ocean sink tab of the
GCB23 data set; these data are explicitly meant to be anthropogenic fluxes (i.e., not including the riverine flux), so the data may be directly compared to the anthropogenic fluxes calculated with the 3C-CDM. The results are shown in **Fig. 12a** as time series of $k_{ao}^{net}$ along with the results of the 3C-CDM. The net transfer coefficient calculated with the 3C-CDM agrees rather closely with the results of most of the models calculating net atmosphere-ocean flux according to the $f$CO$_2$ protocol, with the spread among the models comparable to the uncertainty range of the 3C-CDM.

The 10 models participating in the GOBM study simulated both the natural and the anthropogenic CO$_2$ cycles in the ocean. The increase in ocean DIC stock was evaluated as the difference, after correction for model drift, between ocean stock calculated with historical atmospheric CO$_2$ increase and what those investigators denoted normal-year climate minus that with constant atmospheric CO$_2$ and normal-year climate forcing. Horizontal resolution of the models was sub-1˚ to 2˚ depending on the model; the number of vertical levels in the ocean in the several models ranged from 31 to 75. Atmospheric
CO$_2$ forcing was mainly as provided by the GCB, with annual or monthly resolution, and in some instances with partial pressure adjusted by local total pressure. Further information is provided in Table S2 of the Supplementary Material of GCB23. Again in calculating the global-mean net transfer coefficient the global-mean atmosphere-ocean flux data were used as provided in the ocean sink tab of the GCB23 data set. The results are shown in **Fig. 12b**, again with the results of the 3C-CDM. Again, the atmosphere-ocean flux calculated with the 3C-CDM agrees closely with the results of most of the
models, with the uncertainty range of the 3C-CDM, about ± 20 %, essentially overlapping the data reported by the several models contributing to the GOBM protocol.



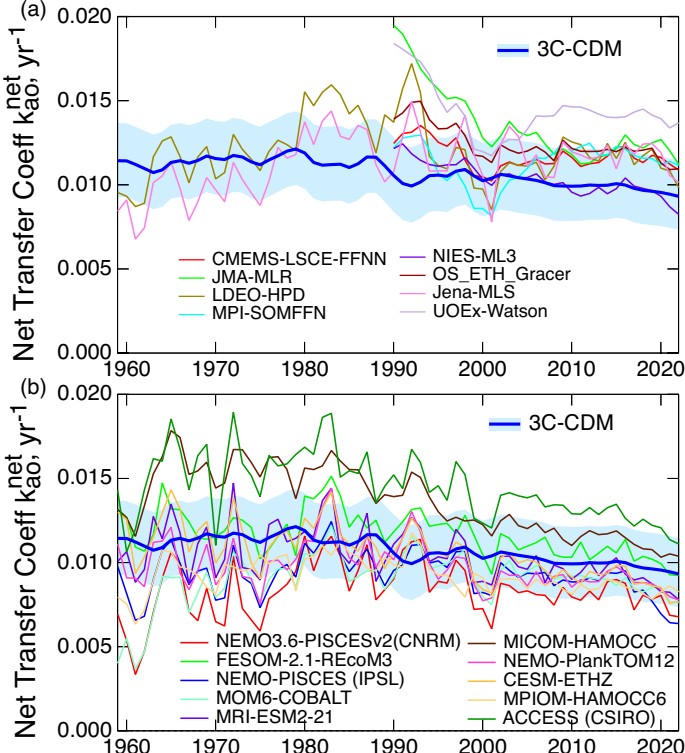

**Figure 12**. (a) Time dependence of the net transfer coefficient from the atmosphere to the global ocean as calculated from net atmosphere-ocean fluxes for eight observation-derived data sets ($f$CO$_2$) presented by the GCB23 (the data from the UOEx-Watson model, although derogated by the GBC23, are shown for completeness), together with $k_{ao}^{net}$ as calculated by the 3C-CDM. (b) as in (a), but for 10 concentration-driven global ocean biochemistry models calculated according to the GOBM protocol of the GCB23 project. For references to the $f$CO$_2$ and GOBM models see Friedlingstein et al. (2023).

A further set of modeling data against which to compare the results of the 3C-CDM is that presented by Melnikova et al. (2023), who compared net atmosphere-ocean fluxes as calculated in concentration-driven model runs in two different sets of models, Earth System Models (ESMs) and what they denote as Simple Climate Models (SCMs), the latter having roughly four ocean and four land compartments, depending on the model. In their study Melnikova et al. (2023) made use of the results from a model intercomparison project RCMIP (Reduced Complexity Model Intercomparison Project; Nicholls et al., 2020, 2021). For brief description of the models examined and references to the models see Melnikova et al. (2023). For each of the models the net transfer coefficient $k_{ao}^{net}$ was calculated by **Eq. 5.8**. using the ocean uptake rates and the





atmospheric $CO_2$ mixing ratios compiled by Melnikova et al. (2023). (I thank Irina Melnikova for making available the data developed in that study; the data are available at https://zenodo.org/records/10162686/files/SSP2.xlsx?download=1). In calculating $k_{ao}^{net}$ the values of $S_a$ for the individual models were used if presented; otherwise the values specified in the RCMIP were employed. The resulting time series are shown in **Fig. 13** for the together with the results from the 3C-CDM.

The values of $k_{ao}^{net}$ for the ESMs and several of the SCMs exhibited large fluctuations in the early years of the intercomparison runs; the fluctuations were even greater in the time period extending back to 1850, the first year for which the data are available, presumably indicative of transients associated with initializations of some components of the models. However, by the end of the time series the values of $k_{ao}^{net}$ for all the models fall within the 1-$\sigma$ uncertainty range of the 3C-CDM results, roughly $0.01 \pm 0.002$ yr$^{-1}$ at year 2022.

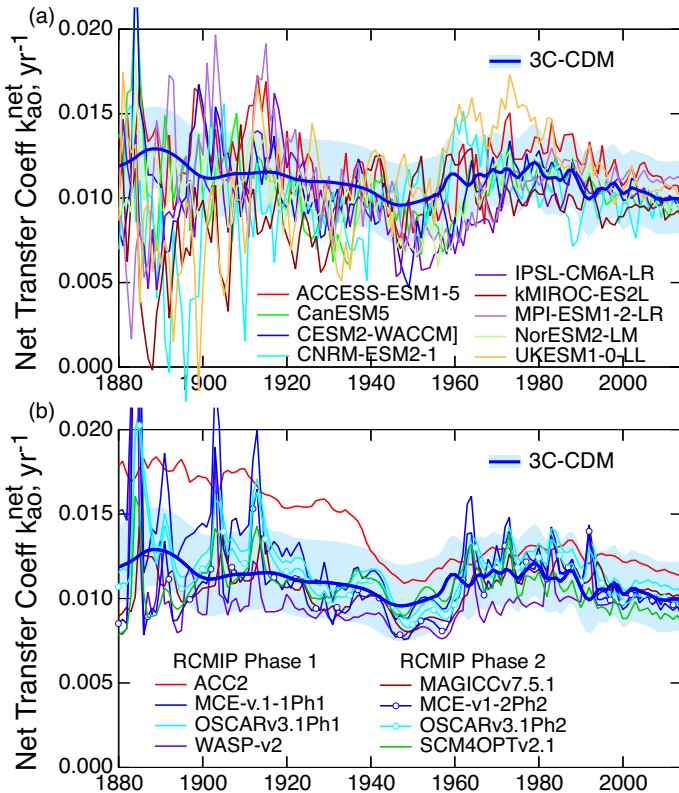


**Figure 13**. Time dependence of the net transfer coefficient from the atmosphere to the global ocean as calculated from net atmosphere-ocean fluxes in concentration-driven runs by (a) nine Earth System Models and (b) eight Simple Climate Models. Values of $k_{ao}^{net}$ were calculated from net atmosphere-ocean fluxes determined in the several studies



as compiled by Melnikova et al. (2023). Also shown are results from the 3C-CDM. Instances of the same model being
employed in the two phases of the RCMIP are distinguished by markers on the curves.

While models are plentiful, measurements are sparse, requiring sufficient samples as a function of location and depth to permit evaluation of a global integral, requiring in turn a sustained effort to conduct soundings along a sufficient number of transects to permit interpolating, and with the further requirement that the measurements be sufficiently accurate to quantify the small anthropogenic increment in DIC against the much larger total DIC concentration. A fair assessment of the status of
measurement-based assessment of global stock of ocean DIC is that of DeVries (2022), shown by the markers in **Figure 14**. Based on that standard the 3C-CDM accurately reproduces the measurements. That's probably as much as can be said about the accuracy, based on direct measurement of incremental DIC concentration, of the 3C-CDM or any model to calculate total ocean DIC. Also shown in the figure is the stock of ocean DIC calculated using the Ocean Circulation Inverse Model developed by DeVries and colleagues, which also accurately represents the measurements.

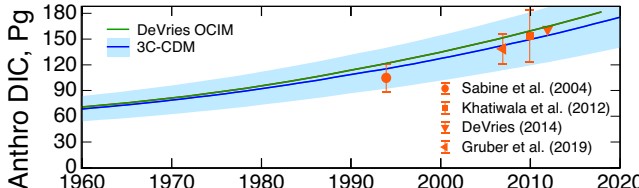


**Figure 14**. Time dependence of the anthropogenic stock of DIC in the global ocean as calculated by the Ocean Circulation Inverse Model of DeVries and colleagues together with assessed stock of anthropogenic DIC in the global ocean based on measurement campaigns. The figure is based on Figure 6a of DeVries (2022) to which is added the anthropogenic stock as calculated by the 3C-CDM with associated 1-$\sigma$ uncertainty. For citations to the measurement
data sets see DeVries (2022).

**Figure 15** compares the results from the 3C-CDM with three further observation-derived products. The most direct method (Gruber et al., 2019) determined the rate of increase in the amount of DIC in the global ocean, determined as the difference over the time period 1994-2007, divided by the time between the measurements in the earlier and later parts of the time period; the width of the band in the figure denotes that time period. The height of the band in the figure corresponds to the
1-$\sigma$ uncertainty of the Gruber et al. (2019) net uptake flux, reduced from those investigators' 2-$\sigma$ estimate for consistency with uncertainties in other data in the figure. Graven et al. (2012) scaled the rate of uptake of anthropogenic carbon to the rate of transfer of excess $^{14}CO_2$ (from nuclear weapons testing in the 1950's and 1960's) from the ML to the DO that was based on measured difference in radiocarbon amounts between the 1988–1995 and 2001–2007 time frames; this uptake, like that of anthropogenic $CO_2$, is now governed principally by transfer from the ML to the DO. The uncertainty range estimates
those investigators' estimates of the bounds on the net uptake flux.





Takahashi et al. (2009) evaluated the global net total flux of $CO_2$ into the ocean (i.e., sum of anthropogenic plus natural) based on local, time-dependent fluxes calculated as the product of the local windspeed-driven transfer coefficient times the local partial pressure difference between atmospheric $CO_2$ and $pCO_2$ of dissolved $CO_2$ in surface seawater. The $CO_2$ and $pCO_2$ data were obtained via a time- and space-interpolation method via a 2-dimensional diffusion–advection transport

equation using a data base of roughly 3 million measurements taken over the period 1970-2007 and expressed for the reference year 2000, accounting for the increase in atmospheric $CO_2$ over that time. The local, time-dependent flux was calculated using monthly-mean values of $\Delta pCO_2$ so obtained and mass-transport coefficient from reanalysis winds, typically on a 4° x 5° grid. That quantity was then integrated over time and space to obtain the global net flux; here it should be noted that that integrand comprised large positive and negative values, reflecting the local and temporal variability of the flux. As

this quantity obtained by the integration is the total net flux, it must be decremented by the natural flux, which those investigators took as 0.4 Pg yr$^{-1}$, to obtain the anthropogenic flux. The resulting global mean flux is shown by the open circle, with accompanying 1-$\sigma$ error estimate, located at year 2000, the reference year chosen by those investigators.

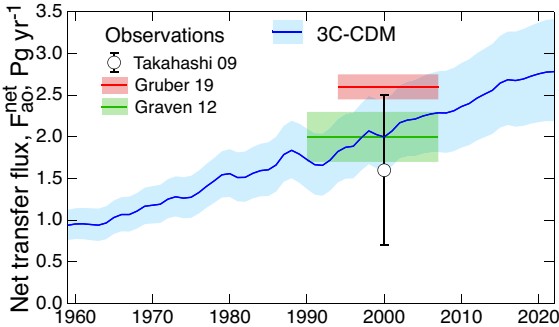

**Figure 15**. Comparison of net transfer flux of anthropogenic atmospheric $CO_2$ into the global ocean as determined by
several observation-based studies. Widths of boxes for Gruber et al. (2019) and Graven et al. (2012) denote time periods over which differences were taken; heights denote uncertainties, 1-$\sigma$ for Gruber et al., reduced from those investigators' 2-$\sigma$ estimate; "bounded by" for Graven et al. Uncertainty bar for Takahashi et al. (2009) denotes 1-$\sigma$ uncertainty. Also shown is result from the 3C-CDM with 1-$\sigma$ uncertainty range.

In sum, this section, and also **Fig. 3**, have presented comparisons of the anthropogenic enhancement to ocean carbon stock,
the net anthropogenic atmosphere-ocean flux, and the net atmosphere-ocean transport coefficient as calculated with the present concentration-driven model with results from multiple other models, and available observational data. These comparisons show consistent agreement of the results from this model with the numerous other model calculations and observational data sets examined.



There is a paucity also of ocean radiocarbon data with which to compare the present model results. Additionally, there is the
complication of baseline. The few observational studies assume zero anthropogenic ocean carbon prior to the bomb era, and
likewise most model studies examine the changes in stocks subsequent to 1950 and hence set anthropogenic stocks to zero at
that time, versus the 3C-CDM, which is initiated at preindustrial time, 1750. Hence the changes in stocks are best compared
as changes over the post-bomb era. This is achieved in **Fig 16** by offsetting the 3C-CDM results downward by about $150 \times$
$10^{26}$ atoms, the amount of increase in ocean stock in the pre-bomb industrial era (**Fig. 8**) so that all the curves are forced to
coincide at about year 1957, which was taken as the zero point in previous studies. With that offset there is fairly good
agreement between the 3C-CDM and other model results and between the models and measurement-based assessments, with
almost uncanny agreement with the hand-drawn curve of Broecker et al. (1995), which was based largely on measurements
in the GEOSECS program (black triangle). A concern in the comparison with the emissions-driven calculation of Naegler
and Levin (2006) is that that study shows substantial positive anthropogenic $^{14}$C in the terrestrial biosphere relative to its
value in 1957 at all subsequent time, decreasing only slightly after about 1980, whereas the anthropogenic TB stock
calculated here as the difference between emissions and stocks in the other compartments (**Fig. 9**) decreases rather quickly
over the post-bomb era, reaching its pre-bomb level by about year 2004. Resolution of that concern will have to await
another day.

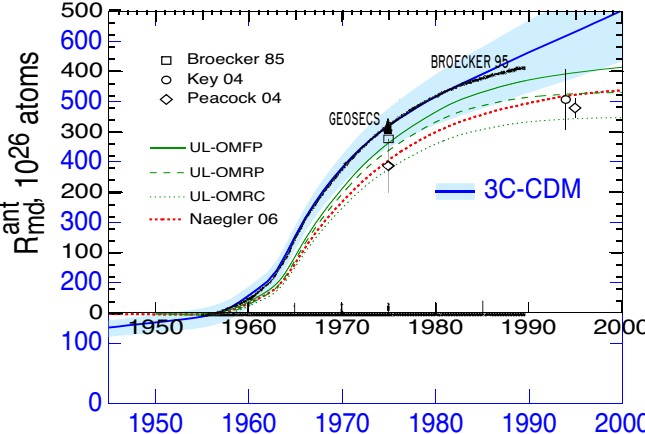

**Figure 16**. Anthropogenic dissolved inorganic radiocarbon [DI$^{14}$C] in the global ocean as inferred from observations
(Broecker et al., 1985, GEOSECS: Broecker et al., 1995, Key et al., 2004, and Peacock, 2004) and as calculated with
the 3C-CDM and by Naegler and Levin (2006) and with several model variants denoted UL from Mouchet (2013), and
hand drawn sketch from Broecker et al. (1995). Blue scale denotes 3C-CDM results offset downward by $150 \times 10^{26}$
atoms to bring values into coincidence at about year 1957.



**7. Discussion and Conclusions**

**7. 1 Discussion**

There is keen geophysical and societal interest in the disposition of emissions of anthropogenic $CO_2$ into the atmosphere, past, present and future. Although there are confident measurement-based assessments of atmospheric $CO_2$ over the Anthropocene, measurements of the fraction of emitted uptake into the ocean are sparse, and there is no confident measure

of the amount of $CO_2$ that has been (or is being) taken up by the terrestrial biosphere. Projections of future atmospheric $CO_2$ rest entirely on models. As noted wryly by Knutson and Tulyea (2005), "If we had observations of the future, we obviously would trust them more than models, but unfortunately observations of the future are not available at this time." Consequently there is acute need for assessment of the confidence that can be placed in model-based projections. The present approach to modeling of the $CO_2$ budget favors complex carbon-cycle models that treat uptake of atmospheric $CO_2$ into the ocean and

the terrestrial biosphere. These models typically represent the processes that govern uptake and transport of natural and anthropogenic carbon at horizontal scales of the order of 1˚, with the number of vertical levels in the ocean of order ten. Time steps may be as short as hours. As noted in the introduction, this approach requires dozens to hundreds of parameters that in turn depend on numerous situational variables such as temperature and wind speed, and, for uptake into the TB, on water availability and insolation, for numerous vegetation types. As a means of assessing their accuracy, these models are

run in emissions-driven mode over time historically over the Anthropocene to obtain the integrated net uptake of emitted $CO_2$ into the global ocean and the terrestrial biosphere; the difference between emissions and uptake is the anthropogenic increase of atmospheric stock, which can then be compared to atmospheric measurements. Such comparisons, together with comparison of results of multiple models serve as a measure of the confidence that can be placed in the models and their predictive capability. As an important example, the increase in the mixing ratio of atmospheric $CO_2$ from 1850 to 2014 was

found in the most recent model intercomparison project of the World Climate Research Program (Coupled Model Intercomparison Project Phase 6; CMIP6) to range from 100 to 135 ppm (Hajima et al., 2024), straddling the observed increase, 111 ppm. However such intercomparisons do little to explain the reasons for the departure from observations or the spread among models. Thus, from the perspective of many users of model output, this approach to representing the processes governing the disposition of emitted atmospheric $CO_2$ into the receiving compartments is opaque: the user of the model

results must simply take them "on faith" with only the vaguest understanding of the method by which the results were obtained, the input parameters employed and their values, and the sensitivities to those parameters, the reasons for the departure from observations and differences among the several models, and ultimately the confidence that can be placed in projections obtained with these models.





The present study takes a different tack. Here a simple model is developed that describes the rate and extent of uptake of excess $CO_2$ by the global ocean. The model has three global compartments, the atmosphere, the Mixed Layer Ocean and the Deep Ocean, with the depth of the ML taken as 100 m. As reported here the model is run in concentration-driven mode, that is using the stock of atmospheric $CO_2$, from measurements, to drive the model and with examination only of the uptake into the two ocean compartments. Although such concentration-driven model runs are only diagnostic, not prognostic, they are capable of yielding considerable insight into the controlling processes and their rates. The 3C-CDM has only two independent parameters, the deposition velocity of low- to intermediate-solubility gases from the atmosphere to the ML ocean (adjusted for the solubility and diffusive properties of $CO_2$), $k_{am}$, and the piston velocity characterizing the rate of water exchange between the ML ocean and the Deep Ocean, $v_p$. Importantly, both of these parameters may be considered intrinsic and universal properties of Earth's geophysical system, not specific to $CO_2$, and not determined by measurements in the $CO_2$ system. The model also incorporates well understood and characterized equilibria involving the several dissolved inorganic carbon species: $CO_2$, bicarbonate, and carbonate.

The keystone of the present approach to modeling the uptake of $CO_2$ by the global ocean is the use of the piston velocity $v_p$ characterizing the rate of exchange of water between the ML and the DO that is equal to the piston velocity governing transport of heat from the ML to the DO. Specifically this piston velocity is obtained, **Section 3**, from the increase over time of the heat content of the global ocean as determined by multiple measurements; likewise the uncertainty in the piston velocity, ± 30 %, derives from the uncertainty associated with the heat transport rate between the ML and the DO. The rate of uptake of DIC into the DO in the model thus rests rather firmly on the measured rate of uptake of heat by the global ocean over the time period 1960 to the present, the same process and the same time period as that governing the bulk of uptake of DIC, thus serving as an ideal surrogate for $CO_2$ transport.

In the work presented here the 3C-CDM is used to calculate the rate and extent of uptake of atmospheric $CO_2$ into the two ocean compartments, with atmospheric $CO_2$ taken as a forcing function. This application of the model yields transparent understanding of the processes controlling ocean uptake of $CO_2$, transport between the atmosphere and the ML and between the ML and the DO, and the basis in measurement that underlies the values of the two intrinsic parameters. The present study provides a direct connection between uncertainties in the two independent parameters governing the net uptake of excess atmospheric $CO_2$ by the global ocean and the uncertainty in the net uptake rate. The largest contribution to uncertainty in the rate and extent of uptake of anthropogenic $CO_2$ by the global ocean, about ± 20 %, 1-$\sigma$, at year 2022 is due to uncertainty in $v_p$. As DIC in the ML ocean is in near equilibrium with atmospheric $CO_2$, uncertainty in the transfer coefficient (deposition velocity) governing the gross rate of uptake of atmospheric $CO_2$ by the ML, $k_{am}$, makes only a minor contribution to the overall uncertainty in the rate of uptake of atmospheric $CO_2$ by the global ocean. There is some indication in the model results and based on theoretical understanding that the net transfer coefficient $k_{ao}^{net}$ characterizing



uptake of excess $CO_2$ by the global ocean may have decreased slightly (~17 %) over time because of decrease in solubility of excess $CO_2$ in the ML and slight return flux from the DO to the ML. All of this understanding may be compared to the absence of such qualitative and quantitative understanding in analysis and interpretation of results from carbon cycle models.

The rate of uptake of excess $CO_2$ by the global ocean obtained with the 3C-CDM is 2.84 ± 0.6 Pg yr$^{-1}$ at year 2022. This value is in essential agreement with the value 3.0 ± 0.3 Pg yr$^{-1}$ for 2009-2019 given in an assessment by Gruber et al. (2023)

based on multiple methods. The uptake rate determined here corresponds to a net atmosphere-ocean transfer coefficient referred to excess atmospheric $CO_2$, $k_{ao}^{net}$ 0.010 ± 0.002 yr$^{-1}$. The net transfer coefficient is found to be a more sensitive measure for comparing transfer fluxes across models and in examining the variation of this flux over time within a given model than is the net transfer flux itself. This intensive quantity exhibits only slight secular variation because of the normalization to the excess atmospheric stock, in contrast to the net flux, the time dependence of which is dominated by the

increase of atmospheric $CO_2$. As the net transfer coefficient is much more stable with time than the net transfer flux, because it does not increase with increasing atmospheric $CO_2$, it would seem useful for the net transfer coefficient to be reported in future studies.

The results of the 3C-CDM agree well with measurements of ocean uptake of $CO_2$ over the time period mid 1990's to early 2010's. The results obtained with the 3C-CDM compare fairly well with the limited available measurements of the

anthropogenic increase in ocean DIC stock (**Fig. 14**) and its rate of increase (**Fig 15**), although it must be underscored that such measurements are fraught with difficulty, relying on differences between total stocks and natural stocks, or between total stocks at two different time periods. Measurement of the rate of uptake of total $CO_2$ by the global ocean as the spatial integral of net flux, as by Takahashi et al. (2009), would seem to be particularly challenging, in part because the net flux between the atmosphere and the ocean is a small fraction (7 %, global and annual average) of the gross flux (**Fig. 5a**). The

model for radiocarbon also compares well with limited measurements of radiocarbon in the global ocean (**Fig. 16**).

The comparisons presented in **Section 6** of the results of the concentration-driven model developed here, 3C-CDM, and those obtained from a variety of complex carbon-cycle models to more simple models show surprisingly good agreement, within the roughly ± 20 % uncertainty range associated with present results, in the net atmosphere-ocean transport rate of excess $CO_2$ over the Anthropocene. Why "surprisingly"? Because of the vastly different approaches taken – the complexity

of carbon cycle models versus the simplicity of the present model. The comparisons with other models span the entire Anthropocene. Despite the multifarious approaches of current $CO_2$ models, and the wide ranges of $k_{ao}^{net}$ as calculated from results of model calculations in earlier years of the Anthropocene, the multiple studies converge to present-day value of $k_{ao}^{net}$ equal to about 0.010 ± 0.002 yr$^{-1}$ or about 1 % yr$^{-1}$. This convergence in turn lends strong support to the use of the piston velocity characterizing the rate of exchange of water between the ML ocean, as derived from measurements of the increase



in the heat content of the global ocean, as a tracer to describe the rate of uptake of excess $CO_2$ and to the value of that piston velocity determined here, $7.5 \pm 2.2$ m yr$^{-1}$.

The present model study shows that the fraction of anthropogenically emitted $CO_2$ that is taken up by the global ocean is $25 \pm 5$ %, at present or as integrated over the industrial period. The present study apportions that uptake, with the fraction of the total net ocean uptake that is transported into the DO being about 80 %. Given the long turnover time characterizing

return of water mass and any conservative tracers from the DO to the ML ocean $k_{dm}^{-1}$, or about $450 \pm 140$ years based on the piston velocity determined here, this uptake into DO may be considered irreversible, at least on time scale of the Anthropocene thus far. Ultimately, however, on the millennial time scale the DO would no longer serve as a sink for anthropogenic $CO_2$.

The 3C-CDM results, **Fig. 7c**, suggest a slight decrease in $k_{ao}^{net}$, from 0.0113 yr$^{-1}$ to 0.0093 yr$^{-1}$, over the period 1900 to the

present, for $v_p = 7.5$ m y$^{-1}$ and $k_{am} = 0.119$ yr$^{-1}$ (with similar change for other values of these parameters). Such a slight decrease, which might not otherwise be discernable, is suggested because of a similar decrease in the fractional amount of $CO_2$ dissolved in the ML over the time period; an increase in return flux from the DO to the ML over this period, **Fig. 4**, would also contribute to decrease in $k_{ao}^{net}$. From an observational perspective, the difference between emissions and growth of atmospheric stock indicates that over the time period 1959-2013 the net transfer coefficient from the atmosphere to the

combined ocean and TB compartments, $k_{a\text{-}ot}^{net} = k_{ao}^{net} + k_{at}^{net}$, determined as the difference between emissions and atmospheric growth, appears to have decreased substantially, from 0.034 yr$^{-1}$ to 0.022 yr$^{-1}$, a decrease of 0.012 yr$^{-1}$ (Raupach et al., 2014). This decrease is six-fold greater than the decrease in $k_{ao}^{net}$ calculated with the 3C-CDM over the same time period, 0.002 yr$^{-1}$, **Fig. 7c**. It remains to be understood whether the large difference is due to decrease in the terrestrial sink over this time period, error in the present model, or error in the observations. Certainly it is essential that the

reasons for the difference in the two quantities be resolved.

The model developed and examined here is essentially the same as that of Bolin and Eriksson (1959), but because the amount of atmospheric $CO_2$ has been determined in the intervening years, by measurements in ice cores and direct atmospheric measurements, it can be concentration driven rather than emissions driven. The value of piston velocity determined here from measurements of ocean heat content, $7.5 \pm 2.2$ m yr$^{-1}$, agrees remarkably well with that obtained from

the quantities assumed by those investigators: $z_m = 1/50$ of the total depth of the sea, i.e., $z_d = 3609$ m, and $\tau_d = 500$ yr, i.e., $k_{dm} = 1/\tau_d = 0.002$ yr$^{-1}$, yielding $v_p = z_d * k_{dm} = 7.22$ m yr$^{-1}$. Bolin and Eriksson also noted " *the net increase in the atmosphere is almost independant of the precise rate of exchange between the atmosphere and the sea.* " (italics in original). Thus despite all the work in the intervening years the description of the kinetics of the uptake of $CO_2$ by the global ocean remains unchanged, but because of that work it can be used with far greater confidence .



As the version of the model presented here, being concentration-driven, is diagnostic, uptake of excess $CO_2$ into the terrestrial biosphere (TB), either annually or integrated over the Anthropocene, is not directly modeled, but, by conservation of matter, is evaluated as the difference between net anthropogenic emissions and the sum of the increase in the atmospheric stock and uptake into the ocean. For both the integrated uptake and current uptake rate, the fraction of anthropogenic emissions taken up by the TB is $34 \pm 5$ %, comparable to, but with central value slightly greater than the amount taken up by

the global ocean. Representation of uptake of excess $CO_2$ by the global ocean as developed here places a confident constraint on the uptake of excess $CO_2$ into the TB that will permit confident extension of the model developed here into emissions-driven mode that must actively represent this uptake. Such extension, to be presented in subsequent papers in this series, will allow the model to be used prognostically rather than diagnostically.

       The reliance in this study on the rate of heat uptake by the global ocean underscores the value of continued sustained

measurement of heat content in the global ocean, not just as a measure of change in Earth's energy budget, important in its own right, but also as surrogate for transport of other tracers, importantly $CO_2$. Increasing numbers of measurements of global heat content are now being obtained by a flotilla of autonomous profiling buoys under the umbrella of the Argo program as summarized recently by Cheng et al. (2022). It is to be expected that this increasing data set will result in a decrease in uncertainty in the piston velocity $v_p$, and thus improved understanding and prognostic capability for the

disposition of anthropogenic $CO_2$ under prospective $CO_2$ emission profiles. It would, of course, be of enormous value to have a system of well calibrated autonomous floats, similar to the Argo system, that directly measure ocean DIC content.

       Finally, although the emphasis here has been on the budget of anthropogenic $CO_2$, knowledge of the anthropogenic increase in the ocean DIC is highly pertinent also to the issue of ocean acidification.

**7. 3 Conclusion**

A simple, transparent three-compartment, two-parameter model describing the net transport of atmospheric $CO_2$ into the global ocean is developed and exercized over the Anthropocene. The two parameters are derived from measurements of global ocean heat uptake in response to increasing atmospheric temperature since 1960 and the deposition velocity of low- to intermediate-solubility gases to ocean water. The results of this model agree closely with available observations and with results of complex carbon cycle models. These findings lead to the conclusion that this simple carbon model represents the

net uptake of anthropogenic $CO_2$ into the global ocean as well as or better than current carbon cycle models. Hence from the perspective of describing the uptake of anthropogenic $CO_2$ into the global ocean over the industrial era, this model would meet the needs of many in the research and policy communities concerned with the disposition of anthropogenic $CO_2$



emissions. Moreover, as the current model is transparent, it readily allows examination of the dependence of the results on the governing parameters.

## Appendix A. Equilibrium solubility of $CO_2$ in ocean surface water.

The equilibrium solubility of $CO_2$ in seawater directly affects the distribution of excess $CO_2$ between the atmospheric and mixed–layer compartments. As noted in the main text, in conjunction with **Fig. 1**, the equilibrium concentration of DIC in seawater exhibits a sublinear dependence on $CO_2$ mixing ratio; *i.e.*, solubility decreasing with increasing $CO_2$ mixing ratio $x_{CO2}$. For the analysis presented here the solubility of $CO_2$ was calculated with the program CO2SYS (Lewis and Wallace, 1998) for temperature 18˚C, representative of the mean for the global ocean, and for alkalinity 2349 μmol $kg_{sw}^{-1}$, selected to obtain preindustrial stock of DIC equal to 900 Pg as given in previous analyses. Several measures of the equilibrium DIC are shown in **Fig. A1** for the entire range of $x_{CO2}$ over the Anthropocene. The sublinearity of the dependence of DIC on $x_{CO2}$, over the Anthropocene is manifested in **Fig A1a** in the small relative increase of the equilibrium concentration of DIC or in the stock in the ML (depth 100 m) $S_m$, about 7 % versus the much greater relative increase in the stock in the atmosphere $S_a$, about 50 %. This has the effect of distributing incremental anthropogenic $CO_2$ increasingly into the atmosphere versus the ocean. The conventional measure of the solubility, the equilibrium constant, the ratio of the total stocks in the two compartments at equilibrium,

$$K_{am} = \left( \frac{S_m}{S_a} \right)_{eq} \tag{A1}$$

decreases over this period, by about 45 %, **Fig. A1b**. Perhaps a more relevant measure of the decrease in solubility is the decrease in what may be denoted the anthropogenic equilibrium constant,

$$K_{am}^{ant} = \frac{S_m^{ant}}{S_a^{ant}}, \tag{A2}$$

the ratio of the changes in the two stocks, **Fig. A1c**. (**Figs. A1b, c, and d)** are drawn to the same relative scale so that the relative changes in the several quantities may be readily perceived.). Finally **Fig A1d** shows what is denoted here as the differential equilibrium constant that affects the rate of change in the ML stock for a given rate of change in the atmospheric stock. This differential equilibrium constant is developed in **Appendix B**.



**Appendix B. Transfer fluxes of $CO_2$ between the atmosphere and the ocean mixed layer**

*B1. Kinetics of transfer of $CO_2$ between gas and aqueous phases*

The only $CO_2$ species in the atmosphere is gaseous $CO_2$. In solution, however, dissolved $CO_2$ (commonly denoted as the hydrate, carbonic acid, $H_2CO_3$) dissociates to an equilibrium mixture of $H_2CO_3$, bicarbonate ion $HCO_3^-$, and carbonate

1075   ion $CO_3^{2-}$, the totality of the three species being denoted dissolved inorganic carbon DIC. The equilibria are well characterized, and the kinetics of these dissociation-association reactions are sufficiently rapid that equilibrium can be assumed on the time scales of interest here. Because the only species that exchanges between solution and the gas phase is $CO_2$, the rate of this exchange is proportional to the concentration of $H_2CO_3$, not to that of DIC. This has an effect on the kinetics of the equilibration between gaseous $CO_2$ and DIC and on how the rates of exchange between the two phases are

1080   related to the stocks of $CO_2$ and DIC, the quantities of principal interest here. This phenomenon is well recognized (*e.g.*, SG06, pp. 330-331) but is nonetheless worth revisiting to develop relations between flux densities (expressed in terms of concentrations) and fluxes (expressed in terms of stocks in the two compartments).

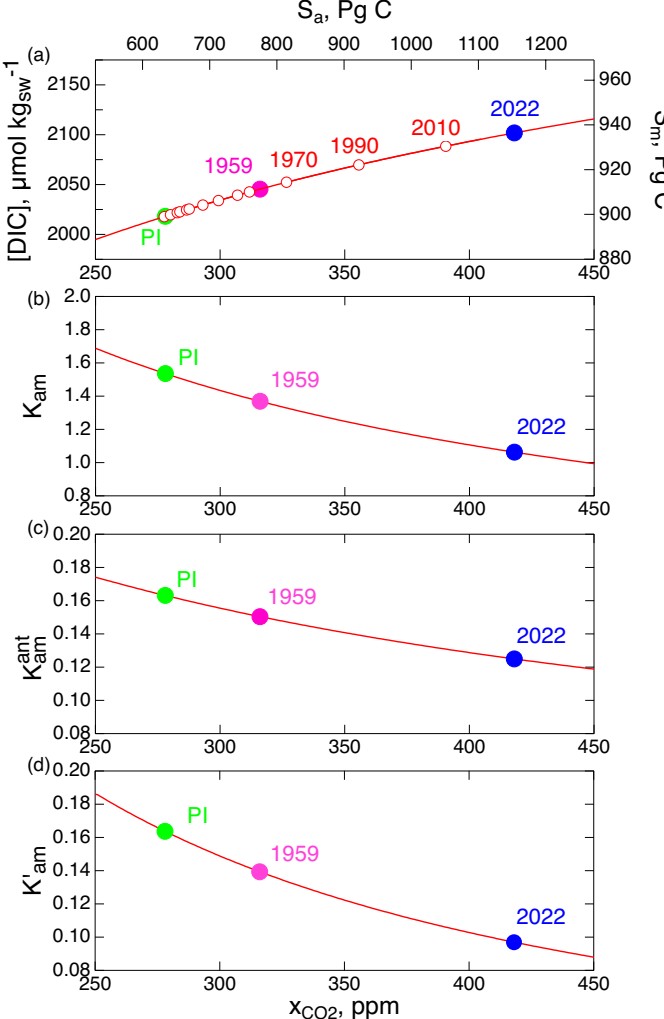

**Figure A.1**. **(a)**, Dependence of the equilibrium aqueous concentration [DIC], left axis, and mixed-layer stock $S_m$, right axis, of dissolved inorganic carbon, *i.e.*, total $CO_2$, on the atmospheric dry-air mixing ratio $x_{CO2}$ or stock $S_a$ of $CO_2$. $S_m$ is calculated for mixed layer depth $z_m = 100$ m, ocean temperature, 18 °C and total alkalinity, 2349 μmol $kg_{sw}^{-1}$. Open circles denote 20-year intervals of atmospheric $x_{CO2}$, 1750-2010; points are also shown corresponding to preindustrial PI, 1750; the beginning of contemporaneous measurements of atmospheric $CO_2$ mixing ratio, 1959; and the present, 2022. **(b)**, Equilibrium constant for the stocks in the two compartments, **Eq. A1**. **(c)**, Anthropogenic equilibrium constant, **Eq. A2**. **(d)**, Differential equilibrium constant, **Appendix B, Eq B20**.





The gross flux density from the atmosphere to the ML is proportional (by the transfer coefficient $\gamma_{am}$) to the volumetric concentration (denoted by square brackets) of gaseous $CO_2$,

$$\phi_{am} = \gamma_{am}[CO_2(g)].$$ (B1)

As noted in the text the coefficient $\gamma_{am}$ for this phase transfer process is not a constant (at a given temperature and pressure) as would be the case for a rate coefficient of a chemical reaction, but depends on situational variables, importantly the wind speed that induces convective mixing in the vicinity of the interface. For moderate- to low-solubility gases such as $CO_2$ the rate-limiting step is mass transport on the water side of the interface. The gross flux density from the ML to the atmosphere is similarly proportional to the concentration of aqueous $H_2CO_3$,

$$\phi_{ma} = \gamma_{ma}[H_2CO_3(aq)]$$ (B2)

The concentrations have dimension amount per volume; the flux densities $\phi$ have dimension amount per area and time; and the transfer coefficients $\gamma$ have dimension amount per area and time per (amount per volume) or length per time (and are thus frequently denoted a transfer "velocity"). The net flux density is the difference between the gross flux densities

$$\phi_{am,net} = \gamma_{am}[CO_2(g)] - \gamma_{ma}[H_2CO_3(aq)].$$ (B3)

At equilibrium the net flux $\phi_{am,net} = 0$, from which it is seen that the two transfer coefficients are related by the equilibrium constant for dissolution of $CO_2$, commonly denoted the Henry's law solubility constant, which is a function of temperature but is only weakly dependent on solution composition and very weakly on atmospheric pressure and $CO_2$ partial pressure.

$$H_s^{cc} = \frac{\gamma_{am}}{\gamma_{ma}} = \left( \frac{H_2CO_3(aq)}{CO_2(g)} \right)$$ (B4)

Here the notation $H_s^{cc}$ follows the convention of Sander et al. (2022) wherein the subscript s denoting solubility states that the ratio is solution/per gas phase, and where the superscripts c and c denote concentrations in both phases. As an equilibrium constant, $H_s^{cc}$ is not dependent on situational variables wind speed, turbulent intensity, and the like, in contrast to the individual transfer coefficients $\gamma_{am}$ and $\gamma_{ma}$.

Equation **B4** permits examination of the kinetics of relaxation of a perturbation from a system initially at equilibrium in order to determine the kinetics of this relaxation. For a closed system initially at equilibrium that is perturbed by addition of an incremental small amount of $CO_2$,

$$\phi_{am,net} = \gamma_{am}\left([CO_2(g)]_0 + \delta[CO_2(g)]\right) - \gamma_{ma}\left([H_2CO_3(aq)]_0 + \delta[H_2CO_3(aq)]\right)$$ (B5)



where the subscript 0 denotes the initial equilibrium state, and where $\delta$ denotes the departure from equilibrium, from which

$$\phi_{\mathrm{am,net}} = \gamma_{\mathrm{am}}\delta[\mathrm{CO}_2(\mathrm{g})] - \gamma_{\mathrm{ma}}\delta[\mathrm{H}_2\mathrm{CO}_3(\mathrm{aq})]. \tag{B6}$$

By **Eq (B4)**

1120
$$\phi_{\mathrm{am,net}} = \gamma_{\mathrm{am}}\left(\delta[\mathrm{CO}_2(\mathrm{g})] - \frac{1}{H}\delta[\mathrm{H}_2\mathrm{CO}_3(\mathrm{aq})]\right). \tag{B7}$$

It is desired to express the reverse flux density in terms of [DIC] rather than [H$_2$CO$_3$]. Following Sarmiento and Gruber (SG06, p. 330), application of the chain rule

$$\delta[\mathrm{H}_2\mathrm{CO}_3] = \frac{d[\mathrm{H}_2\mathrm{CO}_3]}{d[\mathrm{DIC}]}\delta[\mathrm{DIC}]. \tag{B8}$$

together with the definition

1125
$$\beta \equiv \frac{d[\mathrm{H}_2\mathrm{CO}_3]}{d[\mathrm{DIC}]} \tag{B9}$$

yields the expression

$$\phi_{\mathrm{am,net}} = \gamma_{\mathrm{am}}\left(\delta[\mathrm{CO}_2(\mathrm{g})] - \frac{\beta}{H}\delta[\mathrm{DIC}]\right) \tag{B10}$$

The quantity $\beta$ is an equilibrium property of the CO$_2$–DIC system that can (and can only) be evaluated numerically from knowledge of the equilibrium constants for dissociation of H$_2$CO$_3$. $\beta$ has the dimension of an equilibrium constant, the ratio

1130    of the concentrations of reagent and product, but it is a differential quantity, the ratio of the changes in concentrations resulting from a slight perturbation, rather than the ratio of the concentrations themselves, and consequently it is denoted here a differential equilibrium constant. Importantly $\beta$ is dependent on ocean alkalinity and to lesser extent, through the equilibrium constants, on salinity and temperature. For the purpose of the present analysis the alkalinity of seawater is taken as 2349 $\mu$mol $\mathrm{kg}_{\mathrm{sw}}^{-1}$ (consistent with the values of $S_{\mathrm{m}}$ and $S_{\mathrm{a}}$ given for preindustrial conditions in the several prior versions

1135    of **Fig. 1**) with salinity 35, and temperature 18˚C. The dependence of $\beta$ on CO$_2$ mixing ratio over the range of interest for the Anthropocene is shown in **Figure B1** for these conditions.



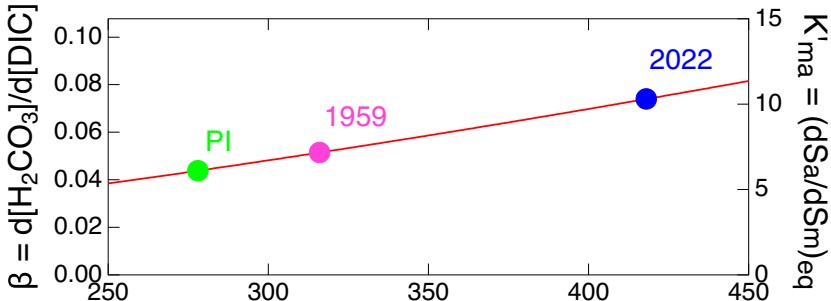

**Figure B1.** (left axis) Differential equilibrium constant $\beta = d[H_2CO_3]/d[DIC]$ denoting equilibrium change in $H_2CO_3$ concentration per change in DIC concentration as a function of $CO_2$ mixing ratio (in dry air) evaluated for seawater alkalinity 2349 µmol $kg_{sw}^{-1}$, salinity 35, and temperature 18°C, and (right axis) differential equilibrium constant $K'_{ma} = dS_a / dS_m$, the change in $S_a$ per change in $S_m$ evaluated for depth of mixed layer $z_m$ = 100 m, as developed in **Appendix B2**. Points are also shown corresponding to preindustrial PI, 1750; the beginning of contemporaneous measurements of atmospheric $CO_2$ mixing ratio, 1959; and the present, 2022, as in **Fig. A1**.

*B2. Application to transfer of CO₂ between the atmosphere and the global ocean*

For consideration of rates of transfer of $CO_2$ in the atmosphere–ML system it is desired to relate rates of change of stocks in the several compartments, expressed in terms of flux $F$ rather than flux density $\varphi$, to stocks $S$ rather than concentrations $C$. The flux between the AC and the ML and the stocks in the two compartments are related to the flux density and the two concentrations as

$$\phi = \frac{F}{A_o}; \quad S_m = A_o z_m [DIC]; \quad S_a = V_{atm}(T_{sfc}, p_{sfc})[CO_2(g)] \tag{B11}$$

where $A_o$ is the area of the global ocean, $z_m$ is the depth of the ML, and $V_{atm} = N_{air}RT_{sfc} / p_{sfc}$, with $N_{air}$ the amount (moles) of air in the global atmosphere. Substitution into **Eq B10** yields the global net flux of $CO_2$ from the atmosphere to the mixed layer for small departure from phase equilibrium, $\delta S_a$ and $\delta S_m$

$$F_{am,net} = \frac{\gamma_{am} A_o}{V_{atm}} \left( \delta S_a - \frac{V_{atm}}{A_o z_m} \frac{\beta}{H} \delta S_m \right). \tag{B12}$$

Comparison with **Eq 2.1 of the main text** that defines the transfer coefficient relating the gross flux of $CO_2$ from the atmosphere into the global ocean to the atmospheric stock,

$$k_{am} \equiv F_{am} / S_a, \tag{2.1}$$

permits the identification





$$k_{am} = \frac{\gamma_{am} A_o}{V_{atm}} \qquad (B13)$$

so that

$$F_{am,net} = k_{am} \left( \delta S_a - K'_{ma} \delta S_m \right), \qquad (B14)$$

where

$$K'_{ma} = \frac{V_{atm}}{A_o z_m H} \beta = \left( \frac{dS_a}{dS_m} \right)_{eq}. \qquad (B15)$$

More generally, with $K'_{ma}$ viewed as a function of DIC concentration in the ML, or equivalently of $S_m$,

$$F_{am,net} = k_{am} \left( S_a - K'_{ma}(S_m) S_m \right), \qquad (B16)$$

where the dependence of $K'_{ma}$ on $S_m$ is explicitly indicated. Like $\beta$, $K'_{ma}$ may be thought of as a differential equilibrium constant but pertinent not to $x_{CO2}$ and [DIC], but to the two stocks. However unlike $k_{am}$, which can be considered a geophysical constant, independent of the $CO_2$ partial pressure or of the DIC concentration, $K'_{ma}$ exhibits a dependence on DIC concentration (or equivalently on the $x_{CO2}$ in equilibrium with DIC concentration); as well, $K'_{ma}$ depends on the arbitrary choice of the depth taken for the mixed layer $z_m$. As $K'_{ma}$ is proportional to $\beta$ by geophysical constants (as well as inversely proportional to $z_m$) it can be shown on the same graph as $\beta$, by a proportional axis, **Fig. B1**, with numerical value of increasing from 5 to 10 over the Anthropocene; for $z_m = 100$ m, as employed throughout this analysis, the proportionality factor in **Eqs B12 and B15** $V_{atm}/(A_o z_m H) = 139.26$. $K'_{ma}$ can equivalently be interpreted as the ratio of the reverse transfer coefficient (from the ML to the AC) to the forward transfer coefficient (from the AC to the ML) $k'_{ma}$,

$$K'_{ma} = \frac{k'_{ma}}{k_{am}}. \qquad (B17)$$

where $k'_{ma}$ represents the transfer coefficient characterizing the flux from the ML to the AC expressed in terms of the departure of $S_m$ from its equilibrium value as

$$F_{ma} = k'_{ma} \left( S_m - S_{m,eq} \right) = K'_{ma} k_{am} \left( S_m - S_{m,eq} \right). \qquad (B18)$$

The transfer coefficient $k'_{ma}$ so defined is used in solution of the differential equations for evolution of the stocks of $CO_2$ in response to the anthropogenic perturbation as it automatically takes into account the redistribution of the DIC species associated with transfer of $CO_2$ from the ML to the AC.

The differential equilibrium constant $K'_{ma}$ is closely related to a quantity known as the buffer factor or Revelle factor (SG06, p.332)





$$\mathfrak{R} \equiv \left( \frac{d \ln p_{CO2}}{d \ln [\text{DIC}]} \right)_{\text{eq}} \tag{B19}$$

that is commonly employed to relate the change in $CO_2$ partial pressure to the change in DIC taking into account the equilibria of the DIC species.

$$K'_{\text{ma}} = \left( \frac{dS_{\text{a}}}{dS_{\text{m}}} \right)_{\text{eq}} = \left( \frac{S_{\text{a}}}{S_{\text{m}}} \right)_{\text{eq}} \left( \frac{d \ln S_{\text{a}}}{d \ln S_{\text{m}}} \right)_{\text{eq}} = \left( \frac{S_{\text{a}}}{S_{\text{m}}} \right)_{\text{eq}} \left( \frac{d \ln p_{CO2}}{d \ln [\text{DIC}]} \right)_{\text{eq}} = \left( \frac{S_{\text{a}}}{S_{\text{m}}} \right)_{\text{eq}} \mathfrak{R} . \tag{B20a}$$

Expressing $F_{\text{ma}}$ by **Eq B18** thus automatically satisfies this Revelle relation.

The inverse of $K'_{\text{ma}}$ ,

$$K'_{\text{am}} = \left( K'_{\text{ma}} \right)^{-1} = \left( \frac{dS_{\text{m}}}{dS_{\text{a}}} \right)_{\text{eq}} . \tag{B20b}$$

shown in **Fig. A1d**, is about an order of magnitude less than $K_{\text{am}}$ . It is $K'_{\text{am}}$ that appears in in the differential equation for the evolution of the stock of carbon in the ML, **Eq 5.6**, whereas in the differential equation for the evolution of the radiocarbon stock **Eq 5.13** it is $K_{\text{am}}$ .

**Competing interests**

The author declares no competing interests.

**Acknowledgements**

This work was initiated while the author was associated with Brookhaven National Laboratory, supported in part by the US Department of Energy under Contract No. DE-SC0012704; views expressed here do not necessarily represent the views of BNL or DOE. This article and corresponding preprints are distributed under the Creative Commons Attribution 4.0 License. I thank Ernie Lewis and Yin Nan Lee for valuable suggestions.

**Supplementary Material**

This paper is accompanied by an Excel workfile consisting of 3 sheets containing the time series data from this study.



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
