# Peer review of "Three-Compartment, Two-Parameter, Concentration-Driven Model for Uptake of Excess Atmospheric CO2 by the Global Ocean"

_EGUsphere, 2024_

## Referee Comment (RC3)

Three-Compartment, Two Parameter, Concentration-Driven Model for Uptake of Excess Atmospheric CO2 by the Global Ocean

By Stephen Schwartz

As the title suggests, this paper describes a diagnostic, three-compartment, two-parameter, concentration-driven global model of the ocean $CO_2$ sink. The paper is well written and contributes to understanding of the processes governing the ocean's uptake of anthropogenic $CO_2$ since the beginning of the industrial age. It should be published after addressing a few small issues listed below.

The primary advantages of the model describe here are its relative simplicity and associated mechanistic transparency. This model divides the Earth system into atmospheric, mixed-layer ocean (ML) and deep ocean (DO) reservoirs. A third critical carbon reservoir, the terrestrial biosphere (TB) is not modeled explicitly, but is estimated from the difference between the anthropogenic $CO_2$ emissions and the anthropogenic contributions to the other reservoirs. The exchange of carbon between these global reservoirs is determined by diffusion coefficients, directly traceable to available, time-dependent observations of the atmospheric $CO_2$ concentration and ocean heat transport.

The exchange of carbon from the atmosphere to ML is determined by the transfer coefficient, $k_{am}$, and that between the ML and atmosphere is given by $k_{ma}$, which are governed by turbulent mixing and (to a lesser extent) concentration gradients across the interface. The transfer of carbon between the ML and DO reservoirs, $k_{md}$, is parameterized by a piston velocity, $v_p$, such that $k_{md} = v_p / z_m$, where $z_m$ is the depth of the mixed layer. Because measurements of stocks and fluxes of carbon between the ML and DO are not adequate, the piston velocity is derived from the rate of heat uptake by the global ocean over the past 50 years.

With these definitions, the exchange of anthropogenic carbon between the atmosphere, ML and DO is described by a pair of differential equations (Equations 5.6). The deposition of anthropogenic carbon from the atmosphere into ML and DO is then described by solving these equations, with appropriate initial conditions, for the period extending from 1750 to the present. Changes in carbon stocks predicted by this model are then validated against global totals derived from global ocean biogeochemistry models (GOBMs) by the Global Carbon Project (c.f., Fig 3).

As noted above, a key asset of this simple model is its ability to identify the relationship between key transport processes and their changes over time. For example, the results presented in Figure 4 indicate that while the gross carbon transport coefficient between the ML and DO is assumed to be constant over the industrial era, the net transport coefficient has actually decreased, slightly, largely due presumably to the turnover of the DO and associated return flux to the ML during this period. However, as acknowledged here, the model cannot explicitly identify the root cause of this change.

More importantly, this simple model clearly shows that the rate of uptake of anthropogenic $CO_2$ by the global ocean is governed largely by the rate of exchange of carbon between the ML and DO, parameterized by the piston velocity, $v_p$, in this model. In contrast, the ML stock is insensitive to the value of $v_p$, such that the transfer coefficient between the atmosphere and ML can be approximated by assuming that the atmosphere and ML are in equilibrium without introducing large errors (lines 580-582). This result is not a surprise to those most familiar with existing measurements and models of the

ocean carbon sink, but it is not generally recognized across the larger carbon cycle community, and is not as obvious from results of GOBMs or Earth System Models.

This result has two critical implications for existing carbon cycle models. First, as noted in Section 7, because the efficiency of the ocean $CO_2$ sink depends strongly the exchange of anthropogenic carbon between the ML and DO, uncertainties in this transport (parameterized as $v_p$ in this model) drive uncertainties in the ocean sink. More importantly, and NOT mentioned here, any CHANGE in ocean dynamics associated with climate change that alters the net ML to DO transport of carbon could have significant consequences for the efficiency of the ocean sink. The author is strongly encouraged to reinforce this point.

Minor points:

Line 12: "This piston velocity is determined from the measured the rate of uptake of heat …'

➔ This piston velocity is determined from the measured rate of uptake of heat …

Line 140: "… the net land-use–change (LUC) emission, represents the net annual carbon flux from the TB into the atmosphere from net deforestation, i.e., the flux from deforestation minus that from afforestation."

Land use change also affects soil carbon, grasslands, etc. Is that ignored here, or just rolled into this term?

Line 145: This paragraph suggests that the only significant changes in the TB are LUC from deforestation/afforestation.  These are the primary processes, but other processes should be acknowledged. For example, the TB also exchanges carbon with the ocean through river runoff. This is mentioned in the next paragraph and discussed in sections 4.2 and 5, and 6.  However, at this point, the reader does not know whether this process is ultimately ignored (as acknowledged in section 4.2, line 422, section 6, line 785) or incorporated into the ocean or TB reservoirs.  In addition, on longer time scales, atmospheric $CO_2$ is lost through weathering and sedimentation. These terms are all small on annual to centennial time scales, but should be acknowledged.

Line 153: "The PI ML and the DO are in steady state."

Do you mean "In in the PI, ML and DO are assumed to be in steady state."

Line 180: "…the TB is not actively modeled but is evaluated, mainly for reference, as the residual between time-dependent integrated anthropogenic emissions and the anthropogenic stocks in the other three compartments."

The author might note that this assumption is partially justified by the fact that over the industrial era, stock changes associated the TB sink roughly equal those from land use change, such that the these two processes largely cancel out, leaving the ocean as the only net sink of anthropogenic carbon during this period.

Lines 689 – 700 '… Where is this [14]C coming from? It must be coming from the terrestrial biosphere, …

That is at least one source. Another possibility is that it is coming from land use change (LUC) that disturbs soil that was exposed to [14]C during the atmospheric test period.

Line 904-905: " …there is no confident measure of the amount of $CO_2$ that has been (or is being) taken up by the terrestrial biosphere"

This statement is slightly dated. The amount of $CO_2$ that is being taken up by the terrestrial biosphere is now being monitored at increasing spatial resolution by a growing fleet of space-based sensors. While the $CO_2$ fluxes estimated from these measurements still have significant uncertainties (~0.3 to 1 PgC/yr when integrated over the globe), there is growing confidence in their results.

Line 902: The focus on prognostic $CO_2$ models in the opening paragraph of the Discussion seemed a little strange, given that the model described here is a diagnostic model that "cannot be used prognostically" as acknowledged on line 25.  Later in this (long) paragraph, it states "As a means of assessing their accuracy, these models are run in emissions-driven mode over time historically over the Anthropocene to obtain the integrated net uptake of emitted CO2 into the global ocean and the terrestrial biosphere; the difference between emissions and uptake is the anthropogenic increase of atmospheric stock, which can then be compared to atmospheric measurements. Such comparisons, together with comparison of results of multiple models serve as a measure of the confidence that can be placed in the models and their predictive capability." Only then can we make this connection. You might consider reorganizing this paragraph to introduced the value of the diagnostic approach before documenting the challenges of Earth System Models.

Lines 954-959: "The largest contribution …"

These two sentences state what is perhaps the most profound scientific conclusion presented in this paper. While not entirely new, these points are often obscured by the complexity of GOBMs and Earth System Models. You might consider moving these points the beginning or end of a paragraph. You might also consider rewording this conclusion to emphasize its implications beyond the details of the model described here. For example,

"The largest contribution to uncertainty in the rate and extent of uptake of anthropogenic CO2 by the global ocean is due to uncertainties in the transport between the ML and DO, simulated here as $v_p$, which is uncertain by ~20% 1-sigma.  As DIC in the ML ocean is in near equilibrium with atmospheric $CO_2$, uncertainty in the transfer coefficient (deposition velocity) governing the gross rate of uptake of atmospheric $CO_2$  by the ML, (simulated here by $k_{am}$), makes only a minor contribution to the overall uncertainty in the rate of uptake of atmospheric $CO_2$ by the global ocean."

Then start a new paragraph with "There is some indication in the model results …"

Finally, throughout the paper, a few dozen commas are needed to improve readability. Whenever a sentence starts with a preposition, (line 40: "About 250 years ago humankind …" → "About 250 years ago, humankind …"), it should be followed by a comma. This is currently done about half the tiem. Variables introduced as appositives should also be offset by commas (line 217, "The rate of volume exchange $F_V$ may" → "The rate of volume exchange, $F_V$, may …" (line 218, "denoted piston velocity $v_p$." → "denoted piston velocity, $v_p$."

---

## Author Response (AR1)

**Point-by-point identification of changes to the manuscript**

Most changes have been previously noted in the Discussion Replies to Reviewers and Community Commenter, all concatenated below. Some further changes are noted here

5      The community commentator had suggested to avoid the use of the term piston velocity to characterize the rate of water exchange between the ML and the DO. On reflection, I choose not to replace this term with another. None of the reviewers expressed any issue with the term. No change.

Line 249 of Revision: Added correspondence between values of mass transfer from ML to Deep Ocean in Sv and piston velocity m s-1.

10     Added references to review article by Crisp et al. (2021) and study by McKinley et al 2020. Added curve from McKinley to Fig 12a. Added para describing McKinley study and results, line 810??

Added further justification of 100 m as depth of ML (lines 159-167) with reference to comment reply. Is referencing a comment reply permissible?

Other minor changes throughout (clarifications, grammatical) shown in track changes.

**Reviews, Community Comments, and Author Responses.**

Reviews and Community Comments are in black

Responses publshed in Discussion are in blue

20     Changes and adds to those responses are in red

**Review1**

I read the manuscript (MS) by Stephen E. Schwartz with great interest. The manuscript details a carbon cycle model that aims to quantify the partitioning of anthropogenic $CO_2$ emissions between the terrestrial biosphere and the ocean. Unlike other models, the one proposed here

25     does not rely on existing parameterizations but rather strives for a data-first approach. For example, the transfer velocity of $CO_2$ from the atmosphere to the ocean uses heat transfer data as

a proxy, rather than approximations of difficult-to-obtain $CO_2$ concentration measurements. Being more of a paleo person, I find this approach elegant and intuitive. I also applaud the author for demonstrating that simple box models are fully capable of resolving many aspects of planetary carbon chemistry.

I thank the Reviewer for what I consider a highly positive review, and appreciate his/her recognition of the value of what he/she refers to as a "data -first" approach. I greatly appreciate also his/her favorable comment regarding the demonstration in this manuscript that "simple box models are fully capable of resolving many aspects of planetary carbon chemistry." That indeed was a major intent of this study.

I correct a statement in para 1 of the Review. The Reviewer refers to the use of heat transfer as a proxy for the transfer velocity from the atmosphere to the ocean. Actually the heat transfer proxy is for transfer from the mixed-layer (ML) ocean to the deep ocean. Transfer from the atmosphere to the ML makes use of a deposition velocity (that term is commonly used in the atmospheric chemistry community; the quantity is frequently referred to as piston velocity in the chemical oceanography community) that is more-or-less universal for low- to medium-solubility gases, after accounting for Henry's law solubility and water-side diffusion coefficient. As it turns out, the uptake rate is very insensitive to this deposition velocity, as examined in the manuscript.

The model is described carefully and in exhaustive detail, which brings me to the main points I struggled with in the manuscript. With the exception of the Discussion and Conclusions sections, the language used throughout the manuscript is extremely technical, full of acronyms, and rather difficult to read. While a detailed model description is both welcome and necessary, the manuscript in its current form is unlikely to appeal to a broader audience. I suggest splitting the manuscript into two parts: a pure model description submitted to *Geoscientific Model Development* and a shortened, less technical submission focusing on model results, perhaps in *Biogeosciences*. This strategy would likely increase the appeal and reach of the manuscript.

In para 2 the Reviewer states that "model is described carefully and in exhaustive detail." As emphasized in the manuscript, that was the intent, at least the careful part, not the exhaustive part. I would say better "described carefully and in *complete* detail" The Reviewer goes on to suggest that the manuscript be split into two papers, a pure model description and results. I would respectfully decline the suggestion. I find too often when I read a paper with modeling results, that it refers me to an earlier paper, which in turn refers me to yet an earlier paper. Here it is all together, soup to nuts; assumptions to model to results. I think that this complete presentation is a strength of the paper, not a shortcoming. I want the reader of the results paper to be aware of everything that has gone into the calculations, not to take the author's word for what he or she did in the companion paper, nor to have to have both papers open at the same time, going back and forth between them.

Further, I understand that the manuscript provides an updated estimate for $CO_2$ uptake by the oceans, but I remain unclear on how uncertainties in the underlying data (e.g., heat transfer, carbon stocks, and emission data) affect the results. This may already be addressed in the

manuscript, but a straightforward discussion of these uncertainties, presented in plain language in the discussion section, would be very welcome.

In para 3 the Reviewer suggests that it is "unclear on how uncertainties in the underlying data (e.g., heat transfer, carbon stocks, and emission data) affect the results." Actually much of the manuscript is directed to examination of the effects of these uncertainties or, where possible, to their elimination. First, I emphasize that the model is "concentration-driven". That completely removes the effects of uncertainties of emissions (and for that matter uncertainties in uptake by the terrestrial biosphere) that would be inevitable and dominant in "emissions-driven" models, which are much more favored by the community. Use of the concentration driven approach readily allows examination of the dependence of rate (or better net transfer coefficient) on excess $CO_2$ (above preindustrial). That examination in turn shows a possible slight dependence of the net uptake rate (expressed as net transfer coefficient) in Fig 7c, that would be wholly indiscernible in examination of anthro stock (Fig. 7a) or net uptake rate itself (Fig 7b). The dependences of net uptake rate and net transfer coefficient on uncertainty in the transfer coefficient between the ML and the deep ocean are thoroughly examined in the manuscript (light blue band in multiple figures). I consider the ability to readily examine the effect of this uncertainty to be a great strength of the approach. The effect of propagated uncertainty in rate of heat uptake from the ML to the deep ocean is explicitly addressed in the discussion at lines 954-955 and is given for year 2022 as ± 20 %, 1-σ.

Lastly, I wonder whether changes in carbonate saturation depth can truly be ignored (see, e.g., Boudreau et al. 2010). While these changes may be small on the timescale considered here, they are likely to affect ocean pH in the coming decades.

Boudreau, Bernard P., Jack J. Middelburg, Andreas F. Hofmann, and Filip J. R. Meysman. 2010. "Ongoing Transients in Carbonate Compensation." *Global Biogeochemical Cycles* 24. doi:10.1029/2009gb003654.

I note the Reviewer's comment about the carbonate saturation depth; however this would seem of little relevance to the present study. The referenced paper (Beaudreau et al., 2010) focuses on the time scale of the next 2000 years, whereas the present study examines changes over the Anthropocene 1750-2022.

Specific Comments

- Figure 2: Are the confidence intervals in this figure the CI for the regression, or for the prediction? What confidence level is depicted (1, 2, or 3σ?) Also, since these are linear regressions, maybe add the regression stats ($r^2$, p-value)?

  - Regarding the linear regressions in Figure 2, as stated in the caption "Confidence intervals (CI) and fitting coefficients, 68 % CI, are shown with the fits." I would think that these are sufficient measures of the uncertainties associated with the fits.

- For clarity the caption is changed to read "Confidence band (68 % CI) and fitting coefficients, 68 % CI, are shown with the fits."

- 339: differentiate between the dissolved concentration and atmospheric by using $pCO_2$ and $[CO_2]_{aq}$ ?

Regarding terminology in air-sea transfer of $CO_2$, the language of the manuscript is: "the global-annual-mean gas exchange velocity for water-side mass transport of $CO_2$, expressed in terms of the concentration of $CO_2$ (not DIC) on the water side of the interface is about 17 cm $hr^{-1}$." In the development of the transfer coefficient expressed in terms of the atmospheric stock, Eq 3.13, it is necessary to convert from the water-side mass transfer coefficient to the gas-side mass transfer coefficient. This done in terms of concentrations (not gas-side partial pressure) via the concentration-concentration Henry's law solubility coefficient (Eq. 3.11), where the concentration on the gas-side is evaluated from the molar mixing ratio of $CO_2$ in the atmosphere $x_{CO2}$ and the molar concentration of dry air. This approach obviates the need to use the partial-pressure quantity and terminology commonly used in reporting local fluxes. The square bracket notation advocated by the Reviewer for water-side concentration is used in Eqs 3.18 - 3.20.

In conclusion I thank the Reviewer for the positive review. I hope that the responses suffice to allay the Reviewer's concerns over aspects of the presentation.

**Review 2**.

Unfortunately, only after reading the manuscript did I realize that I am not qualified to fully review the paper for its technical merits. There are too many assumptions that need review by someone with more intuition for the terms and their constraints than I possess at this time. As such, please disregard my formal recommendations above, which are just a best guess that is required by the submission system.

A few general thoughts follow:

1. The writing is clear but long-winded and occasionally repetitive. It should be edited for brevity.

2. The presented model seemed perhaps miss-marketed. This paper seemed to be presenting an update and slight modification to a >50 year old model. While the model is indeed elegant and does indeed have surprising fidelity to some of the newer reconstructions as tuned, it nevertheless seems inferior to alternatives in its treatment of several processes due to its simplicity.

I appreciate the Reviewer's characterization of the model developed in the present study as "indeed elegant," and his/her observation that the model exhibits "surprising fidelity to some of the newer reconstructions as tuned." (It is not clear whether in using the phrase "as tuned" the Reviewer is referring what he/she characterizes as "newer reconstructions" or to the present model. Here I would simply state that *there is no tuning in the present model*.)

The Reviewer expresses concern that the present manuscript presents "an update and slight modification to a >50 year old model," that is, a model with a few global compartments, here the atmosphere, the mixed-layer (ML) ocean and the deep ocean (DO). The strong justification for revisiting such a modeling framework at the present time is the availability of time-dependent measurements of global ocean heat content and its rate of increase, permitting determination of the net heat transfer coefficient from the ML ocean to the deep ocean, and its uncertainty. This heat transfer coefficient serves as a proxy for the transfer coefficient of excess (anthropogenic) dissolved inorganic carbon (DIC) from the ML to the DO, the rate limiting step of uptake of excess atmospheric $CO_2$ by the DO, one of two long term sinks for this excess $CO_2$ (the other being the terrestrial biosphere), permitting confident calculation of this net uptake rate and its uncertainty. The present model also takes advantage of improved understanding of mass transfer rates between the atmosphere and of $CO_2$-DIC chemistry, although this improved understanding is of only secondary importance to the net drawdown of excess atmospheric $CO_2$ because of near-equilibrium between the atmosphere and the ML, which makes transfer between the ML and the DO the rate controlling step in this uptake.

The reviewer goes on to suggest that the present model "nevertheless seems inferior to alternatives in its treatment of several processes due to its simplicity."

While simplicity is arguably a virtue and this model does have fewer terms than most, my sense is that it would be comparably easy to implement the OCIM as to get this model up and running on my computer, or perhaps easier, and I would trust the results from the OCIM considerably more because it allows for plausible 3D circulation. The comparable ease of coding is because it is basically the same amount of work to code up the fundamental air-sea exchange equations for one grid cell as for 3 as for several hundred thousand, and the OCIM has mathematical structures that make the exchanges between the grid cells easy for computers to handle.

The Reviewer continues "While simplicity is arguably a virtue and this model does have fewer terms than most, my sense is that it would be comparably easy to implement the OCIM [Ocean Circulation Inverse Model]." The Reviewer then advocates use of the OCIM in lieu of the present three-compartment model, stating "I would trust the results from the OCIM considerably more because it allows for plausible 3D circulation." Here I would observe first that the use of the ocean uptake rate of excess heat from the atmosphere to the ocean automatically accounts for ocean circulations. The open-source version of OCIM, the AWESOME OCIM (John et al., 2020), uses a model grid with 2° latitude by 2° longitude boxes, and 24 vertical levels, i.e., order ~ $10^5$ grid cells. Application of such a model requires a matrix of transfer coefficients, which albeit sparse, involving coupling of a given grid cell only with its neighbors, also consists nonetheless of order ~ $10^5$ elements. Such a model for the present application, examination of uptake of excess $CO_2$ by the global ocean would be massive overkill. Moreover use of such a model would run totally counter to the philosophy underlying the present study, namely that of all parameters (and their uncertainties) being observationally based. Here the transfer coefficient from the ML to the DO is obtained from the measured rate of uptake of heat by the global ocean; the inverse transfer coefficient is obtained by the requirement of zero net transfer at steady state. In any event, as shown in Fig. 14 of the present manuscript, the central value of the time-dependent

anthropogenic stock of DIC in the global ocean obtained with the present model, with its two independent parameters, is virtually identical to the stock calculated by the Ocean Circulation Inverse Model (DeVries, 2022) and to the assessed stock of anthropogenic DIC in the global ocean based on measurement campaigns. Just because a more complex model *can* be used, that does not mean that a more complex model *should* be used.

For this reason, I believe that this model should be marketed as a teaching/communications tool rather than a scientific tool. If the author follows this suggestion, then I would double down on my first suggestion, as the audiences that would benefit most from this model will appreciate a more concise paper and having many of the technical details surrounding the fitting of parameters being moved to a supplement.

Finally the Reviewer suggests that the present model "should be marketed as a teaching/communications tool rather than a scientific tool." To this I take great exception. The model developed in this study, a three-compartment, two-parameter, concentration-driven model for the extent, net rate, and net transfer coefficient characterizing uptake of excess atmospheric $CO_2$ by the global ocean over the Anthropocene, should be viewed as an important, novel contribution to the research literature, in great part *because* of its simplicity, and in great part also because the parameters are derived entirely from observations without tuning. As well, the model presents ranges of uncertainty in the several quantities that derive from uncertainty in observations, mainly net rate of heat uptake by the global ocean. Yes, the model is simple, and yes it provides insight, but those features of the model should be viewed as a strengths of the model and of a research paper that presents the development the model and the results of its application, not as grounds for "marketing" the model merely as "a teaching/communications tool rather than a scientific tool."

While I am not well versed in the literature surrounding simplified climate models, my belief is that there are competitors in the simple model space that should be discussed and compared. Many are designed with Paleo applications in mind, but it would be interesting to see how they perform when used for the Holocene. If the author asserts that this paper has already done this and no other simplified models exist that could work for this purpose, then I would take them at their word.

With regard to the Reviewer's supposition that a simple model such as this may have been used in other applications, such application would not be surprising, given the historical importance of such simple models. The novelty here rests on the transfer coefficients being derived entirely from observations, mainly the rate of uptake of excess heat by the global ocean.

Apologies to all for not being able to provide a more rigorous review at this time.

In conclusion, I hope that the recapitulation here of the motivation and approach of this study, and the responses presented here to the concerns raised by the Reviewer will serve to convince the Reviewer of the value of the model developed and applied in this study. I would hope as well that the Reviewer will come to appreciate the importance of the findings obtained in this study, and

also the confidence that can be placed in them, toward quantifying the ocean sink of excess atmospheric $CO_2$ and toward improved understanding of the controlling processes.

**References**

DeVries, T.: The Ocean Carbon Cycle, Annu. Rev. Env. Resour., 47, 317–341, https://www.annualreviews.org/content/journals/10.1146/annurev-environ-120920 111307, 2022.

John, S.G., Liang, H., Weber, T., DeVries, T., Primeau, F., Moore, K., Holzer, M., Mahowald, N., Gardner, W., Mishonov, A. and Richardson, M.J. AWESOME OCIM: A simple, flexible, and powerful tool for modeling elemental cycling in the oceans. Chem. Geol., 533, 119403. https://www.sciencedirect.com/science/article/pii/S0009254119305327, 2020.

**Review 3.**

The paper is well wriiten and contributes to understanding of the processes governing the ocean's uptake of anthropogenic CO2 since the beginning of the industrial age. It should be published after addressing a few small issues listed in the attached review.

As the title suggests, this paper describes a diagnostic, three-compartment, two-parameter, concentration-driven global model of the ocean CO2 sink. The paper is well written and contributes to understanding of the processes governing the ocean's uptake of anthropogenic CO2 since the beginning of the industrial age. It should be published after addressing a few small issues listed below.

The primary advantages of the model describe here are its relative simplicity and associated mechanistic transparency. This model divides the Earth system into atmospheric, mixed-layer ocean (ML) and deep ocean (DO) reservoirs. A third critical carbon reservoir, the terrestrial biosphere (TB) is not modeled explicitly, but is estimated from the difference between the anthropogenic CO2 emissions and the anthropogenic contributions to the other reservoirs. The exchange of carbon between these global reservoirs is determined by diffusion coefficients, directly traceable to available, time-dependent observations of the atmospheric CO2 concentration and ocean heat transport.

The exchange of carbon from the atmosphere to ML is determined by the transfer coefficient, kam, and that between the ML and atmosphere is given by kma, which are governed by turbulent mixing and (to a lesser extent) concentration gradients across the interface. The transfer of carbon between the ML and DO reservoirs, kmd, is parameterized by a piston velocity, vp, such that kmd = vp / zm, where zm is the depth of the mixed layer. Because measurements of stocks and fluxes of carbon between the ML and DO are not adequate, the piston velocity is derived from the rate of heat uptake by the global ocean over the past 50 years.

With these definitions, the exchange of anthropogenic carbon between the atmosphere, ML and DO is described by a pair of differential equations (Equations 5.6). The deposition of anthropogenic carbon from the atmosphere into ML and DO is then described by solving these equations, with appropriate initial conditions, for the period extending from 1750 to the present.

I appreciate the review and the overall assessment. I expect to be able to accommodate most of the Reviewer's suggestions in the revision.

255    In his review Dr. Crisp accurately summarizes the advantages of this study. I am gratified by his statement, "The primary advantages of the model described here are its relative simplicity and associated mechanistic transparency." That statement indeed captures the intent of the study. The review rather accurately states the approach of the study: "The exchange of carbon between these global reservoirs is determined by diffusion coefficients, directly traceable to available, time-
260    dependent observations of the atmospheric $CO_2$ concentration and ocean heat transport." I think Dr. Crisp meant to use the term "transfer coefficients," which are the quantities I adduced and used in the paper, rather than "diffusion coefficients." Dr. Crisp continues, "Because measurements of stocks and fluxes of carbon between the ML and DO are not adequate, the piston velocity is derived from the rate of heat uptake by the global ocean over the past 50 years."
265    Here I would demur slightly; the approach of the study was to use measured geophysical quantities that are independent of the carbon system, of which the rate of heat uptake by the global ocean is key, rather than to calculate stocks and fluxes based on measurements in the carbon system, adequate or not.

Changes in carbon stocks predicted by this model are then validated against global totals derived
270    from global ocean biogeochemistry models (GOBMs) by the Global Carbon Project (c.f., Fig 3).

From a philosophical perspective I would take some exception to the Reviewer's statement "Changes in carbon stocks predicted by this model are then validated against global totals derived from global ocean biogeochemistry models (GOBMs) by the Global Carbon Project. First, the present model is not "predictive; it is diagnostic;" thus better "changes in carbon stocks
275    calculated with this model . . ." Second, in general, I would say, models are not "validated, except, perhaps, for their use in regulatory contexts; they are *evaluated* , by comparison with observations. Third, the results of the present model are not evaluated (or "validated") in this study by comparison with the calculated time series obtained with other, more complex, models; they are simply compared. The key comparison with GOBMs in the paper is Figure 13b. Given the
280    inter-model spread in the time-dependent net atmosphere-ocean transfer coefficients obtained from the several GOBMs, I would certainly not use the results of those models as a standard for validating, or evaluating, the results of the present model (or any model). Rather I would stay with the neutral language of the paper (line 800) ". . . the atmosphere-ocean flux calculated with the 3C-CDM agrees closely with the results of most of the models, with the uncertainty range of the
285    3C-CDM, about ± 20 % 1 SD, essentially overlapping the data reported by the several models contributing to the GOBM protocol." To this I would add, as shown in Fig. 14, that the model results compare well with limited available measurements.

As noted above, a key asset of this simple model is its ability to identify the relationship between key transport processes and their changes over time. For example, the results presented in Figure
290    4 indicate that while the gross carbon transport coefficient between the ML and DO is assumed to be constant over the industrial era, the net transport coefficient has actually decreased, slightly, largely due presumably to the turnover of the DO and associated return flux to the ML during this

period. However, as acknowledged here, the model cannot explicitly identify the root cause of this change.

The Reviewer calls attention (positively) to the indication from the present model results (he cites Fig. 4 for the net transfer coefficient from the MD to the DO, but perhaps more saliently in Fig. 7b for the net transfer coefficient from the atmosphere to the ocean) of decrease of about 17% in transfer coefficient over the Anthropocene. This is an important finding coming out of this study, noted in the Abstract and the Discussion. The ability to tease out such a small change relies on examination of the intensive quantity, the net transfer coefficient; such a small change would not be evident in examination of the extensive quantity net transfer rate and even more so in examination of time-dependent extent of uptake. Certainly it would seem essential to plot results of future studies as net transfer coefficient in order to discern such slight changes. As the Reviewer accurately notes, I was loath to ascribe a single reason for such decrease; the model indicates decrease in equilibrium solubility of $CO_2$ (as DIC) and also increased return flux from the DO to the ML as contributing factors.

More importantly, this simple model clearly shows that the rate of uptake of anthropogenic CO2 by the global ocean is governed largely by the rate of exchange of carbon between the ML and DO, parameterized by the piston velocity, vp, in this model. In contrast, the ML stock is insensitive to the value of vp, such that the transfer coefficient between the atmosphere and ML can be approximated by assuming that the atmosphere and ML are in equilibrium without introducing large errors (lines 580- 582). This result is not a surprise to those most familiar with existing measurements and models of the ocean carbon sink, but it is not generally recognized across the larger carbon cycle community, and is not as obvious from results of GOBMs or Earth System Models.

This result has two critical implications for existing carbon cycle models. First, as noted in Section 7, because the efficiency of the ocean CO2 sink depends strongly the exchange of anthropogenic carbon between the ML and DO, uncertainties in this transport (parameterized as vp in this model) drive uncertainties in the ocean sink. More importantly, and NOT mentioned here, any CHANGE in ocean dynamics associated with climate change that alters the net ML to DO transport of carbon could have significant consequences for the efficiency of the ocean sink. The author is strongly encouraged to reinforce this point.

Finally, the Reviewer notes that ". . . any CHANGE in ocean dynamics associated with climate change that alters the net ML to DO transport of carbon could have significant consequences for the efficiency of the ocean sink," and goes on to state, "The author is strongly encouraged to reinforce this point." While I concur with the statement of the Reviewer, and accept his suggestion that I call attention to this, I would have to add to that, that such a change would rest upon a change in the heat transport coefficient that drives the net uptake of $CO_2$. Much more important than the implications of such a change on $CO_2$ uptake would be the implications of such change in heat transport in Earth's climate system itself, to which the effect on net transport of $CO_2$ from the ML to the DO would be a mere codicil.

A para is added to this effect, and more broadly to the use of the parameters determined here in models of evolution of CO2 in the future is added, lines 1064-1072 of revision

Minor points:

Line 12: "This piston velocity is determined from the measured the rate of uptake of heat …'

-> This piston velocity is determined from the measured rate of uptake of heat …

Typo corrected

Line 140: "… the net land-use–change (LUC) emission, represents the net annual carbon flux from the TB into the atmosphere from net deforestation, i.e., the flux from deforestation minus that from afforestation."

Land use change also affects soil carbon, grasslands, etc. Is that ignored here, or just rolled into this term?

Line 145: This paragraph suggests that the only significant changes in the TB are LUC from deforestation/afforestation. These are the primary processes, but other processes should be acknowledged. For example, the TB also exchanges carbon with the ocean through river runoff. This is mentioned in the next paragraph and discussed in sections 4.2 and 5, and 6. However, at this point, the reader does not know whether this process is ultimately ignored (as acknowledged in section 4.2, line 422, section 6, line 785) or incorporated into the ocean or TB reservoirs. In addition, on longer time scales, atmospheric CO2 is lost through weathering and sedimentation. These terms are all small on annual to centennial time scales, but should be acknowledged.

Line 140, 145. All rolled into net deforestation/afforestation; will be clarified. The minor roles of weathering and sedimentation on the centennial time scale will likewise be noted.

Phrase is added line 145 of the revision: "and other anthropogenic changes affecting carbon in the terrestrial biosphere includng grasslands, soil carbon and the like."

These processes are acknowledged as follows beginning line 149 of the revision: "Other pertinent processes, minor on the centennial time scale examined here, include exchanges carbon with the ocean through river runoff and loss of atmospheric CO2 through weathering and sedimentation. For a recent review see Crisp et al. (2022).These terms of the carbon budget, all of which are small on annual to centennial time scales, are neglected here. "

Line 153: "The PI ML and the DO are in steady state."

Do you mean "In the PI, ML and DO are assumed to be in steady state."

Line 153. Will be clarified.

Language (line 169 of the revision) is revised and elaborated slightly to read

The ML and the DO are assumed to be in steady state (not equilibrium) in the PI. The departure from equilibrium . . .

Line 180: "…the TB is not actively modeled but is evaluated, mainly for reference, as the residual between time-dependent integrated anthropogenic emissions and the anthropogenic stocks in the other three compartments."

The author might note that this assumption is partially justified by the fact that over the industrial era, stock changes associated the TB sink roughly equal those from land use change, such that the these two processes largely cancel out, leaving the ocean as the only net sink of anthropogenic carbon during this period.

Line 180. Does not seem to require justification. The treatment does not rest on the near equality between loss of carbon from the TB by LUC and uptake by the TB due to enhanced atmospheric $CO_2$ is a happenstance that does not affect the reasoning. No change.

Lines 689 – 700 '… Where is this 14C coming from? It must be coming from the terrestrial biosphere, … That is at least one source. Another possibility is that it is coming from land use change (LUC) that disturbs soil that was exposed to 14C during the atmospheric test period.

Line 689-670.The subject of the para is the increase, during pre-bomb Anthropocene, of $^{14}$C stocks in atmosphere, ML, and DO and the complementary decrease in stock in the TB, required for the closed system examined here. Prior to emissions of $^{14}$C-free $CO_2$ during the pre-bomb Anthropocene, the atmosphere and the TB were in near isotopic equilibrium, i.e., having the same $\Delta^{14}$C. Introduction of $^{14}$C-free $CO_2$ into the atmosphere associated with fossil fuel combustion led to a decrease of $\Delta^{14}$C of atmospheric $CO_2$, perturbing the dynamic isotopic equilibrium between the two compartments. The tendency toward restoration of the dynamic isotopic equilibrium between the two compartments in response to this perturbation would require adjustment of the stocks of $^{14}$C (and $^{12}$C) in the two compartments in the direction of achieving the same $\Delta^{14}$C in each compartment; this would occur mainly by net transfer of $^{14}$C from the TB to the atmosphere, with the resultant decrease in $^{14}$C stock in the TB prior to 1950 shown in Fig. 9. Also, net transfer of $^{14}$C from the TB to the atmosphere as a component of LUC emissions (total C emissions 0 - 1.2 Pg yr$^{-1}$ during this period) would be dwarfed by net transfer of $^{14}$C associated with total C exchange between the TB and atmosphere (GPP), increasing from roughly 120 Pg yr$^{-1}$ to perhaps 180 Pg yr$^{-1}$ over this period.

Line 904-905: " …there is no confident measure of the amount of CO2 that has been (or is being) taken up by the terrestrial biosphere"

This statement is slightly dated. The amount of CO2 that is being taken up by the terrestrial biosphere is now being monitored at increasing spatial resolution by a growing fleet of space-based sensors. While the CO2 fluxes estimated from these measurements still have significant uncertainties (~0.3 to 1 PgC/yr when integrated over the globe), there is growing confidence in their results.

Lines 904-905. I suggest that the Reviewer may be a bit optimistic in the accuracy of satellite-based measurements of net carbon uptake by the TB. As an example I cite a recent survey of above-ground carbon biomass (El Masri and Xiao, 2025) as calculated from five satellite-based global data sets, which yielded a 25% standard error in the global quantity, with much greater variation when examined as a function of latitude. I would welcome the opportunity to chat offline with the Reviewer regarding his statement on recent space-based estimates of net atmosphere to TB carbon flux. Pending that I leave the statement unchanged.

Line 902: The focus on prognostic CO2 models in the opening paragraph of the Discussion seemed a little strange, given that the model described here is a diagnostic model that "cannot be used prognostically" as acknowledged on line 25. Later in this (long) paragraph, it states "As a means of assessing their accuracy, these models are run in emissions-driven mode over time historically over the Anthropocene to obtain the integrated net uptake of emitted CO2 into the global ocean and the terrestrial biosphere; the difference between emissions and uptake is the anthropogenic increase of atmospheric stock, which can then be compared to atmospheric measurements. Such comparisons, together with comparison of results of multiple models serve as a measure of the confidence that can be placed in the models and their predictive capability." Only then can we make this connection. You might consider reorganizing this paragraph to introduced the value of the diagnostic approach before documenting the challenges of Earth System Models.

Line 902. The purpose of the discussion section starting here is to set the scene for the contrast between the conventional approach using emissions-driven process models with large numbers of compartments and processes and parameters and the approach taken in this study of a simple model with three compartments and two independent, observationally constrained parameters. The reviewer suggests reorganizing the paragraph. I think the contrast is better as it is. No change.

Lines 954-959: "The largest contribution …"

These two sentences state what is perhaps the most profound scientific conclusion presented in this paper. While not entirely new, these points are often obscured by the complexity of GOBMs and Earth System Models. You might consider moving these points the beginning or end of a paragraph. You might also consider rewording this conclusion to emphasize its implications beyond the details of the model described here. For example,

"The largest contribution to uncertainty in the rate and extent of uptake of anthropogenic CO2 by the global ocean is due to uncertainties in the transport between the ML and DO, simulated here as vp, which is uncertain by ~20% 1-sigma. As DIC in the ML ocean is in near equilibrium with atmospheric CO2, uncertainty in the transfer coefficient (deposition velocity) governing the gross rate of uptake of atmospheric CO2 by the ML, (simulated here by kam), makes only a minor contribution to the overall uncertainty in the rate of uptake of atmospheric CO2 by the global ocean."

Then start a new paragraph with "There is some indication in the model results …"

440 Line 954-959. I appreciate the Reviewer's characterization of the two sentences as "perhaps the most profound scientific conclusion presented in this paper," and his desire to improve the impact of the paragraph. On re-reading the paragraph I think the impact comes through loud and clear. No change.

Finally, throughout the paper, a few dozen commas are needed to improve readability. Whenever a sentence starts with a preposition, (line 40: "About 250 years ago humankind …" -> "About 250 years ago, humankind …"), it should be followed by a comma. This is currently done about half the time. Variables introduced as appositives should also be offset by commas (line 217, "The rate of volume exchange FV may" -> "The rate of volume exchange, FV, may …" (line 218, "denoted piston velocity vp." -> "denoted piston velocity, vp."

Final remark about readability. I will take the opportunity in revising the paper to look for where commas might improve the readability.

Again, I thank the Reviewer for his attention to the manuscript, its objectives, its findings, its conclusions.

**Reference (to statement in this response to Reviewer, not for the paper)**

El Masri, B., & Xiao, J. (2025). Comparison of global aboveground biomass estimates from satellite observations and dynamic global vegetation models. J. Geophys. Res.: Biogeosci, 130, e2024JG008305. https://doi.org/10.1029/2024JG008305

**Community Comment 1 Peter Köhler**

Since I am interested in simple models myself I found this approach with a few equations at least interesting and I had a closer look at this draft, which led me to make the following comments:

I thank Dr. Köhler for his Comment, and appreciate his interest in simple models such as that in the submitted manuscript. I respond to his comments seriatim; for convenience I paste in his comments.

1. Schwartz calculates in section 5.1 changes in 14C in the terrestrial biosphere as the residual of the 14C anthropogenic emissions (bomb 14C) and the changes in the three reservoirs atmosphere, mixed layer ocean and deep ocean. I believe this is not correct, since to my understanding the calculations do not consider how the 14C-free anthropogenic (fossil fuel-based) CO2 is entering the ocean. This can in my view only be considered properly, if the model is driven by CO2 emissions (and the related 14C content of the emissions which are different for fossil fuel fluxes or land use changes), and not if the model is driven by CO2 concentrations. I therefore believe the part in the draft on 14C is buggy and should be deleted. You find an example of an emission driven setup including 14C in an older paper of mine (Köhler, 2016, doi:10.1088/1748-9326/11/12/124016).

*Response*. The Commenter states that to his understanding "the calculations do not consider how the $^{14}C$-free anthropogenic (fossil fuel-based) $CO_2$ is entering the ocean." Calculation of uptake of

ordinary $CO_2$ by the global ocean by a concentration-driven model is the principal objective of the study; if treatment of this uptake is flawed, that would be fatal to the manuscript. If, on the other hand, the Commenter accepts the treatment of ordinary $CO_2$, then it would seem he must accept treatment of $^{14}CO_2$, which is also concentration-driven and which is the same as for ordinary $CO_2$, but for a very different time profile of atmospheric $CO_2$, and with the slight exception that the dissolution of $^{14}CO_2$ does not affect the $CO_2$-bicarbonate-carbonate equilibria, making the treatment even simpler than for ordinary $CO_2$. Driving the models by observed atmospheric $CO_2$ or $^{14}CO_2$ stocks eliminates reliance on emissions as well as the fact that these emissions are different for ordinary $CO_2$ and $^{14}CO_2$.

Evaluation of changes of $^{14}C$ in the terrestrial biosphere (TB), to which the Commenter takes particular exception, is based on the conservation requirement that changes in $^{14}C$ summed over the atmosphere, ocean, and TB compartments must equal anthropogenic emissions. Hence the change in $^{14}C$ in the TB can be confidently evaluated as the difference between emissions and the sum of changes of $^{14}C$ in the atmosphere (measured) and the global ocean (modeled here) shown in Fig. 9 of the manuscript.

Regarding the Commenter's statement that "how $^{14}C$-free anthropogenic (fossil fuel-based) $CO_2$ is entering the ocean can only be considered properly, if the model is driven by $CO_2$ emissions . . . and not if the model is driven by $CO_2$ concentrations," I would respond that the requirement of models is that they represent the processes that are taking place in the real world, here the processes governing net uptake of $CO_2$ by the ocean. Hence representation of these processes should (must) be the same in an emissions-driven model as in a concentration-driven model. The motivation and great advantage of a concentration-driven model is that such a model allows examination of the treatment of uptake of $CO_2$ from the atmosphere to the ocean in isolation from other processes. The stock of atmospheric $CO_2$, which is needed to calculate uptake by the ocean, is confidently obtained from observations, as opposed to being calculated, so that there is no need for modeling net uptake of $CO_2$ by the terrestrial biosphere and no need for knowledge of anthropogenic emissions. I would hope that the Commenter would recognize and concur in the value of this approach.

2. Ocean heat content of the upper 300m of the ocean is in von Schuckmann et al. (2023, Figure 8 in doi: 10.5194/essd-15-1675-2023) less than 100 ZJ in the years 1971-2020 from a total of 381 ZJ (about 1/4 in top 300m). The top 100m should gain about a third of that number (< 33 ZJ or < 10% of the total heat content change), while here Schwartz calculates (Figure 2) from 1960-2023 a rise by ~150 ZJ in the top 100m (from >450 ZJ in the total ocean, ->about 1/3 in top 100m). So this heat content exercise seems to be at odd with the data.

*Response*. The Commenter suggests that the heat content of the upper 100 m should be 1/3 of that of the upper 300 m presented by von Schuckmann et al. (2023), i.e., 1/3 of 75 ZJ or 25 ZJ (The Commenter says 1/3 of 100 ZJ, or 33 ZJ), rather than the 169 ZJ shown in Fig. 2b of the manuscript and used in the calculations. This statement can be rebutted on several grounds.

515  First, the present analysis rests on the more recent study of heat uptake by the global ocean of Cheng et al. (2024a), of which von Schuckmann is a coauthor, rather than on the paper (von Schuckmann et al., 2023) cited by the Commenter. Von Schuckmann et al. (2023) analyzed data only through year 2020, whereas Cheng et al. (2024a) extended the analysis through year 2023 obtaining total global ocean heat content (OHC) 464 ZJ, well in excess of the 339 ZJ through year 2020, and for the upper 300 m, about 183 ZJ. The increases in total OHC in 2021, 2022, and 2023

520  were 18, 19, and 18 ZJ, respectively (their Fig. 1a); these subsequent increases account for much of the difference between OHC reported by Cheng et al. (2024a) versus von Schuckmann et al. (2023).

The question also arises what fraction of the secular heat increase of global OHC over the Anthropocene has been taken up in the top 100 m. Both Cheng et al. (2024a) and von

525  Schuckmann et al. (2023) present global OHC by depth intervals, with the top interval, i.e., the interval nearest the surface being 0 to 300 m, for which Cheng et al. (2024a) give the incremental heat content as about 183 ZJ. The Commenter suggests that the heat uptake of the top 100 m should be apportioned equally between the depth intervals 0-100 m, 100-200 m, and 200-300 m, i.e., about a third of the increase of the top 300 m going into the top 100 m, which would only be

530  about 61 ZJ. However, as developed below, such uniform apportioning would seem very unlikely.

Cheng et al. (2024b, of which von Schuckmann is also a coauthor) give the increase in global ocean surface temperature over the period 1960-2023 as 0.8 K. Under the assumption that the incremental temperature of the top 100 m is the same as that at the surface the heat of the top 100 m as the product of Earth surface area 5.10 E14 $m^2$, ocean area fraction 0.29, and volumetric

535  heat capacity of seawater 4.11 E06 J $m^{-3}$ $K^{-1}$, yielding the increase in heat content of the top 100 m over this period is 119 ZJ, or about 65 % of the increment in the top 300 m given by Cheng et al. (2024a) and much more comparable to the value given in Fig. 2b of the present manuscript.

A further source of information is systematic time series of ocean temperature at sub-annual intervals, as a function of depth gridded to 0.1 yr intervals and 10 m vertical resolution, based on

540  systematic measurements along commercial shipping lanes through the HRXBT (high-resolution expendable bathythermograph) project https://www-hrx.ucsd.edu/index.html. An example is shown in Fig. 1 which is based on Sutton and Roemmich (2001), extended, constituting the mean over a section extending north from Auckland NZ, 37°S, to 30°S (600 km). Data such as these are highly pertinent to the depth at which the mixed layer, which is tightly coupled thermally to the

545  atmosphere, is decoupled from the deeper ocean. The time-depth profile of temperature $T$ from that data set, Fig 1a, shows the penetration of heat due to the annual cycle of insolation from the surface to a depth of about 100 m; this penetration of the surface temperature is reflected also in the deseasonalized temperature, $T_d$, Fig 1b. Perhaps most illustrative of the decoupling between the ML and the deep ocean is Fig. 1c, which shows sharp maximum in the gradient of

550  temperature with depth at about 75 to 100 m, the latter taken as the depth of the ML in the present study (75 m has been used by some earlier investigators as noted in the manuscript). Similar conclusions about the penetration of the annual signal can be drawn from other studies in this project (e.g., Wijffels and Meyers, 2004).

[Figure]

**Fig. 1**, Time-depth profiles of ocean temperature, deseasonalized temperature $T_d$, and the derivative of $T_d$ with depth, $dT_d/dz$. Mean over a section extending north from Auckland NZ, 37°S, to 30°S (600 km) evaluated from data of Sutton and Roemmich (2001), as extended. Data were provided by Philip Sutton and are available through the Scripps High Resolution XBT program (www-hrx.ucsd.edu)

I added text in support of the choice of 100 m as ML depth based in part on the above (lines 158-167 in the revision). This added text includes reference to this open discussion, line 164 in the Revision.

With respect specifically to the suggestion of the Commenter that the heat uptake in the first 300 m should be apportioned equally between 0-100 m, 100-200 m, and 200-300 m, Fig. 2 shows the temperature and the rate of temperature increase over these three intervals, again evaluated from the data of Sutton and Roemmich (2001, as extended). The slopes of the mean temperatures over the period 1986-2019, show that the apportionment of heat uptake in these three depth bands is 46 %, 31 %, and 23 %, respectively, that is, much greater heat uptake in the top 100 m, consistent with the treatment in the present manuscript that separates the ML from the deep ocean at 100 m.

[Figure]

**Fig. 2**. Deseasonalized ocean temperatures obtained from measurements between Auckland NZ, 37°S, to 30°S (600 km) averaged over the depth bands 0-100 m, 100-200 m, and 200-300 m for the period 1986-2019, together with least-squares fits, with indicated slopes (1-σ uncertainties in each of the slopes about 0.002 K yr⁻¹).

In sum, all these considerations lend strong support to the treatment of the uptake of excess heat by the global ocean in the top 100 m over the years 1960-2023 as a model for the uptake of excess $CO_2$ over the Anthropocene.

3. To my knowledge the term „piston velocity" is mainly (only?) used in the context of air-sea gas exchange, while it is here used for the exchange between mixed layer ocean and deep ocean.

~~*Response*. I accept the Commenter's concern of possible confusion between the term "piston velocity", as commonly used to denote the rate of transfer of gases between the atmosphere and the mixed layer ocean, and as used in the present manuscript to denote the rate of exchange, per area, of water between the mixed layer and the deep ocean. To avoid such confusion I would use the term "exchange velocity" in lieu of "piston velocity" to denote this exchange rate, but with the same meaning as in the manuscript.~~

On reflection I decided to stay with the term "piston velocity." None of the reviewers took exception to that use of the term. A sentence has been added to distinguish this "piston velocity" from that used to characteerize air sea exchange to avoid any possible confounding of the two quantities Line 250 in the revision:

"The piston velocity as employed here should not be confused with the quantity, also denoted piston velocity, characterizing the rate of diffusive transfer of a dissolved gas through the stagnant film between the air-ocean interface and the turbulent region away from the interface (SG06, p. 82)."

I thank the Commenter for raising the several questions and stimulating my thinking along these lines. I hope that my responses will resolve the Commenter's concerns.

**References (in the Open Discussion).**

Cheng, L., von Schuckmann, K., Miniére, A., Hakuba, M. Z., Purkey, S., Schmidt, G. A., and Pan, Y.: Ocean heat content in 2023, Nat. Rev. Earth Environ., 5, 232–234, https://doi.org/10.1038/s43017-024-00539-9, 2024a.

600   Cheng, L., Abraham, J., Trenberth, K. E., Boyer, T., Mann, M. E., Zhu, J., Wang, F., Yu, F., Locarnini, R., Fasullo, J., Zheng, F., Li, Y., Zhang, B., Wan, L., Chen, X., Wang, D., Feng, L., Song, X., Liu, Y., Reseghetti, F., Simoncelli, S., Gouretski, V., Chen, G., Mishonov, A., Reagan, J., Von Schuckmann, K., Pan, Y., Tan, Z., Zhu, Y., Wei, W., Li, G., Ren, Q., Cao, L., and Lu, Y.: New record ocean temperatures and related climate indicators in 2023, Adv. Atmos. Sci., 41, 1068–
605   1082, https://doi.org/10.1007/s00376-024-3378-5, 2024b.

Sutton, P. J. H. and Roemmich, D.: Ocean temperature climate off North-East New Zealand, New Zealand Journal of Marine and Freshwater Research, 35, 553-565, DOI: 10.1080/00288330.2001.9517022, 2001.

von Schuckmann, K., Minière, A., Gues, F., Cuesta-Valero, F. J., Kirchengast, G., Adusumilli, S.,
610   Straneo, F., Ablain, M., Allan, R. P., Barker, P. M., Beltrami, H., Blazquez, A., Boyer, T., Cheng, L., Church, J., Desbruyeres, D., Dolman, H., Domingues, C. M., García-García, A., Giglio, D., Gilson, J. E., Gorfer, M., Haimberger, L., Hakuba, M. Z., Hendricks, S., Hosoda, S., Johnson, G. C., Killick, R., King, B., Kolodziejczyk, N., Korosov, A., Krinner, G., Kuusela, M., Landerer, F. W., Langer, M., Lavergne, T., Lawrence, I., Li, Y., Lyman, J., Marti, F., Marzeion, B., Mayer, M.,
615   MacDougall, A. H., McDougall, T., Monselesan, D. P., Nitzbon, J., Otosaka, I., Peng, J., Purkey, S., Roemmich, D., Sato, K., Sato, K.,

Wijffels, S. and Meyers, G.: An intersection of oceanic waveguides: Variability in the Indonesian throughflow region, J. Phys. Oceanogr., 34, 1232–1253, 2004.

**CC2**: 'Reply on AC2', Peter Köhler, 18 Feb 2025

620   Some more details on my comment why I believe 14C is wrong in the model due to the missing 14C Suess effect.

In Figure 9 anthropogenic 14C stocks in different reservoirs are plotted. The value for 14C in the atmosphere (red line) is slightly increasing before 1950 (which then leads to an uptake of anthropogenic 14C in the ocean) which is compensated by negative stocks in the terrestrial 14C
625   stocks to keep the total balance at 0.

This is certainly wrong and probably only an expression that atmospheric $CO_2$ rose (and more $CO_2$ with the same Delta14CO2 gives you automatically a higher 14C stock in the atmosphere, which then the model brings into the ocean). It is wrong, because $CO_2$ rose due to anthropogenic $CO_2$ emissions, partly by land use change and partly by fossil fuel emissions. These fossil fuel
630   emissions are 14C free (and those from land use change slightly depleted in 14C due to radioactive decay), so the anthropogenic 14C stock in the atmosphere (and therefore also in the ocean) before 1950 should decrease, not increase. The total balance (integrated emissions, black

line in Fig. 9) should fall in that time period, referring to the fact that 14C-free CO2 emissions contribute negative values to the anthropogenic 14C stocks.

635 This is what I referred to with the missing 14C Suess effect. You might find more details, for example in Figs. S1, S2 in my previous paper (Köhler 2016, doi:10.1088/1748-9326/11/12/124016). This 14C Suess effect leads to decrease in  atmospheric D14C by about 20permil between 1820 and 1950 (Stuivers & Quay 1981).

Maybe there is a quick-fix for this problem (which I do not see right now), but to me it indicates
640 (as said in my previous comment) that an emission-driven setup (and not concentration-driven) is needed to depict 14C properly. If fixed most (if not all) of the problem would be solved, that to close the budget negative 14C stocks have to accumulate in the terrestrial C cycle (green line in Fig. 9).

Stuiver M & Quay P D 1981 Atmospheric 14C changes resulting from fossil fuel CO2 release and
645 cosmic ray flux variability Earth and Planetary Science Letters 53, 349–362.

**AC4**: 'Reply on Reply on AC2', Stephen E. Schwartz, 23 Feb 2025

I appreciate Dr. Köhler's continued interest in this manuscript. However he persists in several assertions, which I consider to be wholly incorrect, leading him to question the ability of a concentration-driven model to represent the evolution of anthropogenic $^{14}CO_2$, and in turn
650 anthropogenic $CO_2$. I examine his assertions here and present arguments as to why these assertions cannot be correct. As a consequence the concerns voiced by Dr. Köhler must be dismissed.

First I examine Dr. Köhler's statements regarding the response of the amount of atmospheric $^{14}CO_2$ to the essentially zero $^{14}CO_2$ emissions over the time period 1750-1950 prior to
655 anthropogenic emission of $^{14}CO_2$ from atmospheric weapons testing (pre-bomb era). Dr. Köhler states (correctly) that "These fossil fuel emissions are $^{14}C$ free." He goes on to state (of no consequence here) that the emissions "from land use change are slightly depleted in $^{14}C$ due to radioactive decay." He then draws the conclusion "so the anthropogenic $^{14}C$ stock in the atmosphere (and therefore also in the ocean) before 1950 *should decrease, not increase*"
660 [emphasis added]. He goes on to state, "The total balance (integrated emissions, black line in Fig. 9) should fall in that time period, referring to the fact that $^{14}C$-free $CO_2$*emissions contribute negative values to the anthropogenic $^{14}C$ stocks*" [emphasis added].

In refutation, I stress that emissions of $^{14}C$-free $CO_2$ cannot result in a decrease (or any change) in the totality of $^{14}C$ stocks in the closed system of atmosphere, ML, DO, and terrestrial biosphere,
665 TB). That is, the sum of the stocks of $^{14}C$ in the several compartments *must be constant over this time period*. This is reflected by the black curve in Fig. 9 of the manuscript, which indeed shows constant total $^{14}C$ in the system over the pre-bomb Anthropocene. The time series for the pre-bomb period is shown here in Fig. 1 with expanded vertical scale. As the $^{14}CO_2$ stock in the atmosphere is shown (by observation; see below) to have increased over this time period, and as
670 the concentration-driven model shows substantial net increase in $^{14}C$ stock in the DO (and lesser

increase in the ML stock) the increase in the stocks in these compartments for the closed system can be due only to net transfer of ¹⁴C from to these compartments from the TB, to maintain constant total amount of ¹⁴C in the system, as explicitly stated in the manuscript. (Radioactive decay is essentially negligible over this time period.)

[Figure]

675

Fig. 1. Time series of the stock of anthropogenic ¹⁴C in the several compartments of the present model in the pre-bomb Anthropocene. Atmospheric ¹⁴CO₂ is determined by observation; ¹⁴C in the mixed-layer ocean and the deep ocean are calculated from atmospheric ¹⁴CO₂ by the concentration-driven model developed here using the transfer coefficients determined in the present study. ¹⁴C in the terrestrial biosphere is evaluated as the negative sum of the stocks of the three other compartments for the four compartments comprising a closed system.

680

Secondly, Dr. Köhler goes on to speak to what he refers to as a "missing ¹⁴C Suess effect," noting that "This ¹⁴C Suess effect leads to decrease in atmospheric $\Delta$¹⁴C by about 20 permil between 1820 and 1950." Here I would emphasize that the ¹⁴C Suess effect speaks to the *isotopic abundance of ¹⁴CO₂ in the atmosphere relative to total mass of CO₂-carbon in the atmosphere, not to atmospheric stock of ¹⁴CO₂*. It is the *relative abundance* that decreased over the pre-bomb Anthropocene (the Suess effect). It would thus seem that Dr. Köhler is confounding the two measures of the amount of ¹⁴CO₂ in the atmosphere (and other reservoirs), the departure of the isotope ratio of ¹⁴C activity to C mass, relative to a standard, commonly expressed for atmospheric CO₂ as $\Delta$¹⁴CO₂, and the stock of ¹⁴C in the reservoir, which for atmospheric ¹⁴CO₂ is commonly expressed as number of ¹⁴C atoms, $n_{14CO2}$ in ¹⁴CO₂ in the global atmosphere. The two quantities are readily related, but as shown below exhibit qualitatively different time dependence over the pre-bomb Anthropocene, $\Delta$¹⁴CO₂ decreasing with time but $n_{14CO2}$ increasing with time. The reason for this is given here.

685

690

The quantities $n_{14CO2}$ and $\Delta$¹⁴CO₂ are related (Karlén et al., 1964; Stuiver, 1980) as

695

$$n_{14CO2} = f\left[\frac{1+\delta^{13}CO_2}{1-0.025}\right]^2 \left(1+\Delta^{14}CO_2\right)n_{CO2}$$

where $f$ is a constant, $1.176 \times 10^{-12}$; the square of the quantity in brackets, which accounts for mass-dependent isotopic fractionation, is nearly constant over the Anthropocene (1.037); and $n_{CO2}$ is the number of $CO_2$ molecules in the global atmosphere. The quantity $\Delta^{14}C$, the depletion or enrichment of $^{14}C$ relative to an absolute standard corrected for mass-dependent fractionation is defined (Stuiver, 1980, Eq 1) as

$$\Delta^{14}C = \left(1+\delta^{14}C\right)\left[\frac{1-0.025}{1+\delta^{13}C}\right]^2 - 1$$

The quantities $\delta^{13}C$ and $\delta^{14}C$ denote the fractional depletion or enrichment with respect to the respective standards. Equations (1) and (2) yield

$$n_{14CO2} = f\left(1+\delta^{14}C\right)n_{CO2}$$

Important here is the fact that $\delta^{14}C$ is based on the ratio of the amount of $^{14}C$ in a sample of atmospheric $CO_2$ (or, as is the case for historical data, in a tree-ring sample) to the mass of C in the sample. Thus $\delta^{14}C$ (and hence also $\Delta^{14}C$) can change because of changes in the amount of $^{14}CO_2$ in the atmosphere, the amount of total $CO_2$ in the atmosphere, or both. Over the pre-bomb Anthropocene $\Delta^{14}C$ decreased, Fig. 1a (the $^{14}CO_2$ Suess effect), but the amount of atmospheric $CO_2$, expressed as the mole fraction of $CO_2$ relative to dry air, $x_{CO2}$, increased (the Keeling effect), Fig. 1b; the number of $CO_2$ molecules in the global atmosphere is related to its mole fraction as $n_{CO2} = N_A n_{air} x_{CO2}$, where $N_A$ is the Avogadro constant ($6.022 \times 10^{23}$ molecules mole$^{-1}$) and $n_{air}$ is the number of moles in the global atmosphere $1.765 \times 10^{20}$ mol (Prather et al., 2012). When the two effects are combined to evaluate the number of molecules of $^{14}CO_2$, the increase in $n_{CO2}$ dominates over the decrease in $\Delta^{14}C$ and the amount of atmospheric $^{14}CO_2$, as quantified by $n_{14CO2}$, is seen to have increased, Fig. 2c.

[Figure]

**Fig. 2**. Contributions to the change in the amount of $^{14}CO_2$ in the global atmosphere over the years 1900 - 1950. (a) Departure (in units of parts per thousand, "per mil," ‰) of the ratio of $^{14}CO_2$ to total $CO_2$ in the atmosphere, corrected for fractionation and for radioactive decay subsequent to the year of growth, from the ratio of $^{14}C$ to C in the absolute standard, $\Delta^{14}CO_2$ (samples from Douglas Fir and Noble Fir trees from the US Pacific Northwest (43°7' - 47°46'N, 121°45' - 124°06'W), and an Alaskan Sitka spruce tree (58°N, 153°W); data tabulated in Stuiver et al., 1998). (b) Mole fraction of atmospheric $CO_2$ (mole $CO_2$ per mole of dry air), $x_{CO_2}$, in units of parts per million, ppm (data from Law Dome, Antarctica; Etheridge et al., 1996), (c) Number of $^{14}C$ atoms in atmospheric $^{14}CO_2$ in the global atmosphere. Modified from Schwartz et al. (2024).

The quantities shown in Fig. 2 are based on observation and the well understood relation between $\Delta^{14}CO_2$, and $n_{14CO_2}$. Thus there can be no question about the increase in the stock of atmospheric $^{14}CO_2$ over this period, Fig. 2c. Therefore Dr. Köhler's statement that "the anthropogenic $^{14}C$ stock in the atmosphere (and therefore also in the ocean) before 1950 should decrease, not increase" is refuted by observation.

Based on the considerations presented here Dr. Köhler's concern over the treatment in the present model of the evolution of $^{14}CO_2$, and more broadly of the evolution $CO_2$, must be wholly dismissed.

**References**

Etheridge, D. M., L. P. Steele, R. L. Langenfelds, R. J. Francey, J. M. Barnola, and V. I. Morgan (1996), Natural and anthropogenic changes in atmospheric $CO_2$ over the last 1000 years from air in Antarctic ice and firn, Journal of Geophysical Research: Atmospheres, **101**, 4115-4128, doi:10.1029/95JD03410.

Karlén, I., I. U. Olsson, P. Kllburg, and S. Kilici (1968), Absolute determination of the activity of two $^{14}C$ dating standards, Ark. Geofys., 4, 465–471.

Prather, M. J., C. D. Holmes, and J. Hsu (2012), Reactive greenhouse gas scenarios: Systematic exploration of uncertainties and the role of atmospheric chemistry, Geophysical Research Letters, 39(9), doi:10.1029/2012GL051440.

Schwartz, S.E., Hua, Q., Andrews, D.E., Keeling, R.F., Lehman, S.J., Turnbull, J.C., Reimer, P.J., Miller, J.B. and Meijer, H.A., Discussion: Presentation of Atmospheric $^{14}CO_2$ Data. Radiocarbon, 66(2), pp.386-399, 2024. DOI:10.1017/RDC.2024.27, 2024 .

Stuiver, M., Reimer, P.J. and Braziunas, T.F., 1998. High-precision radiocarbon age calibration for terrestrial and marine samples. Radiocarbon, 40, 1127-1151. https://www.cambridge.org/core/journals/radiocarbon/article/highprecisionradiocarbon-age-calibration-for-terrestrial-and-marine-samples/1660E9D7A43772ACBB56614C1DD09D46

Stuiver, M., 1980. Workshop on $^{14}C$ data reporting. Radiocarbon, 22(3), pp.964-966.

**CC3**: 'Reply on AC4', Peter Köhler, 26 Feb 2025

755 This comment by S. E Schwartz convinced me that my previous comments were maybe a bit unmature. I was trying to understand why the negative anomaly in 14C in terrestrial land was popping up in this approach which I think make its understanding at least difficult (and which still gives me troubles). By proposing some suggesting in the direction of the 14C Suess Effect I did not carefully consider my ideas in full depth and hopefully added not too much confusion to the

760 discussion. On the other hand, maybe this discussion clarified details also for other readers.

**AC5**: ['Reply on CC3'](), Stephen E. Schwartz, 04 Mar 2025

I thank Dr. Köhler for his gracious response to my previous reply. Perhaps the following might aid understanding the negative anomaly in terrestrial $^{14}$C over the pre-bomb Anthropocene. Here I refer to the reply "AC4: 'Reply on Reply on AC2', Stephen E. Schwartz, 23 Feb 2025".

765 It seems the point that Dr. Köhler finds troubling is understanding the decrease in $^{14}$C stock in the terrestrial biosphere (TB) during the pre-bomb Anthropocene reported in my study and shown in greater detail in Fig. 1 in the prior Reply. This decrease is a consequence of the tendency toward restoration of a dynamic equilibrium in response to a perturbation, specifically, here, the dynamic isotopic equilibrium between the atmosphere and the TB in response to the perturbation resulting

770 from the introduction of $^{14}$C-free $CO_2$ into the atmosphere associated with fossil fuel combustion. Let us assume (and it is a quite accurate assumption) that prior to this introduction of $^{14}$C-free $CO_2$ into the atmosphere there was isotopic equilibrium between the TB and the atmosphere, as manifested by the same value of $\Delta^{14}$C in both compartments. With introduction of $^{14}$C-free $CO_2$ into the atmosphere the value of $\Delta^{14}$C of atmospheric $CO_2$ decreased (the atmospheric $^{14}$C Suess

775 effect) as shown in Fig. 2a in the prior Reply, perturbing the dynamic isotopic equilibrium between the two compartments. The tendency toward restoration of that dynamic equilibrium would require adjustment of the stocks of $^{14}$C (and $^{12}$C) in the two compartments in the direction of achieving the same $\Delta^{14}$C in each compartment. Adjustment of $^{14}$C toward restoration of isotopic equilibrium would occur almost entirely by net transfer of $^{14}$C from the TB to the atmosphere with

780 the resultant observed decrease in $^{14}$C stock in the TB shown in Fig. 1 in the prior Reply.

Again I thank Dr. Köhler for his continued interest in this study. I also thank the Copernicus EGUsphere platform for accommodating back and forth discussion such as this between author and interested member of the community.

---

## Author Response (AR2)

**Author Responses 2025-0312 (revised).**
**Further responses 2025-0313**

**Associate editor decision: Publish subject to technical corrections**
by Jack Middelburg
**Public justification (visible to the public if the article is accepted and published)**:
Dear Dr. Schwartz:

Thank you for submitting your well-written and well-prepared revision to Biogeosciences. I have read it with much interest, and I am happy to inform you that your paper is now accepted for publication pending a few minor technical issues (see below). Simple and instructive models as yours are very useful for training the next-generation and for inclusion in integrated assessment models.

I thank the editor for the positive comments. I especially appreciate the thoughts expressed in his concluding sentence.

Minor corrections:
- P.33/Figure 9: I suggest modifying the Y-axis scale so that terrestrial biosphere 14C is fully presented

The cut off in the TB data is not due to the y axis range, but rather due to the fact that the Naegler data set for emissions, necessary to calculate TB, terminates at year 2005 as shown in the figure. There are other assessments of 14C emissions that extend beyond 2005, but they differ somewhat from the Naegler data set. Using some sort of blended data set to avoid step function jumps due to switching from one data set to another would permit extending the emissions beyond year 2005, but I chose not to go that direction for this paper because I thought that the introduction of those further data sets and description of the blending process would be a diversion here.

To allay any concern over the time series for TB being artificially cut off in the figure I will extend the axis range of the figure, as per the following:

[Figure]

I also modified the caption to explicitly read

emissions data for years 1951-2005

- Please check whether all your figures have the optimal color scheme.

Not clear to me what the editor refers to as "optimal color scheme". There are many figures with multiple traces, including light blue shading (and occasionally other colors). I tried to use color codes that would permit distinguishing the multiple traces.

I also used different blue colors, consistently across figures, to represent deep ocean, ML ocean, and total ocean. I would hope to retain that usage. Light blue shading for uncertainty in total ocean quantities. Likewise I consisently use deep green for TB, red for atmosphere, and black for emissions. I believe such a color scheme is an aid to the reader.

When showing results of others, I simply chose colors that could be distinguishable without much thought.

I prepare publication version of all my figures in Adobe Illustrator and could readily change colors of any traces; the legends are also in the figures.

If there is a guide to optimal color scheme, I would appreciate a link to that. For the moment unless there is a more specific guide I would leave the colors as they are.

I checked several figures at https://www.color-blindness.com/coblis-color-blindness-simulator/ for protanopia, which I understand to be the most common colorblindness. I was able (albeit with a bit of effort) to distinguish all curves in the figures I checked. I feel comfortable at this point to state that I have met the journal requirements of checking the figures.

- P. 48/line 1067: closing bracket needed after kmd

Thank you. Done

- P. 53/line 1165: the quantity beta is usually evaluated numerically (easier given the tools available), but analytical expressions do exist (see Hagens and Middelburg, 2016, GCA). I propose rephrasing… that is usually evaluated numerically from …

Thank you; will modify. Modified to read

The quantity $\beta$ is an equilibrium property of the $CO_2$–DIC system that is generally evaluated numerically from knowledge of the equilibrium constants for dissociation of $H_2CO_3$.

More broadly, I thank the editor for shepherding this paper through the review process, which was salutary. I am pleased that this review is nearing an end.

With best regards,

Jack Middelburg, Associate editor